

# Reinitialised versus continuous regional climate simulations using ALARO-0 coupled to the land surface model SURFEX

Julie Berckmans[1,2], Olivier Giot[1,2], Rozemien De Troch[1,3], Rafiq Hamdi[1], Reinhart Ceulemans[2], and Piet Termonia[1,3]

[1]Royal Meteorological Institute, Brussels, Belgium
[2]Centre of Excellence PLECO (Plant and Vegetation Ecology), Department of Biology, Antwerp University, Antwerp, Belgium
[3]Department of Physics and Astronomy, Ghent University, Ghent, Belgium

*Correspondence to:* Julie Berckmans (julie.berckmans@meteo.be)

**Abstract.** The potential of the implementation of the land surface model SURFEX in the atmospheric ALARO-0 model configuration of the ALADIN system is tested in a continuous regional climate simulation. This contribution is evaluated with respect to the regional climate simulated by the original setup of ALARO-0 with ISBA. Next, an assessment has been performed to evaluate the continuous setup with an upper air daily reinitialised setup, where the surface is kept in a freely
continuous mode. The results show that the introduction of SURFEX improves or has a neutral impact on the 2 m temperature and the daily total precipitation. More importantly, the use of an upper air daily reinitialised atmosphere outperforms the setup with a continuous atmosphere, for the 2 m temperature in winter and summer, and for the summer daily total precipitation. The differences between the two downscaling setups in the 2 m temperature and precipitation interact with the soil moisture. This coupling is strong for the continental areas, which motivates the use of a coupled land-atmosphere model to optimise the
representation of the climate.

## 1 Introduction

The first long-range simulation of the general circulation of the atmosphere was performed by Phillips (1956). Today it is still the primary tool for climate projections. However, due to limiting computer resources, the current horizontal resolution of 100-200 km is still coarse. A higher resolution and more spatial details can be obtained by nesting a regional climate model
(RCM), over a smaller domain, into a global climate model (GCM). This is also referred to as dynamical downscaling. The RCM uses the large-scale features from the GCM or from a global reanalysis as lateral boundary conditions (LBCs). This way the global features are translated into regional and local conditions over the region of interest (Giorgi, 2006). Hence, RCMs allow to run climate simulations over a smaller domain with higher horizontal resolution and with an affordable computing cost.

Since the late 60's, the Numerical Weather Predictions (NWP) community uses so-called high-resolution limited area models (LAM). Dickinson et al. (1989) were the first to use the numerical approach for a regional climate simulation. Their climate simulation used the NWP model in forecasting mode with frequent reinitialisations. To be able to run without these frequent



reinitialisations, i.e. in long-term continuous mode, the climate community has increasingly developed the representation of physical processes in RCMs. Still, this continuous simulation is the most common in the RCM community (Leung et al., 2003). Nonetheless, the simulated large-scale circulation deviates from the driving LBCs, when applying the continuous mode (von Storch et al., 2000). The accuracy of the dynamical downscaling has improved by using short-term reinitialisations (Qian et al.,

2003; Lo et al., 2008; Lucas-Picher et al., 2013). These authors showed the advantage of using short-term reinitialisations by reducing systematic errors. However, only few authors adopted this method, mainly because of its higher computational costs.

Most studies (Kotlarski et al., 2012; Qian et al., 2003; Lucas-Picher et al., 2013) dealing with the evaluation of reinitialised versus continuous climate simulations, covered only short time periods. The reinitialised simulation of the precipitation, in particular of the precipitation pattern, improved as compared to the continuous simulation (Kotlarski et al., 2012). Their analysis

covered only a short time period, one month in 2002 during a large flooding event in the Elbe river catchment. Changing the period of reinitialisation, from monthly to 10-daily, the experiments of Qian et al. (2003) showed a reduction in systematic errors for precipitation when using the 10-day reinitialisation. Even in a 20-year RCM simulation forced by reanalysis data, the sequence of events was better preserved by using frequent reinitialisations (Lucas-Picher et al., 2013).

The model used in this study is the ALARO-0 model configuration of the ALADIN system. This setup will short-handedly

referred to as ALARO-0 henceforth. This model configuration is used by the Royal Meteorological Institute of Belgium (RMI) for its operational numerical weather forecasts. ALARO-0 has already proven its ability for regional climate modelling with daily reinitialisations (Hamdi et al., 2012; De Troch et al., 2013). The model initially used the Interaction Soil-Biosphere-Atmosphere Interaction (ISBA) land surface scheme (Noilhan and Planton, 1989; Noilhan and Mahfouf, 1996). The setup of ALARO-0 with ISBA has been validated for continuous climate simulations and is now contributing to the EURO-CORDEX

project (Jacob et al., 2014; Giot et al., 2016). Meanwhile the more recent land surface scheme of Météo-France SURFace EXternalisée (SURFEX, Masson et al. (2013)) has been implemented in the ALARO-0 version. With respect to NWP applications, the introduction of SURFEX within ALARO-0 has shown neutral to positive effects on the 2 m temperature, 2 m relative humidity, 10 m wind speed and on the precipitation scores compared to the previously used ISBA scheme (Hamdi et al., 2014). Therefore, the evaluation of SURFEX within ALARO-0 is highly demanding for regional climate simulations.

Our study consists of three objectives. The first objective is to test the potential of the implementation of SURFEX within ALARO-0 with respect to the original setup of ALARO-0 with ISBA, in a continuous regional climate setup using boundary conditions from ERA-Interim. The second objective is to evaluate the continuous setup with an upper air daily reinitialised setup, where the surface is simulated continuously. Regarding the second setup, the boundary conditions of sea surface temperature (SST) were reinitialised daily together with the upper air. Therefore one expects to see improvements for the second

method over continental parts of the domain, but not so much in coastal areas. A study of a coupling with a sea model is outside the scope of this paper. Beside the two setups with a continuously simulated surface, a setup with a daily reinitialisation of the surface is not recommended since this setup limits the equilibrium of the surface physics (soil moisture and temperature), which is particularly desirable in long-term climate modelling (Giorgi and Mearns, 1999). The third objective is to examine the performance of the two downscaling approaches with respect to the land surface feedback, more specifically the soil mois-

ture feedback. It was hypothesized that the differences in temperature and precipitation between the downscaling approaches





are related to differences in soil moisture. The availability of soil moisture is related to the distribution of the energy fluxes (Seneviratne et al., 2010). Therefore, their daily cycle was analysed for particular locations in this study.

The land-atmosphere feedback was evaluated for the summer season only. The processes that control the soil moisture feedback to the climate system, result in different a coupling strength for summer and winter. It is strongest during summer,

when the surface is characterised by local interactions between land surface and the overlaying atmosphere (Koster et al., 2000; Douville and Chauvin, 2000; Koster et al., 2006). This is in contrast to the winter climate variability, which is primarily determined by large-scale circulation, as the atmospheric circulation anomalies are well correlated during winter with the SST anomalies (Kushnir et al., 2002).

The regional climate model and experimental design are described in section 2. Section 3 covers the observational datasets.

The results with respect to the first and second objective are addressed in section 4. Next, section 5 demonstrates the results with respect to the third objective. Finally, conclusions are given in section 6.

## 2   Model and experimental design

### 2.1   Model definition

The European Centre for Medium-Range Weather Forecasts (ECMWF) ERA-Interim reanalysis (Dee et al., 2011) was dy-

namically downscaled with the RCM ALARO-0. ERA-Interim provided lateral boundary conditions for the period of 1979 until now, at a horizontal resolution of $\approx 79$ km. The RCM ALARO-0 is a version of the Aire Limitée Adaptation Dynamique Développement International (ALADIN) model with improved physical parameterisations (Gerard et al., 2009). The ALADIN model is the LAM version of the Action de Recherche Petite Echelle Grande Echelle Integrated Forecast system (ARPEGE-IFS) (Bubnová et al., 1995; ALADIN International Team, 1997). ARPEGE is a global spectral model, with a Gaussian grid

for the grid-point calculation. The vertical discretization is done according to a following-terrain pressure hybrid coordinate. ALARO-0 has been developed with the ARPEGE Calcul Radiatif Avec Nebulosité (ACRANEB) scheme for radiation based on Ritter and Geleyn (1992).

We simulated the regional climate of ALARO-0 coupled to the land surface model SURFEX. Originally, the parameterisation for the land surface in ALARO-0 was simulated using the land surface model ISBA (Noilhan and Planton, 1989; Noilhan and

Mahfouf, 1996). Later on, the Town Energy Balance (TEB), a parameterisation for urban areas, was developed by Masson (2000). Recently, ISBA and TEB were combined and externalised into SURFEX, based on the algorithm of Best et al. (2004). SURFEX is based on a tiling approach. The tiles provide information on the surface fluxes for different types of surfaces: nature, town, inland water and ocean. A parameterisation exists for each component of the surface, more particularly ISBA for the nature type and TEB for the town type. Each tile is divided in different patches, according to the tile type. These patches

correspond to the plant functional types described in ECOCLIMAP version 2 (Faroux et al., 2013). ECOCLIMAP is a 1 km horizontal resolution global land cover database that is provided with SURFEX. It assigns the tile fraction to SURFEX together with the corresponding physical parameters. The soil component in ISBA provides vertical soil water and heat transfer by either a diffusion method (Boone and Wetzel, 1999), or a force restore method based on two or three layers (Noilhan and Planton,





1989). A two-layer version of ISBA was used in this contribution. The first layer is the surface superficial layer, that directly interacts with the atmosphere, and the second layer is the combined bulk surface and rooting layer, which is determined at the depth were soil moisture flux becomes negligible for a period of about one week and is thus more important as a reservoir for soil moisture during dry periods (Noilhan and Planton, 1989).

The atmospheric forcing was provided by ALARO-0 to SURFEX. The source code of SURFEX is implemented as a model library that is independent from the atmospheric model. At each model time step, the atmospheric forcing consisted of the upper-air temperature, specific humidity, wind speed and direction, atmospheric pressure computed at the lowest atmospheric level ($\approx$ 17 m) and incoming global radiation, incoming longwave radiation and total precipitation. Next, SURFEX simulated momentum, heat and water fluxes, and the standard meteorological variables as the 2 m temperature and relative humidity and
10 m wind using the interpolation method of Geleyn (1988).

  Recently, the performance of the ALARO-0 model driven by ERA-Interim has been evaluated within the Coordinated Regional Climate Downscaling Experiment on the European Domain (EURO-CORDEX, Jacob et al. (2014); Giot et al. (2016). The horizontal resolution of the EURO-CORDEX domain is 0.11° ($\approx$ 12 km) and it encompasses Europe, North Africa and a considerable part of Russia as well. In contrast to the EURO-CORDEX domain, the domain in our study is smaller. The
present study was done in the framework of another project, for which a domain encompassing Western Europe is sufficient. The domain is centered at 46.47° N and 2.58° E with a dimension of 149x149 horizontal grid points and spacing of 20 km in both horizontal axes (Fig. 1). Due to computational limitations, the domain horizontal resolution cannot be similar to the EURO-CORDEX horizontal resolution. The model consists of 46 vertical layers with a model top extending up to 72 km. ALARO uses a Davies (1976) relaxation zone of eight grid points irrespective of the resolution.

**2.2 Experimental design**

For the present study, a first downscaling simulation of ALARO-0 coupled to SURFEX was done in a continuous mode for both atmosphere and surface (hereafter called CON, Table 1). It started at 00UTC on 01 January 1990, and ran continuously until 00UTC on 1 January 2000. The first year was treated as a spin-up year, and the analysis period covers the 10-year period 01 January 1991 to 31 December 2000. The output was stored every 3 hrs for the atmospheric variables and daily for the
SURFEX variables, e.g. deep soil moisture. The SST was updated each month during the simulation to avoid discrepancies throughout the year. Although the 10-year length is arbitrary, it is sufficiently long to include some inter-annual variability and to generate a reasonable sample of extreme events. The use of a NWP model in a climate setting for the performance of extreme precipitation events for a 10-year period was recently demonstrated by Lindstedt et al. (2015).

  The second downscaling simulation of ALARO-0 coupled to SURFEX was initialised daily with ERA-Interim combined
with a continuous running surface (hereafter called FS, Table 1), starting at 12UTC on 01 March each year, with a spinup period of 3 months. This was done in parallel for all the years 1990 to 2000, and the analysis period covered the 10-year period 01 January 1991 to 31 December 2000. The output was stored every 3 hours for the atmospheric variables and daily for the SURFEX variables. Each daily simulation extended up to 60 hrs, of which the first 36 hrs were treated as spinup time. The last 24 hrs were saved for the atmospheric conditions. The soil variables evolved freely after the first initialisation and were never



corrected or nudged in the course of the simulation. This method allowed the surface conditions to be continuous, whereas the atmosphere was reinitialised daily.

A third simulation was applied for both CON and FS (Table 1). A 3-month summer period from 01 June 2000 to 31 August 2000 was simulated, but unlike previous simulations, the output from SURFEX was stored hourly. This allowed us to analyse the diurnal cycle of the parameters related to the surface radiation budget.

The first simulation was compared against the ALARO-0 simulations done within the framework of EURO-CORDEX, that used the land surface scheme ISBA. The output from the EURO-CORDEX simulations was subset to the time period of interest and regridded to the study domain and to a 20 km horizontal resolution (called hereafter CRDX, Table 1), to compare it with the results of both downscaling strategies. This comparison allowed to examine the effect of SURFEX on the climate mode, as was previously done for NWP (Hamdi et al., 2014).

The evaluation of the atmospheric variables was done for seven subdomains across Europe, to cover the spatial variability of the domain (Fig. 1). This was in agreement with the subdomains that were used in the EURO-CORDEX community (Kotlarski et al., 2014) and were defined earlier in the framework of the PRUDENCE project (Christensen et al., 2007). The subdomains used in this study are the British Isles (BI), the Iberian Peninsula (IP), Mid-Europe (ME), France (FR), the Alps (AL), the Mediterranean (MD) and Eastern Europe (EA). For both MD and EA, only part of their domains were used, as our domain is not covering the total subdomain, and also the relaxation zone was excluded.

## 3 Observational data

### 3.1 E-OBS gridded dataset

The results of the climate simulations were validated against E-OBS, a daily high-resolution gridded observational dataset (Haylock et al., 2008). The dataset consists of the daily mean temperature, the daily maximum and minimum temperature, and the daily precipitation sum. The most recent version v12.0 was selected on the 0.22° rotated pole grid, corresponding to a 25 km horizontal resolution in Europe. It covers the period 01 January 1950 to 30 June 2015. With respect to previous versions, some improvements include the new precipitation data series for countries southeast of the Baltic Sea, updated Slovakian series for all variables, updated Croatian series for all variables and a highly extended network for Catalonia, Spain. These improvements also concerned our area of interest and time period of interest. In order to validate the model data, the E-OBS data were bilinearly interpolated towards the ALARO-0 20 km domain. A careful interpretation of E-OBS was necessary, as this regridded non-homogeneously distributed network implied a smoothing out of extreme precipitation and consequently a large underestimation of the mean precipitation (Haylock et al., 2008).

### 3.2 Eddy covariance data

The FLUXNET database provides information on the energy exchanges between the ecosystem and the atmosphere (Baldocchi et al., 2001). FLUXNET is a global network, and consists of flux towers using the eddy covariance method to monitor carbon



dioxide and water vapor exchange rates, and energy flux densities. The technique exists since the late 1950s, yet only recently continuous flux measurements are possible. A number of stations were already part of a flux measurement network (Aubinet et al., 2000). However, only a few stations provided data for the first operating years and so, only the year 2000 was selected for the evaluation of the model with regard to the observations. The model was compared with the station that was nearest to the

model grid box. The model resolution is quite low, and thus the grid point consisted of more land covers beside the particular land cover of the station. Therefore, only FLUXNET stations were selected where the model represented more than 50 % of the corresponding land cover, to show energy fluxes that were representative for the particular land cover. The selected ecosystem towers (Fig. 1) were: (1) Vielsalm in Belgium, at an altitude of 491 m with a tower height of 40 m and mainly covered by deciduous broadleaved forest, and (2) Collelongo in Italy, at an altitude of 1645 m with a tower height of 32 m and mainly

covered by deciduous broadleaved forest.

## 4   Validation of the mean model state

### 4.1   Spatial distribution

#### 4.1.1   Daily mean 2 m temperature

The spatial distribution of the 10-year daily mean temperature bias of CRDX, CON and FS simulations are compared to E-OBS

(Fig. 2a,b), for the winter (DJF: December-January-February) and summer (JJA: June-July-August) season. The average biases during winter and summer for CRDX, CON and FS for the entire domain as well as for specific subdomains are presented in Table 2. CRDX simulates a cold bias in general with a pronounced orographic effect, for both winter and summer (Fig. 2c,d). Within the EURO-CORDEX ensembles, it was shown that the cold bias over the Iberian Peninsula was stronger compared to other subdomains (Giot et al., 2016), which is also presented here by a cold bias of -3.27 °C, though the Alps and the

Mediterranean also result in even stronger biases. The cold bias over the domain is less pronounced in summer than in winter, shown by the average bias of the total domain of -2.94 °C and -0.62 °C respectively. Moreover, the Iberian Peninsula and Mediterranean show very small biases, as a result of compensating cold and warm biases. Still, a strong summer bias of -1.77 °C is obvious at the Alps.

   With respect to CRDX, CON demonstrates a reduction of 20-30 % for the cold mean bias during winter for the Iberian

Peninsula and the Mediterranean (Table 2). Besides, the British Isles, Mid-Europe, France and Eastern Europe show a bias reduction of ≈ 1 °C or more than 40 % for CON compared to CRDX, resulting in a bias of -1 to -2 °C for these subdomains. In contrast to the largely improved winter bias (Fig. 2e), CON has a slightly positive and slightly negative to neutral impact on the summer bias compared to CRDX (Fig. 2f) Overall, the biases are smaller in summer compared to the winter biases. The largest bias reduction appears over Eastern Europe, with a reduction of nearly 60 %, while it is decreased for the Mediterranean

with about 30 %, and increased with 40-70 % for the British Isles, the Iberian Peninsula and Mid-Europe.

   The performance of the FS simulation is different from CON, both for winter and summer (Fig. 2g,h). The winter total mean bias is improved, resulting in -1.03 °C compared to -2.94 °C and -1.85 °C for CRDX and CON respectively (Table 2). For





the British Isles, Mid-Europe, France and Eastern Europe, the winter mean bias is reduced with ≈ 60-70 %, and it is reduced with ≈ 40 % for the Iberian Peninsula and the Mediterranean. Similarly to CON, the cold bias remains over the Alps, and is reduced with 30 %. During summer, the sign of the bias reverses from negative to positive for many subdomains. For the Iberian Peninsula and the Mediterranean, compensating biases result in a close to zero bias. Mid-Europe, France and Eastern

Europe are mainly characterised by a positive bias of around 1 °C. The Alps are much better presented by FS, compared to CON.

In summary, CON underestimates winter and summer 2 m temperature. With respect to CRDX, CON shows a positive effect in Europe during winter and neutral to positive effect in summer, despite being slightly enhanced for particular subdomains. However, the summer bias is largely improved in Eastern Europe when using CON compared to CRDX. The use of FS improves

the representation of the 2 m temperature for both winter and summer compared to CON.

### 4.1.2    Daily accumulated precipitation

The spatial distribution of the 10-year daily accumulated precipitation bias of CRDX, CON and FS are compared to E-OBS, for the winter and the summer seasons (Fig. 3). The mean biases during winter and summer for CRDX, CON and FS are presented for the entire domain as well as for the specific subdomains in Table 3. The precipitation pattern of E-OBS during winter shows

highest values of > 3 mm day$^{-1}$ over Portugal, northwestern Spain, western England, Scotland and Ireland, the Adriatic Coast and the northern flanks of the Alps (Fig. 3a,b). During summer, similar amounts of rainfall are concentrated over the Alps and the Carpathians, while lowest values of < 1 mm day$^{-1}$ at the Iberian Peninsula, the Mediterranean and northern Africa.

CRDX shows a general wet bias in winter and summer, except for northern Africa (Fig. 3c,d). More particularly, the overestimation is strongest in winter in the Mediterranean and Eastern Europe (46.74 % and 54.07 %) and during summer for

the Iberian Peninsula (49.35 %). In addition, excessive amounts of rainfall appear near the Adriatic coast, southern Italy, and southern France (Fig. 3c,d). Similarly to CRDX, CON overestimates winter precipitation, except for northern Africa (Fig. 3e). The largest wet bias in winter is present in the Mediterranean, while the bias in Eastern Europe is reduced with ≈ 37 % compared to CRDX (Table 3). The subdomains France and Mid-Europe show 10-20 % less overestimation of the precipitation as compared with CRDX (Table 3). During summer, the precipitation bias is much more different for CON compared to CRDX

(Fig. 3f). While CRDX results in a large overestimation of the precipitation, CON produces much smaller biases for most subdomains with a reduction of 10-100 %, except for the Alps and the Mediterranean (Table 3). However, the absolute values for precipitation in summer are close to zero for the Mediterranean, as it is characterised by a climate with dry summers (Fig. 3b).

The performance of FS is different than CON for both seasons (Fig. 3g,h). The wet bias during winter is slightly enhanced

for all subdomains (Table 3). However, the values are in the same order of magnitude for the continental subdomains. The deterioration is largest for the British Isles, the Iberian Peninsula and the Mediterranenan. Their wet bias degrades with over 100 %, corresponding to an absolute increase of 1.21 mm day$^{-1}$. These subdomains are exactly under the influence of the Atlantic Ocean and the Mediterranean Sea. As winter is mostly characterised by transient stratiform precipitation systems, the problem of the doubled precipitation bias for these subdomains arrives mainly from SSTs that are reinitialised daily instead of





simulated continuously. During summer however, the precipitation biases for the British Isles, the Iberian Peninsula, the Alps and the Mediterranean are very similar for FS compared to CON (Table 3). This contrasts to Mid-Europe, France and Eastern Europe, where the precipitation signal reverses and a dry bias occurs, though it is rather small (-6.92 % for Mid-Europe, -13.45 % for France, -8.38 % for Eastern Europe respectively). Still, the excessive amounts at the western coast of the UK and the

mountains are present and similar to CON.

In summary, the wet bias as represented by CRDX is reduced by using CON both in winter and summer. This reduction is strongest during summer. This is in line with earlier conclusions by Hamdi et al. (2014) showing that SURFEX reduces the total accumulated precipitation during summer compared to ISBA. The use of FS has a neutral impact on the winter precipitation with respect to CON, except for the Mediterranean, where the SSTs could be the factor explaining the difference. During

summer, FS improves the results, and introduces some small positive biases.

## 4.2 Mean annual cycle

### 4.2.1 Daily mean 2 m temperature

To validate specific subdomains within the larger domain on a monthly scale, the mean annual cycles of the downscaled simulations are compared with the observations (Fig. 4). We focused on the following subdomains (Fig. 1): (1) the Iberian

Peninsula at the western boundary of the domain with its warm and dry summer climate, (2) Mid-Europe with its transitional climate, and (3) Eastern Europe at the eastern boundary of the domain with its continental climate.

The daily mean 2 m temperature reaches about 23 °C at the Iberian Peninsula, while it rises to 20 °C in Mid-Europe and Eastern Europe (Fig. 4a,b,c). For the selected subdomains, the downscaled simulations show very similar autumn (SON) temperatures, but underestimate them with respect to E-OBS. For winter and spring (MAM), FS is closer to the observations,

whereas CON underestimates the 2 m temperatures. Regarding summer, the selected subdomains show different model behaviour with respect to the observations.

At the Iberian Peninsula, both CON and FS underestimate the 2 m summer temperature (Fig. 4a). However, FS is closer to the observations than CON, resulting in a summer temperature bias of only 1 °C compared to 2 °C by CON. For Mid-Europe, FS (CON) overestimates (underestimated) the summer 2 m temperature. Still, the yearly mean is very close to the observations

with FS (Fig. 4b). Similarly to Mid-Europe, FS (CON) slightly overestimates (underestimates) the summer 2 m temperature in Eastern Europe (Fig. 4c). Yet again, the yearly mean results in a very small difference using FS compared to the observations, while larger differences occur using CON.

In summary, the yearly mean temperature is underestimated by CON, while FS is very close to E-OBS. The difference between the downscaled simulations is about 1-2 °C. Along the selected subdomains, there are larger differences between the

simulations in Mid-Europe and Eastern Europe as compared to the Iberian Peninsula.



### 4.2.2 Daily accumulated precipitation

Similarly to temperature, the monthly means of the daily accumulated precipitation, averaged over the 10-year period, are shown in Fig. 4 for the Iberian Peninsula, Mid-Europe and Eastern Europe. When comparing the observations, the yearly cycle is most pronounced at the Iberian Peninsula, with minimum precipitation values of 0.5 mm day$^{-1}$ during summer, and

maximum precipitation values of $\approx$ 3 mm day$^{-1}$ during spring, autumn and beginning of the winter (Fig. 4d). The precipitation in Mid-Europe reaches highest values of $\approx$ 2.5 mm day$^{-1}$ during summer (Fig. 4e). The continental climate of Eastern Europe results in average values of 1 mm day$^{-1}$ for winter and spring, while most rainfall occurs in the summer with values up to 2.5 mm day$^{-1}$ (Fig. 4f).

    For the Iberian Peninsula, the yearly pattern of the downscaled simulations follows the pattern of E-OBS (Fig. 4d). The

model simulations represent an overestimation of the precipitation for all seasons. This overestimation is stronger in winter and in spring. For these two seasons, E-OBS shows an undercatch of the precipitation, which might amplify the model biases (Rauscher et al., 2010). CON and FS compare very well, resulting in mean yearly values of 1.94 mm day$^{-1}$ and 2.09 mm day$^{-1}$ respectively. In Mid-Europe, the model overestimates the precipitation for most of the year, except for summer (Fig. 4e). During summer, FS shows a large underestimation compared to CON, despite an overestimation during winter and spring. The

agreement of CON and FS is largest in autumn. The precipitation in Eastern Europe is overestimated by the model during most of the year, except for summer. (Fig. 4f) The agreement between CON and FS is largest during summer and autunm in Mid-Europe. The simulated winter months and spring show larger differences between the downscaling approaches, with higher precipitation by FS than CON.

    In summary, the downscaled simulations overestimate the precipitation, except for an underestimation during summer. On

a yearly basis, the differences between CON and FS are small, but on a monthly basis, the magnitude of differences depends strongly on the region of interest. There are larger differences for Mid-Europe and Eastern Europe compared to the small differences at the Iberian Peninsula.

## 5   Land-atmosphere feedback

The land-atmosphere feedback is strongest during summer, as the land surface is characterised by more local interactions

with the overlaying atmosphere (Seneviratne et al., 2010). Therefore, the following analysis focuses on this season only. We evaluated the differences in the 2 m temperature and the daily accumulated precipitation between the two downscaled simulations. We assumed two-way interactions between the precipitation/temperature and the soil moisture. This feedback mainly concerns the soil moisture at root zone, where the soil moisture impacts the climate by affecting the plants' transpiration (Dirmeyer et al., 2006; Lawrence and Slingo, 2005).



## 5.1 Effect of downscaling setup on atmospheric and surface variables

The average differences were evaluated between FS and CON for the daily mean 2 m temperature, the daily accumulated precipitation and the daily relative deep soil moisture (Fig. 5). The soil moisture is represented as the volumetric water content, which corresponds to the soil moisture volume to the total soil volume (expressed in [$m^3$ $m^{-3}$]). The second model layer was

selected to represent the deep soil moisture. Figure 5a shows that the temperature differences are mainly positive and the largest mean differences of > 2 °C are situated over France, Mid-Europe and Eastern Europe. The smallest differences of 0.5-1.5 °C are situated at the Iberian Peninsula, the Mediterranean, the British Isles and the Alps. The precipitation differences are mostly negative with values of -20 % to -40 % for Mid-Europe, and mixed positive and negative for the Iberian Peninsula and the Mediterranean (Fig. 5b). The extreme differences in northern Africa are related to boundary effects, as the relaxation zone was

not excluded here, hence they are not physical.

Finally, the soil moisture differences are primarily negative with 20 % drier soils for Mid-Europe, the British Isles and Eastern Europe, while no differences between FS and CON are present at the Iberian Peninsula and Mediterranean (Fig. 5c). We focused again on the previously selected subdomains with distinct climate regimes, as they represented the largest variation in the soil moisture results. For Mid-Europe and Eastern Europe, the deep soil moisture simulated by FS decreases more

sharply during the spring compared to the deep soil moisture simulated by CON, and reaches lower minima during summer (Fig. 6b,c). Besides, the values simulated by FS are not able to restore to the values simulated by CON towards the end of the year. This leads to drier soils in FS compared to CON for these particular subdomains. The soil moisture deficit simulated by FS could amplify the summer temperature extremes (Fischer et al., 2007). This is in contrast to the Iberian Peninsula, where the difference in deep soil moisture between CON and FS is small during spring and only enhances from the end of summer

onwards (Fig. 6a).

## 5.2 Soil moisture-temperature/precipitation feedback

We did not evaluate a one-way effect of the difference (between the downscaled simulations) of one variable on the difference of the other variable, i.e. a cause and effect relationship. Instead, we assumed a two-way interaction between the differences of soil moisture and the differences of temperature/precipitation. Therefore, a correlation analysis was applied, which looked

as follows. First, the mean difference was calculated between FS and CON for each summer. Then, the correlation in time per grid point was determined without any spatial correlation.

The temperature differences are mostly negatively correlated to the soil moisture differences (Fig. 7a), while the precipitation differences were mostly positively correlated to the soil moisture differences (Fig. 7b). A different feedback can be observed for France, Mid-Europe, the Alps and Eastern Europe versus the British Isles, the Iberian Peninsula and the Mediterranean

(Table 4). With respect to the correlation of soil moisture difference with 2 m temperature (precipitation) difference, the former subdomains show values of 0.32-0.55 (0.24-0.40), considerably larger than the values of 0.05-0.25 (0.15-0.17) for the latter subdomains (Table 4). Therefore, the coupling is stronger for the former subdomains compared to the latter ones. A contrast exists between on the one hand an area, including France, Mid-Europe, the Alps and Eastern Europe, where the climate is





sensitive to the choice of the downscaling setup, through a land-atmosphere feedback, and on the other hand an area, including the British Isles, the Iberian Peninsula and the Mediterranean, where this feedback is weaker and the climate is not impacted by the choice of the downscaling setup.

### 5.3 Distribution of energy fluxes

The soil moisture impacts the partitioning of the energy fluxes (Seneviratne et al., 2010). Therefore, the summer daily cycle of the energy fluxes was evaluated for selected stations from FLUXNET, and their corresponding model grid points. The station Vielsalm in Belgium (BE-Vie) is located in the region that shows a strong negative (positive) correlation between the soil moisture difference and temperature (precipitation) difference (Fig. 7). The other station Collelongo in Italy (IT-Col) is located in the region that shows low correlations between the variable differences.

The daily cycle of the net radiation, latent heat flux, sensible heat flux and soil heat flux for the observations and the downscaled simulations is shown in Fig. 8. The net radiation is well simulated for BE-Vie, compared to the observations (Fig. 8a,b). This is in contrast to IT-Col, for which a much lower net radiation is simulated (Fig. 8c,d). This is probably related to the location of IT-Col on a high elevation. The soil heat flux is overestimated by the model, which could be related to an overestimated soil temperature in the model. The simulation of the sensible and latent heat flux very much depended on the

location. For BE-Vie, both CON and FS resulted in higher fluxes compared to FLUXNET observations (Fig. 8a,b). However, while CON results in a higher latent than sensible heat flux, FS results in a higher sensible than latent heat flux. This relation can also be expressed by the Bowen Ratio (BR), the ratio between the sensible and the latent heat flux. When the value is lower (higher) than 1, the latent heat flux is higher (lower) than the sensible heat flux. The BR for BE-Vie at 12UTC was 1.20 for the observations, while it was 0.82 and 1.26 for CON and FS respectively. In other words, CON results in a higher latent

than sensible heat flux for BE-Vie, which does not agree with FS and the observations. Consequently, CON and FS simulate differently the energy heat fluxes and more importantly, FS is much better in simulating the correct partitioning of the fluxes, with respect to the observations. In contrast to BE-Vie, the simulated sensible heat flux at IT-Col by both CON and FS is lower than the latent heat flux, resulting a BR of 0.81 and 0.82 at 13UTC respectively, compared to a BR of 1.23 at 13UTC for the observations (Fig. 8c,d). Consequently, the energy heat fluxes at IT-Col are not influenced by the downscaling setup, which is

in agreement with the previous findings.

### 6 Conclusions

An assessment of two downscaling setups has been performed using the regional climate model ALARO-0 coupled to the land surface model SURFEX, with boundary conditions of ERA-Interim. We evaluated this contribution with respect to the regional climate simulated by the original setup of ALARO-0 with ISBA. The present study corroborates the finding of Hamdi et al.

(2014), that the use of the SURFEX scheme improves the performance of the ALARO-0 configuration of the ALADIN system. We compared the common used method of a continuous climate simulation with the newer method of frequently reinitialising the RCM simulation towards its driving field. FS outperforms CON for summer and winter 2 m temperature, and summer





precipitation. The winter precipitation by FS is slightly degraded with respect to CON, but is still similar to CRDX. However, the biases are doubled for the Brisish Isles, the Iberian Peninsula and the Mediterranean when using FS as compared to CON. The SSTs were not simulated freely, but reinitialised daily together with the upper air. Therefore, the problems occur around the islands and the peninsula, which are under the influence of the ocean and the Mediterranean Sea. More importantly, this

is not a problem for 2 m temperature, as it is mostly locally driven by the surface in contrast to the precipitation, which is influenced by transient stratiform systems during winter.

The differences in 2 m temperature and precipitation between the downscaling setups during summer are demonstrated by an interaction with the soil moisture. The drier soils represented by FS as compared to CON can be coupled to the atmospheric variables, and help reveal a land-atmosphere feedback. Furthermore, the coupling strength depends on the particular subdo-

main. A strong coupling exists for the continental part including France, Mid-Europe, the Alps and Eastern Europe, whereas a weak coupling exists for the British Isles, the Iberian Peninsula and the Mediterranean. In addition, the opposite partitioning of the latent and sensible heat fluxes of the two downscaling simulations at BE-Vie confirms this land-atmosphere feedback. Moreover, the comparison with the energy fluxes distribution at IT-Col supports our finding that the coupling strength has been impacted by the choice of the downscaling setup.

In conclusion, this study demonstrates that the method of an upper air daily reinitialised atmopshere with a free surface is superior over continental areas while the weakness of the SST reinitialisation should be tackled by implementing a sea or an ocean model.

*Code availability* The used ALADIN Codes, along with all related intellectual property rights, are owned by the Members

of the ALADIN consortium. Access to the ALADIN System, or elements thereof, can be granted upon request and for research purposes only. The used SURFEX Codes are freely available, together with the ECOCLIMAP database, at http://www.cnrm-game-meteo.fr/surfex///spip.php?rubrique8.

*Data availability* This study is based on large datasets written in .FA and .lfi format. The relevant output is exported to R datasets. Due to licensing restrictions, this model output is not made publicly available. However, for the purpose of the review,

the data can be made available for the editor and reviewer upon request, by contacting Julie Berckmans.

*Author contributions.* O. Giot and R. De Troch performed the model simulations CRDX. O. Giot and R. De Troch designed R-tools for the analysis. O. Giot designed the experiment CON. R. Hamdi designed the experiment FS and developed the model code for the implementation of SURFEX within ALARO-0. P. Termonia and R. Ceulemans provided overall guidance during the project. R. Ceulemans was the project contractor. J. Berckmans performed the model simulations CON and FS and analysed the results. J. Berckmans drafted the manuscript. All

co-authors contributed to the writing and the revising of the manuscript.

*Acknowledgements.* We acknowledge the E-OBS dataset from the EU-FP6 project ENSEMBLES (http://ensembles-eu.metoffice.com) and the data providers in the ECA&D project (http://www.ecad.eu). This work used eddy covariance data acquired and shared by the FLUXNET community, including these networks: AmeriFlux, AfriFlux, AsiaFlux, CarboAfrica, CarboEuropeIP, CarboItaly, CarboMont, ChinaFlux,





Fluxnet-Canada, GreenGrass, ICOS, KoFlux, LBA, NECC, OzFlux-TERN, TCOS-Siberia, and USCCC. The FLUXNET eddy covariance data processing and harmonization was carried out by the ICOS Ecosystem Thematic Center, AmeriFlux Management Project and Fluxdata project of FLUXNET, with the support of CDIAC, and the OzFlux, ChinaFlux and AsiaFlux offices. This research was funded by the Belgian Federal Science Policy Office under the BRAIN.be program as MASC contract #BR/121/A2. The authors also thank Annelies Duerinckx

5 (Royal Meteorological Institute of Belgium, Brussels) for the useful discussions.

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




**Table 1.** Overview on the experiments with ALARO coupled to SURFEX carried out in the present study and on the existing experiment with ALARO using ISBA.

| Acronym | Description | Land surface model | Downscaling setup | Reference | Historical Period |
|---|---|---|---|---|---|
| CRDX | ALARO-0 20 km | ISBA | Continuous atmosphere - continuous surface | E-OBS | 01-01-1991 - 31-12-2000 |
| CON | ALARO-0 20 km | SURFEX | Continuous atmosphere - continuous surface | E-OBS | 01-01-1991 - 31-12-2000 |
| FS | ALARO-0 20 km | SURFEX | Reinitialised atmosphere - continuous surface | E-OBS | 01-01-1991 - 31-12-2000 |
| CON-FS | ALARO-0 20 km | SURFEX | Both downscaling setups | FLUXNET | 01-06-2000 - 31-08-2000 |

**Table 2.** The daily mean 2 m temperature bias (°C) between the downscaled simulations and E-OBS for the total domain and the subdomains (BI, IP, FR, ME, AL, MD, EA) during DJF and JJA for the 10-year period 1991-2000.

| | | TOTAL | BI | IP | FR | ME | AL | MD | EA |
|---|---|---|---|---|---|---|---|---|---|
| DJF | CRDX | -2.94 | -2.08 | -3.27 | -2.73 | -2.49 | -3.98 | -3.35 | -2.07 |
| | CON | -1.85 | -1.21 | -2.24 | -1.54 | -1.20 | -2.99 | -2.55 | -1.13 |
| | FS | -1.03 | -0.40 | -1.30 | -0.71 | -0.36 | -2.10 | -1.38 | -0.42 |
| JJA | CRDX | -0.62 | -1.22 | -0.25 | -0.72 | -1.35 | -1.77 | -0.59 | -1.27 |
| | CON | -0.60 | -1.71 | -0.43 | -1.19 | -1.28 | -1.77 | -0.38 | -0.54 |
| | FS | 0.84 | -0.71 | 0.51 | 1.02 | 1.19 | -0.02 | 0.75 | 1.26 |

**Table 3.** The daily accumulated precipitation bias (%) between the downscaled simulations and E-OBS for the total domain and the subdomains (BI, IP, FR, ME, AL, MD, EA) during DJF and JJA for the 10-year period 1991-2000.

| | | TOTAL | BI | IP | FR | ME | AL | MD | EA |
|---|---|---|---|---|---|---|---|---|---|
| DJF | CRDX | 20.23 | -3.44 | 15.60 | 32.74 | 29.49 | 18.75 | 46.74 | 54.07 |
| | CON | 15.62 | 4.39 | 15.73 | 28.56 | 24.46 | 9.17 | 46.94 | 33.95 |
| | FS | 34.50 | 16.63 | 30.84 | 37.52 | 34.67 | 24.31 | 109.58 | 60.60 |
| JJA | CRDX | 25.68 | 30.48 | 49.35 | 23.22 | 30.78 | 28.45 | 23.47 | 31.01 |
| | CON | 11.23 | 25.98 | 11.75 | 11.99 | 11.78 | 32.50 | 64.70 | -2.40 |
| | FS | 3.19 | 18.73 | 13.14 | -6.92 | -13.45 | 23.49 | 56.61 | -8.38 |



**Table 4.** The correlation of the difference (FS-CON) in daily deep soil moisture (WG2) and the difference in daily 2 m temperature (T2M), and in daily accumulated precipitation(RR), for the total domain and the subdomains (BI, IP, FR, ME, AL, MD, EA) during JJA for the 10-year period 1991-2000.

|  | TOTAL | BI | IP | FR | ME | AL | MD | EA |
|---|---|---|---|---|---|---|---|---|
| COR:DIFF_WG2-DIFF_T2M | -0.19 | -0.05 | -0.25 | -0.55 | -0.32 | -0.38 | -0.16 | -0.40 |
| COR:DIFF_WG2-DIFF_RR | 0.21 | 0.15 | 0.15 | 0.40 | 0.30 | 0.33 | 0.17 | 0.24 |

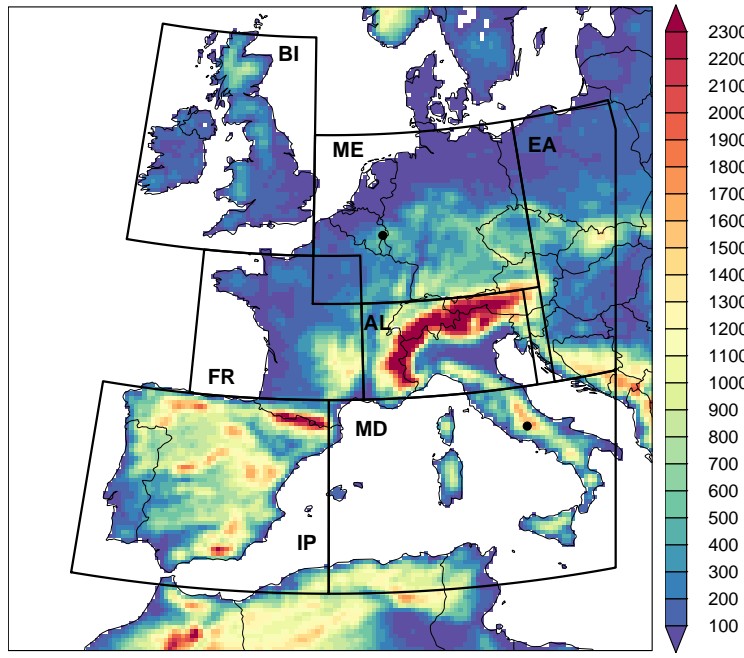

**Figure 1.** The domain on 20 km horizontal resolution and the subdomains (BI, IP, FR, ME, AL, MD, EA) based on the subdomains selected in the EURO-CORDEX framework. The color represents the orography in the ALARO+SURFEX setup. The two black dots represent the FLUXNET stations BE-Vie (Vielsalm,Belgium) and IT-Col (Collelongo,Italy).





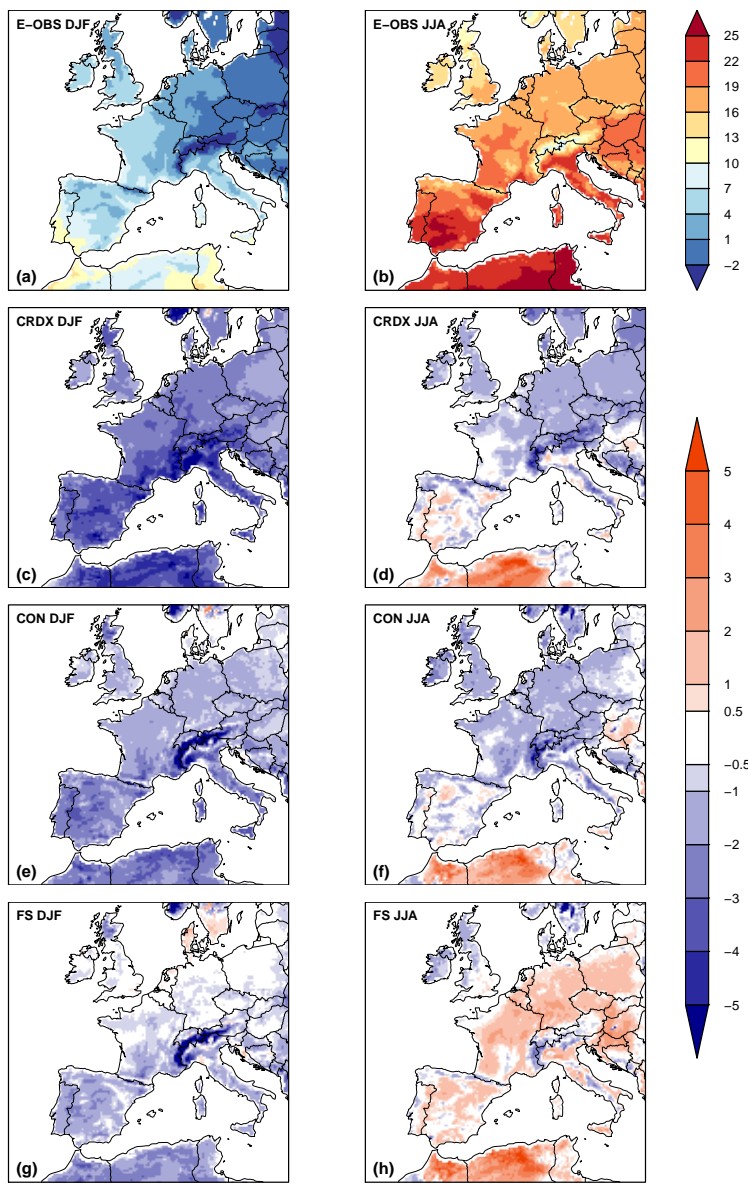

**Figure 2.** Daily mean 2 m temperature, absolute (°C) for E-OBS DJF (a) and JJA (b), and bias (°C) of the model with E-OBS for CRDX DJF (c) and JJA (d), for CON DJF (e) and JJA (f) and for FS DJF (g) and JJA (h), all at a 20 km horizontal resolution for a 10-year period 1991-2000.





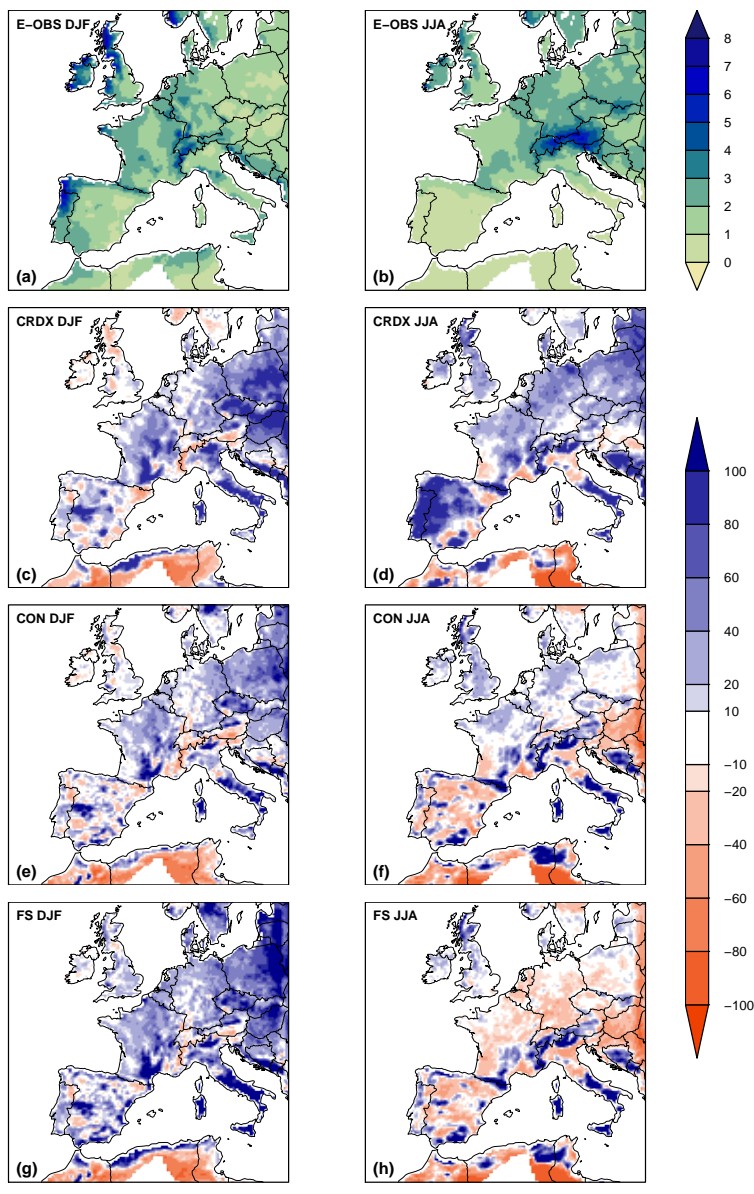

**Figure 3.** Daily accumulated precipitation, absolute (mm) for E-OBS DJF (a) and JJA (b), and bias (%) of the model with E-OBS for CRDX DJF (c) and JJA (d), for CON DJF (e) and JJA (f) and for FS DJF (g) and JJA (h), all at a 20 km horizontal resolution for a 10-year period 1991-2000.





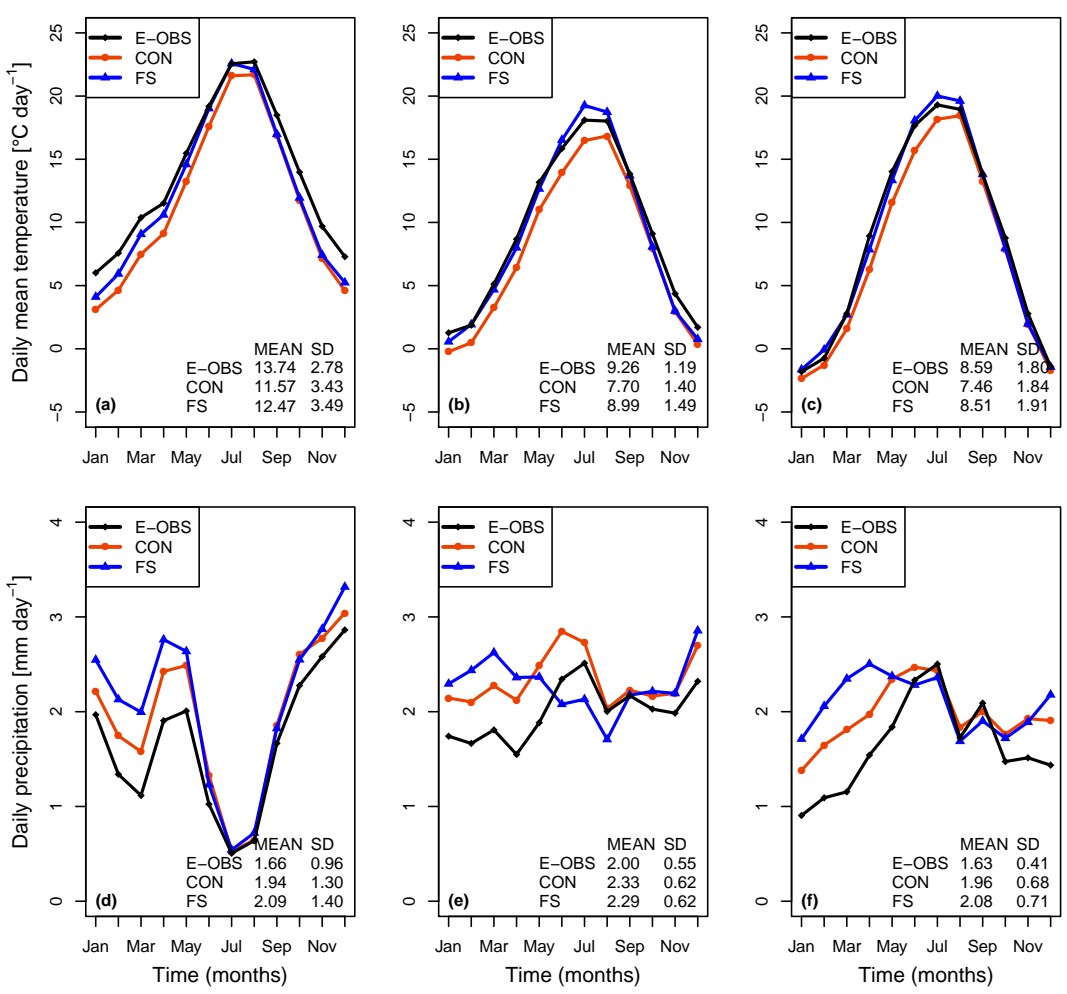

**Figure 4.** Mean annual cycle of the daily 2 m temperature (°C) and daily accumulated precipitation (mm) averaged over a 10-year period 1991-2000 with E-OBS, CON and FS for (a) the Iberian Peninsula, (b) Mid-Europe, and (c) Eastern Europe.





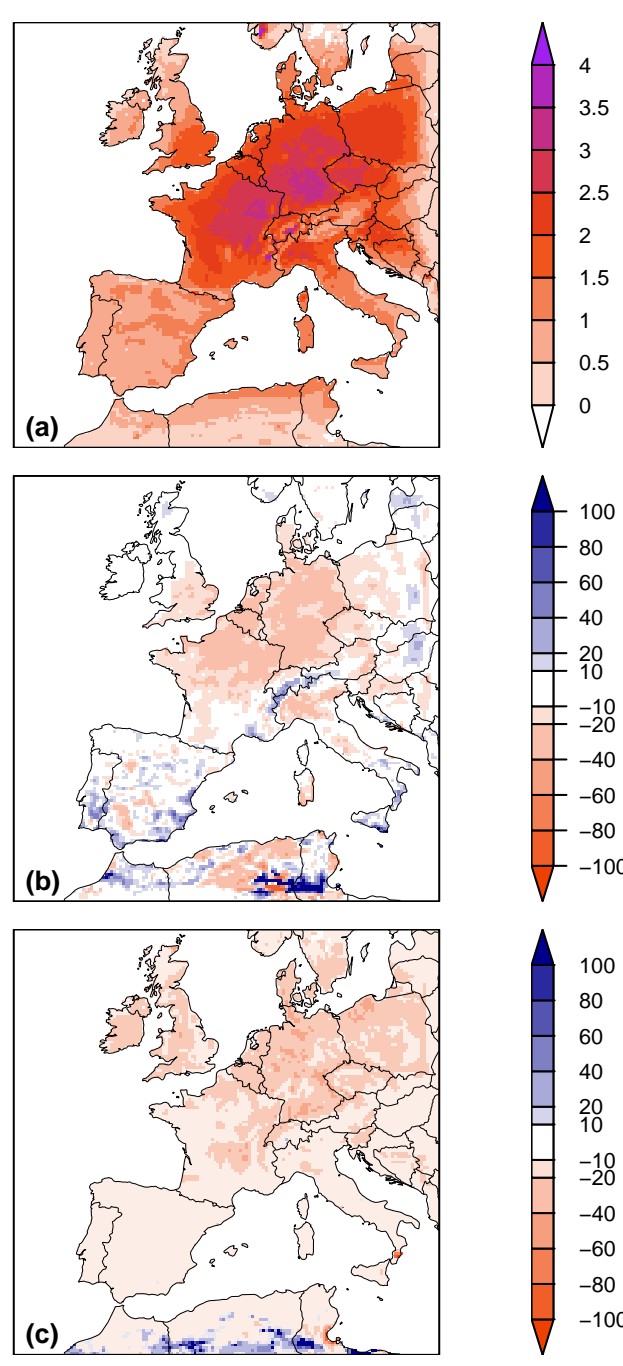

**Figure 5.** Difference between CON and FS for a 10-year JJA period 1991-2000 for (a) daily 2 m temperature ((FS-CON), in °C), (b) daily accumulated precipitation ((FS-CON/CON), in %) and (c) daily deep soil moisture ((FS-CON/CON), in %).





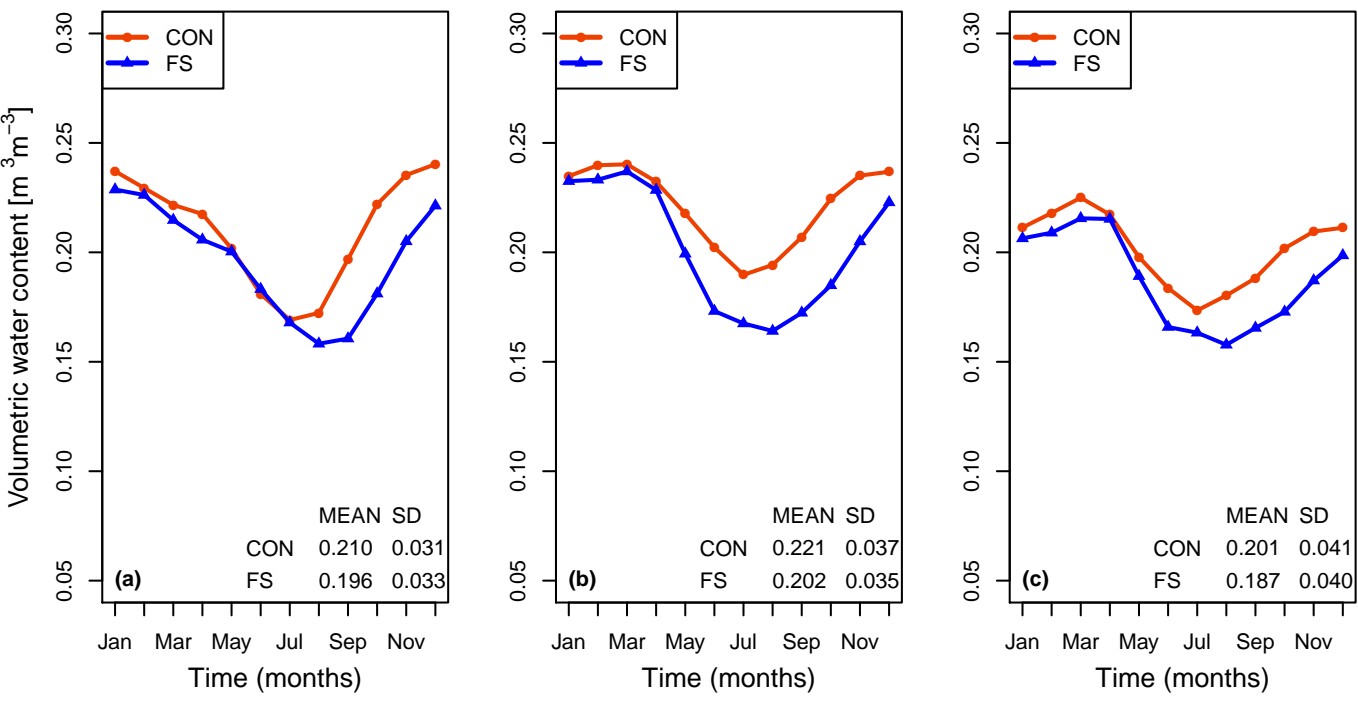

**Figure 6.** Mean annual cycle of the daily deep soil moisture (m³ m⁻³) averaged over a 10-year period 1991-2000 with E-OBS,CONF and FS for (a) the Iberian Peninsula, (b) Mid-Europe and (c) Eastern Europe.





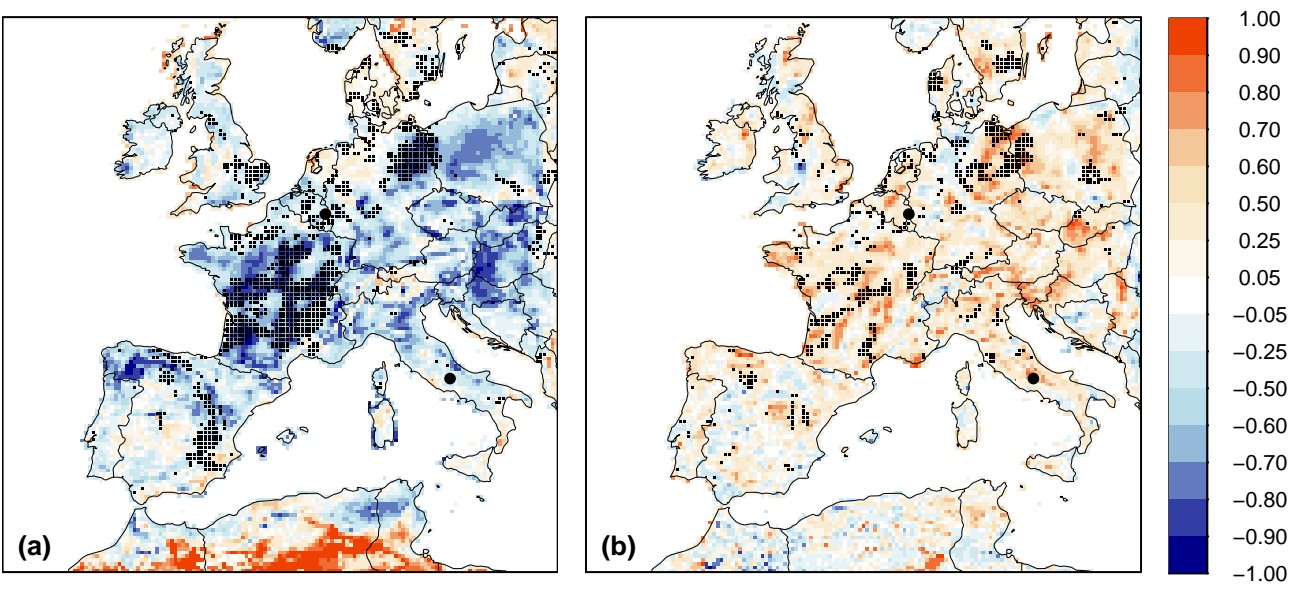

**Figure 7.** Correlation between the difference in daily deep soil moisture ((FS-CON/CON), in %) and (a) the difference in 2 m temperature ((FS-CON), in °C) and (b) the difference in accumulated precipitation ((FS-CON/CON), in %). Significant grid boxes at 0.05 level are identified with black dots.





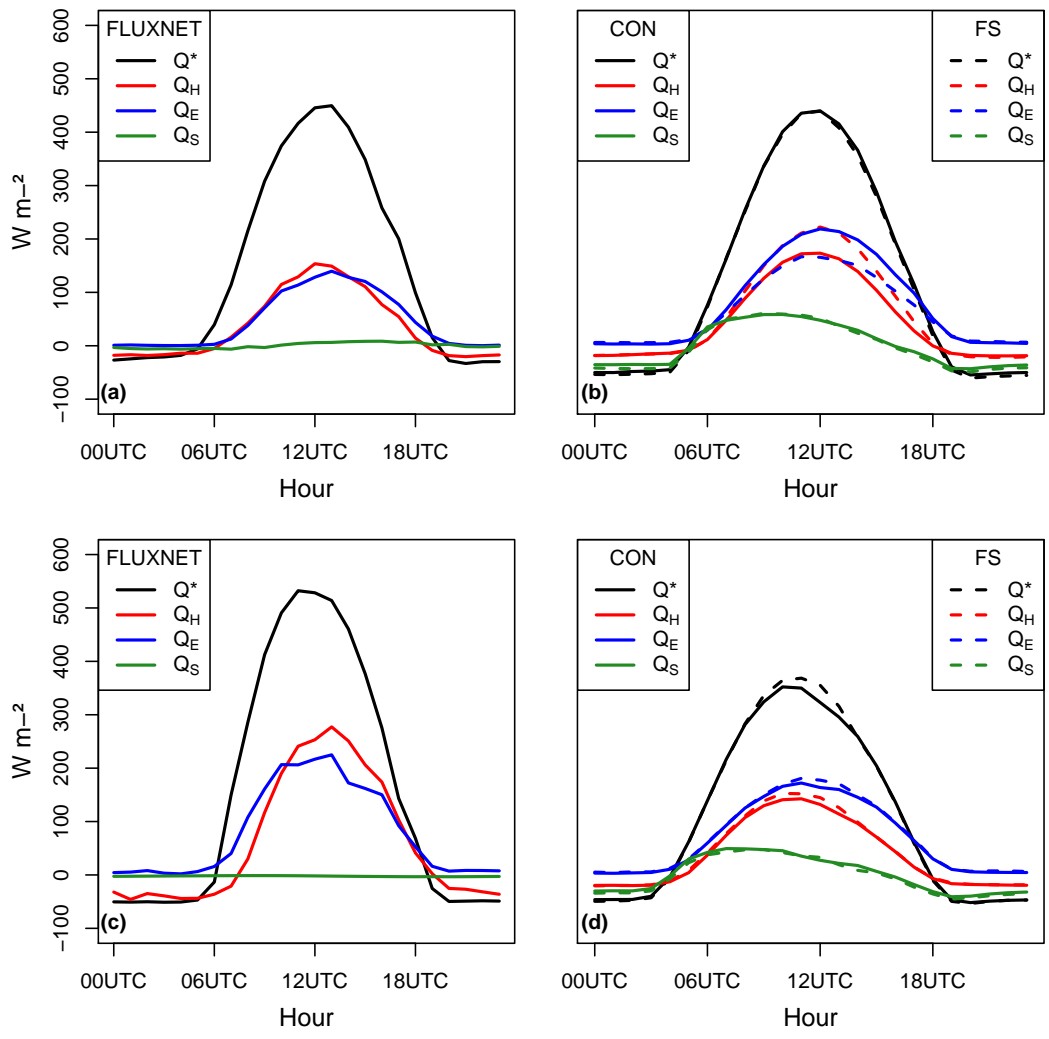

**Figure 8.** Daily cycle of the energy fluxes (W m$^{-2}$) in JJA 2000 for (a) FLUXNET observations at BE-Vie, (b) CON and FS at BE-Vie, (c) FLUXNET observations at IT-Col and (d) CON and FS at IT-Col.