# Peer review of "Reinitialised versus continuous regional climate simulations using ALARO-0 coupled to the land surface model SURFEXv5"

_Geoscientific Model Development, 2016_

## Short Comment (SC1) · 11 Aug 2016

Dear authors,

In my role as Executive editor of GMD, I would like to bring to your attention our Editorial version 1.1:

http://www.geosci-model-dev.net/8/3487/2015/gmd-8-3487-2015.html

This highlights some requirements of papers published in GMD, which is also available on the GMD website in the 'Manuscript Types' section:

http://www.geoscientific-model-development.net/submission/manuscript_types.html

In particular, please note that for your paper, the following requirement has not been met in the Discussions paper:

- "The main paper must give the model name and version number (or other unique identifier) in the title."

Please add a version number for ALARO-0 and SURFEX in the title upon your revised submission to GMD.

Yours,

Astrid Kerkweg

---

## Referee Comment (RC1) · Anonymous Referee #1 · 22 Aug 2016

Referee Comment on "Reinitialised versus continuous regional climate simulations using ALARO-0 coupled to the land surface model SURFEX" by Berckmans et al. Geosci. Model Dev. Discuss. doi:10.5194/gmd-2016-154

This study investigates the impact of different modeling techniques within the ALARO-0 Numerical Weather Prediction (NWP) model. These techniques differ in terms of whether the atmospheric boundary conditions (BCs) and sea surface temperatures (SSTs) are not updated during the simulation (continuous) or updated on a daily interval (reinitialized). I do believe that it is good to document model updates, particularly those concerning the best practice for initialization and running of forecast models. Therefore, I do feel that this manuscript fits within the scope of GMD.

There are however some major concerns over the experimental design and the explanation of the results that require significant amendment before this manuscript is suitable for publication.

(1) I was disappointed on the level of detail describing all the different models referred to in this manuscript: ALARO-0, SURFEX, ISBA, TEB and ALADIN and found that if I wanted a reasonable understanding of what the key parameterizations were or the runtime options used or the hierarchy of these models I would need to consult several other manuscripts…an exhausting exercise when there are so many models referred to here. Furthermore, I found it confusing to follow how the model is run. In RCMs like WRF, reanalysis (or GCM) data is used to update the BCs on a 6 hourly interval, which is necessary when running climate simulations (rather than short-term forecasts) to avoid drift. Therefore from the beginning I was confused by what the authors mean by "continuous" and "reinitialized" for an RCM when running a 10 year simulation. What variables are reinitialized, how are they reinitialized (at the boundaries or across the domain) and at what frequency? This information was not clearly articulated, making it difficult for someone to reproduce the experimental design.

(2) Although there are 3 objectives detailed in the introduction, I could see two possible aims of the manuscript: (a) Trying to show that forecast skill is improved when a new land surface scheme is added (CRDX vs. CON) – which is documented in Hamdi et al. 2014 (b) Trying to show that forecast skill is improved when reinitializing daily (FS vs. CON) If the aim is to present the benefits of the daily reinitialization, then perhaps excluding the CRDX results would improve the focus of the manuscript.

(3) There are different spin-up periods for FS (3 months) and CON (1 year). No explanation is provided for why the set up is different between the experiments. In particular, if the same land surface scheme is used, then best practice would be to have the same spin-up period for the land-surface state variables (soil moisture and soil temperature). If FS and CON have different spin-up lengths then how can the authors be sure that the differences in simulation skill are due to the "continuous/reinitialized" configuration

and not the spin-up? This would actually make me advocate that the whole experiment needs to be run again with same configuration (e.g. BCs, spin-up length, IC) so that differences in skill between "continuous" or "reinitialized" runtime modes can be fairly evaluated. At the moment I don't think this is really possible.

(4) I wasn't convinced at all by the analysis on the land-atmosphere feedback. Perhaps it would be good to consider the coupling metrics in detailed in Lorenz et al. (2015) that are suitable for the fully coupled simulations. It would be more convincing to calculate the coupling metrics for each experiment independently and then evaluate the difference between FS and CON to examine changes in coupling. However, is this analysis relevant here, given that the differences between CON and FS is the frequency that the lateral BCs are updated...not the land surface state as understood from Page 4/5: "The soil variables evolved freely after the first initialisation and were never corrected or nudged in the course of the simulation."

(5) I am a bit concerned about the limited number of sites used to evaluate the experiments against FLUXNET. This is likely due to the choice of simulation period (1991-2000) where FLUXNET data coverage is limited. It sounds like the authors were already aware of this limitation too. Perhaps a more recent simulation period would resolve this issue when more FLUXNET sites are available for a more rigorous validation. Alternatively, the authors could consider the LandFLUX (Mueller et al., 2013) or GLEAM (Miralles et al., 2013) datasets to validate the surface fluxes more comprehensively.

(6) The surface fluxes in the FS and CON configurations are also only evaluated in a second set of shorter simulations (3 months) rather than the original 10 year simulations. It would make better sense to evaluate the fluxes and land-atmosphere feedback in the 10 year simulations given that the purpose of the manuscript is to evaluate the simulation skill of long simulations. Unfortunately this provides the reader with the impression that the experimental design was either poorly designed or that a random bunch of simulations with different set-ups were cobbled together to evaluate the different runtime modes.

(7) Due to the writing style, I found the paper hard to read in many places. The structure also requires refinement, as there are many instances where information is provided in the wrong section that would be more useful in another.

Specific Comments:

Abstract:

- Here I got the impression that the manuscript was about evaluating the updated land surface scheme rather than the different running modes. Please revise the abstract to appropriately reflect the aim and scope of the manuscript and the key results.

- Please define all acronyms.

- It is perhaps not necessary to mention the ALADIN modeling system here to avoid overwhelming the reader with acronyms

- Sentence starting "This contribution . . ." perhaps better to say "We evaluate the dependence of simulation skill on the running mode (continuous or reinitialized) of the ALARO-0 model."

- Sentence starting with: "The results show that the introduction of SURFEX..." Could be revised to something like: "The results show that the SURFEX land surface scheme improves the simulation of 2 m temperature but has a negligible impact of the simulation skill of daily precipitation totals."

Introduction:

- The narrative introduces the reader to global climate modeling, numerical weather prediction, regional climate modeling, downscaling, limited area modeling. However there is insufficient detail on their differences particularly on the frequency that BCs are updated, what is meant by 'continuous' or how the 'reinitialization' is done (at the boundaries or across the domain). This needs simplifying, and can perhaps be resolved by limiting to just a few terms that are explicitly relevant to the study.

- It is never defined explicitly here or in the methods what "frequent reinitializations" and "continuous simulations" means. I got the impression that climate simulations were run using an RCM where the BCs were updated once a month (in CON) or daily (in FS) when most state of the art RCMs would be updating the BCs on a more frequent basis.

- Page 2 Line 6: Please check for spelling errors!

- Page 2 Line 14: "The model used in this study is the ALARO-0 model configuration of the ALADIN system." This won't mean much to those who have never used this model configuration. Perhaps this information is best in the methods where you describe the model and can then elaborate on the specific details of the ALARO-0 model configuration.

- Page 2 Line 17: Interaction appears to be used twice here. Please correct.

- Page 2 Line 19: Please provide the reference evaluating ALARO-0 with ISBA for continuous simulations.

- Page 2 Line 21: "has been implemented in the ALARO-0 version." Seems like the version number is missing at the end of the sentence.

- Page 2 Line 22: "the introduction of SURFEX with ALARO-0 has shown neutral to positive..." This phrasing is used a couple of times, perhaps its best to be more specific; which variables show no sensitivity, which ones are sensitive and what is the sign and magnitude?

- Page 2 Line 24: "Therefore the evaluation of SURFEX within ALARO-0 is highly demanding for regional climate simulations" Hamdi et al. 2014 already evaluates SURFEX within ALARO-0 so perhaps the authors need to be specific here by saying that Hamdi et al. evaluate SURFEX within ALARO-0 for NWP but this manuscript will evaluate the same model environment for longer simulations.

- Page 2 Line 27: "The second objective is to evaluate the continuous setup with an upper air daily reinitialized setup, where the surface is simulated continuously." I think this is where a lot of the confusion on terminology and model runtime configuration stems from. It would help if you can articulate what variables are continuous for each experiment, what variables are reinitialized and the frequency to which this is done. Could also add that information to Table 1.

- Page 2 Line 29: "Therefore one expects to see improvements for the second method over continental parts of the domain, but not so much in coastal areas." Why not in the coastal areas?

- Page 3 Line 2: Update to "Therefore the diurnal cycle of soil moisture was analyzed at particular locations in this study."

- Page 3 Line 3: This paragraph doesn't quite fit in with the previous narrative; please provide an explanation on why the focus is on the summer season (i.e. when the soil moisture limitation on evapotranspiration is greatest) for those less familiar with the land-atmosphere coupling literature.

Methods:

- Are all simulations run with the same ERA-Interim BCs? It would be useful to add this information to Table 1 and manuscript text.

- What data is used for the 'reinitializations' in FS?

- I found most of the narrative of Section 2.1 to go between describing the model/s and information that would be better placed in Section 2.2 Experimental Design – needs revising

- Page 3 Line 14: These sentences would be more suitable in Section 2.2 Experimental Design

- Page 3 Line 17: Another model is introduced but not used in this study. Please

remove to simplify the narrative.

- Page 3 Lines 14-22: There is no detail here on the microphysics, cumulus convection scheme or planetary boundary layer scheme. The reader is not provided with sufficient information on what the ALARO-0 configuration of the ALADIN modeling system actually means. This information is necessary in my opinion, particularly for someone not familiar with the model but interested in what was tested.

- Page 3 Line 20: Please change "following-terrain" to "terrain-following"

- Page 3 Lines 23-29: Based on what is written here I got the impression that SURFEX was basically ISBA with tiling and coupled to TEB. If the intention is to conduct a comparison between SURFEX and ISBA then the model descriptions need to be much more explicit on what the differences are between these models.

- Page 3 Line 30: 'ECOCLIMAP' Please define all acronyms the first time they are introduced

- Page 4 Lines 5-10: I gather that this is an explanation of the what variables are exchanged between the land surface model and the atmospheric model. The wording could be revised to make this easier to understand.

- Page 4 Lines 11-19: If the model is not run on the EURO-CORDEX domain then why provide detail on it? Also: "The present study was done in the framework of another project" please tell the reader what this project is.

- In Table 1 a CRDX experiment is listed but not defined in Section 2.2; which one is it?

- Page 4 Line 22: "It started at 00UTC on 01 January 1990, and ran continuously until 00UTC on 1 January 2000. The first year was treated as a spin-up year, and the analysis period covers the 10-year period 01 January 1991 to 31 December 2000." Please revise the inconsistency here as two different end dates are mentioned.

- Why are simulations run for 1991-2000 when a more recent period would enable

comparison to more FLUXNET sites?

- Why does CON have a spin-up of 1 year starting in January 1990 at 00UTC and why does FS have a 3 month spin-up with simulations starting in March each year at 12UTC? Usually the spin-up and start date should be the same unless the focus of the study is on how long a spin-up is necessary to maximize simulation skill which is not the aim of this paper!

- Why are atmospheric variables saved at a 3 hourly interval and land surface variables at daily? It would make sense to save them all at the same interval.

- Page 4 Line 33: "Each daily simulation extended up to 60 hours of which the first 36 hours were treated as spin-up" This seems to be contradicting the previous explanation where there is a 3 month spin-up for 1 year simulations! Please revise the description of how each experiment was run as it is not clear at the moment and limits the reader's ability to reproduce the experiment.

- Page 5 Line 3: "A third simulation was applied for both CON and FS" Technically third and fourth simulations?

- Page 5 the extra CON and FS simulations saving hourly output...why wasn't this done from the start rather than doing additional shorter simulations?

- Page 5 Line 6: "The first simulation..." Is this referring to the CRDX experiment?

- Page 5 Line 13: This is the first time PRUDENCE is mentioned – define acronym

- Page 5 Line 16: "relaxation zone was excluded" It looks like Figures 3, 5 and 7 need to be cropped to exclude the relaxation zone.

Observational data:

- Perhaps better to combine the model description, experimental design and observational data under one Methods, Models and Datasets section

- Page 5 Line 21: replace 'sum' with 'total'

- Page 5 Line 26: Usually one interpolates to the coarser resolution. i.e. interpolate the model data to the E-OBS resolution.

- Page 5 Line 27: replace 'implied' with 'applied'

- Page 5 Line 30: replace 'exchanges' with 'exchange'

- Page 6 Line 1: remove the sentence starting "The technique. . ."

- Page 6 Line 4: replace 'with regard to the' with 'against these'

- Page 6 Line 5: revise sentence starting "The model resolution is quite low..."

- In Section 5.3 a justification is provided on why the two FLUXNET sites were selected. It would be more appropriate to put that in Section 3.2.

Results – more general rather than line by line:

- The values in Table 2 and 3 are referred to more often than Figures 2 and 3. There are also several instances where it is not clear which results are being referenced. In particular, reporting the percentage change between experiments was quite confusing given that these values are not presented in either the Tables or Figures. This meant that I had to spend a lot of time checking where the values were coming from, or calculating the percentage changes myself. This could be resolved by adding detail in the manuscript text on the values shown in the figures. If the percentage change between experiments is quoted then please include these values in the Table or replace with something along the lines of: "CON has a larger bias of X relative to FS which has a bias of Y". This will make it easier for a reader to match the narrative to what is presented in Figures and Tables.

- There is a tendency to use words such as 'large', 'slight', 'excessive' or 'improved' in the narrative. Please be specific and insert detail. For example, "the bias improves" could be replaced with "the temperature bias decreases by X in experiment Y". There

are instances where this is done, but it has not been applied consistently. However doing so will make it easier for the reader to understand the results that are presented.

- Why are the temperature biases presented in degC but the precipitation biases presented as the percentage change? It would be preferable to use one approach consistently throughout the manuscript. In particular for the precipitation results, the % bias is often large when the observed precipitation is small – this is perhaps an instance where using the bias (MODEL minus OBS) would be more useful where small precipitation values inflate the value of the % bias.

- Page 7 last sentence: I don't agree with this, why would SSTs only be influential in winter?

-There is a tendency to start sentences with "Similarly to Experiment X, Experiment Y . . ." please use either: "Similar to Experiment X, Experiment Y. . ." or just start with "Experiment Y . . ."

- Sentences such as: "FS (CON) overestimated (underestimated) the summer 2 m temperature" are really hard to read and understand. It is actually easier to read: "FS overestimated and CON underestimated the summer 2 m temperature". Please revise all instances of this.

Section 5.1 and 5.2:

- Replace: "We assume two-way interactions" with "There are two-way interactions"

- At no point do we know the depth of the soil layers in the land surface scheme. This would be useful to know.

- It is not clear to me why the land-atmosphere feedback is evaluated by calculating the soil moisture-temperature (ST-T) correlation using (FS minus CON / CON). It would be better to calculate the SM-T correlation for FS, the SM-T correlation for CON and then the difference between these two estimates. I think this analysis needs to be redone. Lorenz et al. (2015) provides a good description of different coupling metrics that could

be applied to the data.

- Here I also think that the spin-up length will have some influence on evaluating the land-atmosphere feedback. If CON has a longer spin-up than FS of the soil moisture and soil temperature then the results are surely already biased as the land surface state fields will be more resolved in the simulations with the longer spin-up.

Conclusions:

- Page 11 Line 31: This definition of 'continuous' and 'reinitialized' simulation should be defined much earlier in the manuscript!

- Page 11 Line 7: "The differences in 2 m temperature and precipitation between the downscaling setups during summer are demonstrated by an interaction with the soil moisture." I don't agree with this because it's not actually calculated for the 10 year simulations where the temperature and precipitation differences are evaluated.

Tables and Figures

- Table 1 – perhaps remove the CON-FS line or separate into two.

- Table 3 – it would be easier if these values were presented as mm day-1 rather than the relative bias because some values are very high but might only be so because these are regions where the precipitation is very low: e.g. MD regions DJF FS experiment.

- Figure 2 – it would be good to include statistical significance as done in Figure 7; perhaps update the labels in the top left hand corner of panels c to h with CRDX – E-OBS

- Figure 3 – please put all panels in the same units it makes it easier to compare panels c to h with a and b. Is there missing data over Africa where there is a weird white triangle shape at the bottom of all panels? It is obvious in this figure that there are boundary affects for the domain and that the figures have not been cropped to

exclude the relaxation zone.

- Figure 4 – if the authors choose to keep the CRDX simulations then they should be included here and other figures. It would also be handy if dashed lines were added to each panel to delineate the seasonal breaks referred to in the manuscript text.

- Figure 5 – obvious boundary affect in all panels. Why is the absolute difference used for temperature but the relative difference used for precipitation and soil moisture? It would be easier if they all presented in the same way.

- Figure 7 – This needs to be redone to show SM-T and SM-P correlations for each experiment separately and then their difference.

- Figure 8 – It would be easier if FLUXNET, CON and FS were all on the same panel to directly compare differences. QS looks very different between the observations and CON and FS, are the authors certain that they are comparing like for like? QS looks like a flat line in panel c – check if there is a plotting error but this may just be because QS is very small relative to the axis scale or that it is not measured. . .

References:

- Lorenz et al. 2015: Intraseasonal versus interannual measures of land-atmosphere coupling strength in a global climate model: GLACE-1 versus GLACE-CMIP5 experiments in ACCESS1.3b. J. Hydrometeor., 16, 2276-2295.

- Miralles et al. 2011: Global land-surface evaporation estimated from satellite-based observations. Hydrol. Earth Syst. Sci., 15, 453–469.

- Mueller et al. 2013: Benchmark products for land evapotranspiration: LandFlux-EVAL multi-dataset synthesis. Hydrol. Earth Syst. Sci., 17, 3707-3720.

---

## Referee Comment (RC2) · Anonymous Referee #2 · 7 Sep 2016

The manuscript has two questions in mind: (1) Is the land surface model SURFEX better than ISBA in climate application with the modelling system ALADIN in ALARO-0 set-up? (2) Is it better to do regional climate simulations or dynamical downscaling (i.e. with re-initialization of the regional atmosphere and land surface) or some mix (e.g., continuous land surface simulation with atmosphere and sea surface temperature re-initialization)? In my opinion these questions are only weakly linked. Of course, if you want to apply continuous land surface simulation there is need to use the best available land surface model (a bad land surface model might render a regional climate simulation useless), but a better land surface model is better even in an NWP context. Therefore, I suggest to focus on one of the questions and I find the second question

more interesting.

The manuscript claims that SURFEX is better than ISBA. It has been shown in NWP context (given reference Hamdi et al., 2014), but the authors target climate time scales. They compare an available CORDEX simulation with ISBA against a new climate simulation with SURFEX which was done using a smaller simulation domain. Different domain sizes limit the comparability crucially. Both simulations were driven by the reanalysis ERA-Interim. Therefore, I would expect limited-area simulations are potentially better with a smaller domain. The presented results are not conclusive.

The authors cite many re-initialization vs regional climate simulation experiments. Even with ALADIN such a re-initialization experiment has been published, but for a 3-monthly period only (Beck et al. 2004). I suggest to make the manuscript more interesting and publishable by doing the list of re-initialization experiments more exhaustive by adding (a) full re-initialization (i.e. with SURFEX re-initialization) and (b) blending (i.e. re-initialization of the large atmospheric scales doing "climate" for the smaller scales, see Beck et al. 2004). Finally, perhaps too much for one paper, I think the ultimate criterium will be how the set-up performs with GCM forcing.

Beck, A., Ahrens, B., and Stadlbacher, K. (2004). Impact of nesting strategies in dynamical downscaling of reanalysis data. Geophysical Research Letters, 31, pp. 5. doi:10.1029/2004GL020115

---

## Author Comment (AC1) · 4 Nov 2016

[1,2]JulieBerckmans [1,2]OlivierGiot [1,3]RozemienDe Troch [1,3]RafiqHamdi [2]Reinhart­Ceulemans [1,3]PietTermonia

[1]Royal Meteorological Institute, Brussels, Belgium [2]Centre of Excellence PLECO (Plant and Vegetation Ecology), Department of Biology, University of Antwerp, Antwerp, Belgium [3]Department of Physics and Astronomy, Ghent University, Ghent, Belgium

11

November 4, 2016

Dear Ms. Astrid Kerkweg,

thank you for this notice.

The version number of ALARO is 0, hereafter called ALARO-0 and the version number of SURFEX version 5 has now been added as well to the title: "Reinitialised versus continuous regional climate simulations using ALARO-0 coupled to the land surface model SURFEXv5".
The manuscript version with the updated title is supplemented as .pdf.

Kind regards,

Julie Berckmans and co-authors

**Supplement:**

[revised manuscript text omitted]

5 model SURFEX, with additional parameterisations for other surface types than nature, has been implemented in the ALARO-0 model. With respect to NWP applications, the introduction of SURFEXv5 within ALARO-0 has shown neutral effects on the winter 2 m temperature and on the vertical profile of the wind speed. However, it has shown positive effects on the summer 2 m temperature, 2 m relative humidity, and resulted in improved precipitation scores compared to the previously used ISBA model (Hamdi et al., 2014). Next to the validation of this setup for NWP, the implementation of SURFEXv5 within ALARO-0

10 is highly demanding for long-term climate simulations. In this study, SURFEX uses the two-layer force restore method for ISBA. The first layer is the surface superficial layer, that directly interacts with the atmosphere, and the second layer is the combined bulk surface and rooting layer, which is determined at the depth were soil moisture flux becomes negligible for a period of about one week and is thus more important as a reservoir for soil moisture during dry periods (Noilhan and Planton, 1989).

15 SURFEX is based on a tiling approach. The tiles provide information on the surface fluxes according to the type of surface: nature, town, inland water and ocean. The initial parameterisation ISBA for the nature tile was conserved, and parameterisations for the other surface tiles were added, such as the Town Energy Balance scheme (TEB, Masson, 2000) for the town tile. TEB uses a canopy approach with three urban energy budgets for the layers roof, wall and road. The ISBA and TEB schemes were combined, together with parameterisation schemes for inland water and oceans, and externalised, based on the algorithm of

20 Best et al. (2004). Each tile is divided in different patches, according to the tile type. These patches correspond to the plant functional types described in ECOCLIMAP (Masson et al., 2003). ECOCLIMAP is a 1 km horizontal resolution global land cover database and assigns the tile fraction and corresponding physical parameters to SURFEX.

**2.2 Experimental design**

The regional climate model was driven by initial and lateral boundary conditions provided by the ERA-Interim reanalysis,

25 available at a horizontal resolution of ca. 79 km. The Davies (1976) relaxation zone consisted of eight grid points irrespective of the resolution. The zonal and meridional wind components, atmospheric temperature, specific humidity, surface pressure and surface components were provided every 6 hrs as lateral boundary conditions and interpolated hourly. They were introduced as initial conditions accross the domain. A spin-up time was considered for the model to reach equilibrium between the lateral boundary conditions and the internal model physics (Giorgi and Mearns, 1999). For the sake of a good understanding, the

30 following description makes a distinction between atmospheric spin-up time, typically of a few days, and surface spin-up time, typically of a few months to one year. The analysis covered a 10-year period from 00UTC on 01 January 1991 to 00UTC on 01 January 2001. Although the 10-year length is arbitrary, it is sufficiently long to include some inter-annual variability and to generate a reasonable sample of extreme events. The use of a NWP model in a long-term climate setting for the performance of extreme precipitation events for a 10-year period was recently demonstrated (Lindstedt et al., 2015). To evaluate the sensitivity

of the model to the update frequency of the initial conditions, three types of downscaling approaches were conducted with ALARO-0 coupled to SURFEXv5.

The first downscaling approach was done by simulating the model in a continuous mode for both the atmosphere and the surface (hereafter called CON ("CONtinuous"), Fig. 2). The model was simulated from 00UTC on 01 January 1990, and ran continuously until 00UTC on 01 January 2001. The first year was treated as both atmospheric and surface spin-up time, and was excluded from the analysis. The simulations were interrupted and restarted monthly to allow for SSTs to be updated. Other surface parameters that were updated monthly using the climatological values from ECOCLIMAP were the vegetation fraction, surface roughness length, surface emissivity, surface albedo, sand and clay fractions.

In the second downscaling approach, the model was reinitialised daily for both the atmosphere as the surface (herafter called DRI ("Daily ReInitialisation"), Fig. 2). The model started at 12UTC on 01 January 1991, and each reinitialisation ran for 60 hrs. The first 36 hours were treated as atmospheric spin-up time, and were excluded from the analysis. By applying this downscaling approach, the regional model stays close to the driving fields (von Storch et al., 2000). As the driving fields provided daily reanalysed data, a spin-up for the surface was redundant.

The third downscaling approach tries to find the best compromise between previous approaches. The atmosphere was reinitalised daily and the surface was simulated continuously with one single initialisation (hereafter called FS ("Free Surface"), Fig. 2). This allowed the model to simulate the atmospheric fields close to the driving fields, together with a surface in equilibrium state. The model was simulated from 12UTC on 01 March 1990 until 31 May 1991, and the atmosphere was reinitialised daily for a simulation time of 60 hrs. The first 36 hrs were treated as atmospheric spin-up time, and were excluded from the analysis. The surface conditions were kept continuous and joined after the atmospheric spin-up time with the surface conditions of the previous daily simulation. In contrast to the atmospheric spin-up time, the surface spin-up lasted from 01 March 1990 until 31 May 1990, and this 3-monthly period was excluded from the analysis. Although CON required one year spin-up time, 3 months were sufficient for the FS deep soil moisture to reach equilibrium state, when starting in March (not shown). The simulations were done in parallel for each year from 1990 to 2000, and the 3 monthly spin-up time was replaced by the analysis of the previous year.

The model output at every 3 hrs was used for the model evaluation. The evaluation of atmospheric variables for winter and summer was done for seven subdomains across Europe, to cover the spatial variability of the domain (Fig. 1). This was in agreement with the subdomains that were used in the EURO-CORDEX community (Kotlarski et al., 2014) and that were defined earlier in the framework of the project "Prediction of Regional scenarios and Uncertainties for Defining European Climate change risks and Effects" (PRUDENCE) (Christensen et al., 2007). The subdomains used in this study were the British Isles (BI), the Iberian Peninsula (IP), Mid-Europe (ME), France (FR), the Alps (AL), the Mediterranean (MD) and Eastern Europe (EA). For the subdomains IP, ME, and EA, the yearly cycle of the atmospheric variables was evaluated. These selected subdomains covered a range of climatic regimes. Additionally, the surface energy fluxes were analysed. As land-surface processes play an important role primarily during summer, the model output was stored at every hour for the summer period of June-July-August (JJA) during the 10-year period. We evaluated the partitioning of the sensible and latent heat fluxes by the daily maximum Bowen Ratio (BR, Bowen, 1926) for the summer periods from 1996 to 2000 for the total study domain,

and compared the selected FLUXNET stations with their corresponding model grid points. The corresponding daily maximum BRs were analysed for the 10-year summer period from 1991 to 2000. When the value is lower (higher) than 1, the latent heat flux is higher (lower) than the sensible heat flux. The diurnal cycles of all surface energy fluxes were analysed and validated against observations.

**2.3 Observational reference data**

The results of the climate simulations were validated against E-OBS, a daily high-resolution gridded observational dataset (Haylock et al., 2008). The dataset consists of the daily mean temperature, the daily maximum and minimum temperature, and the daily precipitation total. The most recent version v12.0 was selected on the $0.22°$ rotated pole grid, corresponding to a 25 km horizontal resolution in Europe. It covers the period 01 January 1950 to 30 June 2015. With respect to previous versions of E-OBS, some improvements include the new precipitation data series for countries southeast of the Baltic Sea, updated Slovakian series for all variables, updated Croatian series for all variables and a highly extended network for Catalonia, Spain. These improvements also concerned our area of interest and time period of interest. In order to validate the model data, the ALARO-0 data at 20 km horizontal resolution were bilinearly interpolated towards E-OBS at 25 km horizontal resolution and replotted to our study domain. A careful interpretation of E-OBS was necessary, as this regridded non-homogeneously distributed network applied a smoothing out of extreme precipitation and consequently a large underestimation of the mean precipitation (Haylock et al., 2008).

For the validation of the surface fluxes distribution in the model, we used measurements from the FLUXNET Level 3 flux tower database (Baldocchi et al., 2001). It provides information on the energy exchange between the ecosystem and the atmosphere. FLUXNET is a global network, and consists of flux towers using the eddy covariance method to monitor carbon dioxide and water vapor exchange rates, and energy flux densities. No gap-filling has been done and the comparison to the model output was only done at hours when no gaps occurred. A number of stations were already part of a separate flux measurement network (Aubinet et al., 2000). However, only a few stations provided data for the first operating years covering the period 1996 to 2000. Two FLUXNET stations were selected, that provided data during this period and where the model grid cell represented more than 50% of the corresponding land cover, to show energy fluxes that were representative for the particular land cover. The selected ecosystem towers cover different climatic regimes (Fig. 1): (1) Vielsalm in Belgium, a temperate climate, at an altitude of 491 m with a tower height of 40 m and covered by deciduous broadleaved forest and evergreen coniferous, and (2) Collelongo in Italy, a Mediterranean climate, at an altitude of 1645 m with a tower height of 32 m and mainly covered by deciduous broadleaved forest.

**3 Validation of the mean model state**

**3.1 Spatial distribution**

**3.1.1 Daily mean 2 m temperature**

The spatial distributions of the 10-year daily mean temperature bias (absolute, (model - observed)) of CON, DRI and FS
simulations were compared to E-OBS (Fig. 3), for the winter (DJF: December-January-February) and summer (JJA: June-July-
August) season. The average biases during winter and summer for CON, DRI and FS for the entire domain as well as for specific
subdomains are presented in Table 1. CON simulated a cold bias in general, except for northern Africa, with a pronounced
orographic effect, for both winter and summer (Fig. 3c,d). The cold bias over the entire domain was less pronounced in summer
with a value of -0.6 °C compared to the winter bias of -1.8 °C (Table 1). Moreover, the Iberian Peninsula was well simulated
during summer as compared to E-OBS, resulting in a bias of -0.5 °C. Additionally, the biases of the Mediterranean and Eastern
Europe resulted in similar small biases, due to compensating errors as can be seen from (Fig. 3d).

With respect to CON, DRI demonstrated a reduction of the cold bias during winter and summer, most prominent at the
eastern part of the domain (Fig. 3e,f). This resulted in a smaller bias for Eastern Europe of -0.3 °C and 0.0 °C for DRI relative
to CON which had a bias of -1.1 °C and -0.5 °C for winter and summer respectively (Table 1). Other subdomains showing a
large improvement of the 2 m temperature simulation by DRI, were Mid-Europe and the Alps with a winter bias of -0.7 °C and
-1.4 °C respectively that is about half of the bias of CON, and a summer bias of -0.3 °C and -0.8 °C, even more than half of
the bias of CON for these subdomains.

The performance of the FS simulation was different for winter and summer as compared to CON and DRI (Fig. 3g,h). The
simulation of the 2 m temperature during winter was best of all three approaches when using FS. Large parts of the domain
resulted in biases close to zero, such as the British Isles, France, Mid-Europe and Eastern Europe (Fig. 3g). The bias decreased
by ca. 1 °C in FS compared to CON for these subdomains (Table 1). During summer, the sign of the bias reversed from
negative to positive, except for some isolated areas (Fig. 3h). The Alps were much better presented by FS, resulting in a zero
bias as compared to CON and DRI which showed a bias of -1.8 °C and -0.8 °C (Table 1). For the Iberian Peninsula and the
Mediterranean, compensating biases resulted in positive and close to zero summer biases (Fig. 3h). Mid-Europe, France and
Eastern Europe were mainly characterised by a positive bias of around 1 °C (Table 1). The summer absolute bias simulated by
FS was very similar to CON for the Iberian Peninsula and the Mediterranean, but slightly enhanced for Eastern Europe with
ca. 0.6 °C.

In summary, CON underestimated winter and summer 2 m temperature with 1-2 °C on average. With respect to CON, DRI
and FS showed a positive effect during winter and summer. In spite of a slight enhancement by FS of the bias during summer
for Eastern Europe, the winter bias was improved for most subdomains by using FS. Overall, the use of a daily reinitialised at-
mosphere improved the representation of the 2 m temperature for both winter and summer compared to a continuous simulation
of the atmosphere.

**3.1.2 Daily accumulated precipitation**

The spatial distributions of the 10-year daily accumulated precipitation bias (relative, (model-observed)/observed) of CON, DRI and FS were compared to E-OBS, for the winter and the summer seasons (Fig. 4). The mean biases during winter and summer for CON, DRI and FS are presented for the entire domain as well as for the specific subdomains in Table 2. The precipitation pattern of E-OBS during winter displayed highest values of $> 3$ mm day$^{-1}$ over Portugal, northwestern Spain, western England, Scotland and Ireland, the Adriatic Coast and the northern flanks of the Alps (Fig. 4a,b). During summer, similar amounts of rainfall were concentrated over the Alps and the Carpathians, while lowest values of $< 1$ mm day$^{-1}$ at the Iberian Peninsula, the Mediterranean and northern Africa.

During winter, all simulations demonstrated a similar spatial variability of the wet bias, except for a dry bias in northern Africa (Fig. 4c,e,g). In general, ALARO was forced towards the too wet driving fields of ERA-Interim (Lucas-Picher et al., 2013), which can explain part of the overestimated precipitation. More particularly, the overestimation of winter precipitation was strongest in the Mediterranean and Eastern Europe with values of 46.0 % and 35.3 % respectively (Table 2). However, the bias averaged over the entire domain was larger for FS with ca. 36% compared to less than 25 % for CON and DRI. This corresponded to a higher precipitation bias of 10-20 % for all specific subdomains and even more than 50% higher for the Mediterranean.

During summer, the simulations showed different spatial variability (Fig. 4d,f,h). Regarding CON, the sumer precipitation bias was reduced over the continental part as compared to winter and positive and negative biases occurred over the southern part of the domain (Fig. 4d). The Mediterranean expressed a high wet bias of 60.5%, but the absolute values in summer were close to zero, as it is characterised by a climate with dry summers (Fig. 4b). The bias pattern over the continental part was very similar for DRI compared to CON during summer, while Southern Europe showed increased wet biases (Fig. 4f). The Iberian Peninsula, France and the Mediterranean demonstrated a bias of 30.0%, 18.3% and 84.8% respectively compared to 11.5%, 12.0% and 60.7% with CON (Table 2). The performance of FS was similar to CON for Southern and Eastern Europe (Fig. 4h). This contrasted to the continental part of the domain, where the precipitation signal reversed and dry biases occurs, though it was rather small (-7.0% for France, -13.4% for Mid-Europe, -8.2% for Eastern Europe respectively). Consequently, the summer precipitation was simulated better by FS than CON and DRI.

In summary, the model was characterised by a wet bias in winter and summer. The spatial variability during winter was very similar for all simulations, but during summer the precipitation showed a different behaviour. For the southern part of the domain, DRI established increased precipitation biases, while FS was more different to CON for the continental part, but not so much for the southern part. The use of a daily reinitialised atmosphere in DRI and FS had a neutral impact on the winter precipitation. FS improved the summer precipitation bias. Therefore, the combination of the daily reinitialised atmosphere together with a continuous surface is crucial in summer to get the best results.

**3.2 Mean annual cycle**

**3.2.1 Daily mean 2 m temperature**

[revised manuscript text omitted]

The daily maximum BR showed a strong gradient of increasing values towards the south of the domain (Fig. 6a,b,c). However, large differences appeared for the three downscaling approaches, particularly for the continental part of the domain. Relatively low values of 0 to 1 were represented by CON, while DRI showed BR values of 0.5 to 1 and highest values of 2 to

30    3 were expressed by FS. When the value is lower (higher) than 1, the latent heat flux is higher (lower) than the sensible heat flux. The FLUXNET observations for Vielsalm and Collelongo were displayed, and indicated best agreement with DRI (Fig. 6b), expressed by values of 1.12 and 1.32 respectively (Table 3). Though this validation was based on 5 summer periods only from 1996 to 2000, it was still robust as indicated by the corresponding plot for the 10-year summer period from 1991 to 2000

(Fig. 6d,e,f, Table 3). In spite of highest BR values presented by FS, the stations of Vielsalm and Collelongo were located into isolated parts of lower BR, indicated by the average values of 0.61 and 0.83 respectively (Table 3).

The net radiation was underestimated for all simulations (Table 3), but this underestimation was larger for Collelongo, which could be related to its complex topography. The model generally underestimated H, and overestimated LE. The ground heat flux (G) showed much higher values than the observed ones. G is dependent on the soil temperature, which is overestimated by the land surface model. This is due to the representation of the soil-surface leaf litter in the model. Wilson et al. (2012) showed that without an explicit formulation of water and energy exchanges within the residue layer, their surface model overestimated LE, G and soil temperature and underestimated H. As the net radiation and ground heat flux were simulated very similarly for all simulations, they were not shown in Fig. 7.

For Vielsalm, H was simulated well by DRI and FS during nighttime and daytime, whereas CON underestimated H during daytime (Fig. 7a). The daily maximum H by CON was only 118 $W^{-2}$, as compared to 151 and 139 $W^{-2}$ for DRI and FS respectively (Table 3). Yet again, this validation was only done for 5 summer periods from 1996 to 2000, but the corresponding daily maximum values for the 10-year summer period 1991-2000 indicated that the 5-year period was representative for the validation of the fluxes (Table 3). The LE was overestimated by all simulations, but the difference with the observations was smallest for DRI, while it was highest for CON. The daily maximum BR was lower than 1 for all downscaling approaches (Table 3). This means that they all simulated a higher latent than sensible heat flux. Still, DRI and FS showed higher values for BR than CON. Therefore, the partitioning of the surface energy fluxes was better represented by DRI and FS for the station of Vielsalm.

For Collelongo, H was underestimated by the model during daytime and overestimated during nighttime, except for DRI which demonstrated a good agreement with the observations. Yet again, the model overestimated LE during daytime, except for DRI. The daily maximum H for DRI of 247 $W^{-2}$ was close to the observed value of 253 $W^{-2}$, whereas CON and FS simulated much lower values of 159 $W^{-2}$ and 197 $W^{-2}$ respectively (Table 3). The simulated LE showed the largest difference with the observed one using CON. Regarding BR, the simulation by DRI with a value of 1.35 was in very good agreement with the observations. The DRI simulation resulted in the correct partitioning of the surface energy fluxes at Collelongo. CON was not performing well in simulating the correct partitioning, while FS had already much improved as compared to CON.

In summary, RN was underestimated by the model, whereas H was underestimated and LE was overestimated. However, DRI performed well for H at Vielsalm and for LE at Collelongo. For Colellongo, this resulted in a correct simulation of the partitioning of the surface energy fluxes, translated into an excellent value for BR. Least well simulated were CON and G. The use of a daily reinitialised atmosphere improved the correct partitioning of the surface energy fluxes. FS could not improve the representation of the surface energy fluxes for both stations with respect to DRI. The validation of G was not conclusive, as this parameter needs to be revised with an improved residue layer.

**5 Conclusions**

An assessment of three downscaling approaches has been performed using the regional climate model ALARO-0 coupled to the land surface model SURFEXv5, with lateral and initial boundary conditions from ERA-Interim. The simulations were applied for a 10-year period from 1991 to 2000, for a Western European domain. The performance of ALARO-0 with SURFEX has already been validated for NWP applications (Hamdi et al., 2014), but not yet for long-term climate simulations.

We compared the common used approach of a continuous climate simulation with two alternative aprooaches of frequently reinitialising the RCM simulation towards its driving field, combined with either a daily reinitialised or continuous surface. The use of a daily reinitialised atmosphere outperformed the continuous approach for winter and summer 2 m temperature, and detoriorated the summer precipitation. However, the use of a continuous surface next to a daily reinitialised atmosphere improved the summer precipitation with respect to the continuous approach. Furthermore, it improved the winter 2 m temperature, whereas it resulted in a neutral impact on the summer 2 m temperature and the winter precipitation, despite a slight deterioration at the Mediterranean. The SSTs were reinitialised daily together with the atmosphere, as compared to the monthly updated SSTs in the continuous approach.

The seasonal cycle of the 2 m temperature and precipitation was different for three selected subdomains that covered large climate variability. Both the temperature climate of Mid-Europe and the continental climate of Eastern Europe indicated more seasonal variability than the Mediterranean climate of the Iberian Peninsula. The simulation of the 2 m temperature had improved when applying daily reinitialised atmosphere with continuous surface, despite an overestimation of the summer 2 m temperature. The model disagreed more for precipitation, because of the forcing towards the too wet driving field of ERA-Interim and the low spatial coverage by the observations in some regions. It was clear that the agreement for the precipitation between the model and the observations was highest during summer, while other seasons showed stronger deviations.

During summer, the interaction between the land surface and the overlaying atmosphere is largest. The 2 m temperature interacts with the soil moisture and influences the partitioning of the surface energy fluxes. The daily reinitialisation of the atmosphere improved the representation of a correct partitioning, though the latent heat was highly overestimated for Vielsalm and resulted in a too low value as compared to the FLUXNET observations. Still, this approach outperformed the use of a continuous simulation. For a more comprehensive analysis, we recommend to include more FLUXNET stations. A more in-depth analysis on the interaction between 2 m temperature, precipitation, and surface energy fluxes can reveal soil-moisture-temperature coupling (Jaeger et al., 2009), but this lies outside the scope of this study.

In conclusion, this study demonstrated that the approach of a daily reinitialised atmosphere was superior over the continuous approach. The use of a continuous surface next to a daily reinitialised atmosphere even improved the winter temperature and summer precipitation. The latter approach is highly recommended in a setup with GCM forcing, as imperfect initial and lateral boundary conditions are applied.

[revised manuscript text omitted]
 1991-2000. The dots represent the grid points with a significant difference at 5%, using the Student's t-test with a null hypothesis stating that the means of the model and observations are equal.

[Figure]

**Figure 4.** Daily accumulated precipitation (mm day$^{-1}$) for E-OBS DJF (a) and JJA (b), and relative bias (%) of the model with E-OBS for CON DJF (c) and JJA (d), for DRI DJF (e) and JJA (f) and for FS DJF (g) and JJA (h), all at a 20 km horizontal resolution for a 10-year period 1991-2000. The dots represent the grid points with significant different variations at 5%, using the F-test with a null hypothesis stating that the variances of the model and observations are equal.

[Figure]

**Figure 5.** Mean annual cycle of the daily 2 m temperature (°C) with E-OBS, CON and FS for (a) the Iberian Peninsula, (b) Mid-Europe, and (c) Eastern Europe, and daily accumulated precipitation (mm day$^{-1}$) for (d) the Iberian Peninsula, (e) Mid-Europe, and (f) Eastern Europe, averaged over the 10-year period 1991-2000. Both the mean and standard deviation (SD) are displayed as text.

[Figure]

**Figure 6.** Daily maximum Bowen ratio averaged over the 5 year JJA period 1996-2000 for (a) CON, (c) DRI and (e) FS and averaged over the 10-year JJA period 1991-2000 for (b) CON, (d) DRI and (f) FS. The dots represent the values for the FLUXNET stations Vielsalm (Belgium) and Collelongo (Italy).

[Figure]

**Figure 7.** Daily cycle of the energy fluxes (W m$^{-2}$) in JJA 1996-2000 for Vielsalm in the top row and Collelongo in bottom row for (a,c) H, and (b,d) LE, for the FLUXNET observations and their corresponding model grid points by CON, DRI and FS. The error bars represent the standard deviation of the observations.

---

## Author Comment (AC2) · 4 Nov 2016

[1,2]JulieBerckmans [1,2]OlivierGiot [1,3]RozemienDe Troch [1,3]RafiqHamdi [2]Reinhart Ceulemans [1,3]PietTermonia

[1]Royal Meteorological Institute, Brussels, Belgium [2]Centre of Excellence PLECO (Plant and Vegetation Ecology), Department of Biology, University of Antwerp, Antwerp, Belgium [3]Department of Physics and Astronomy, Ghent University, Ghent, Belgium

Julie Berckmans (julie.berckmans@meteo.be)

11

[Figure]

November 4, 2016

**1   Reply to the Editor**

Dear Editor,

we have prepared a majorily improved version of the manuscript by incorporating all suggestions and critical comments raised by both reviewers. Please find attached our response to the referees' comments on our above mentioned manuscript, titled "Reinitialised versus continuous regional climate simulations using ALARO-0 coupled to the land surface model SURFEX". Below mentioned you will find our detailed responses to all the reviewers' comments and suggestions (put in italics and red). We have also explained where and how they were incorporated in the revised manuscipt.

**2   Reply to Anonymous Referee #2**

We would like to thank the Anonymous Referee #2 for the encouraging and constructive comments, which have improved the manuscript. Below is a list of modifications that we have implemented based on your comments.

[Figure]

*The manuscript has two questions in mind: (1) Is the land surface model SURFEX better than ISBA in climate application with the modelling system ALADIN in ALARO-0 set-up? (2) Is it better to do regional climate simulations or dynamical downscaling (i.e. with re-initialization of the regional atmosphere and land surface) or some mix (e.g., continuous land surface simulation with atmosphere and sea surface temperature re-initialization)? In my opinion these questions are only weakly linked. Of course, if you want to apply continuous land surface simulation there is need to use the best available land surface model (a bad land surface model might render a regional climate simulation useless), but a better land surface model is better even in an NWP context. Therefore, I suggest to focus on one of the questions and I find the second question more interesting.*

Thank you for this suggestion. Even though the comparison between SURFEX and ISBA is valid, it is only weakly linked to the question of the performance of ALARO to SURFEX in multiple downscaling approaches. The above mentioned comparison has been done in an NWP context, and the comparison for long-term climate simulations builds upon these results. However, we do not elaborate on this in the revised manuscript. We focus on the second question only, as this was the main focus of the manuscript from the onset and it is indeed more interesting.

*The manuscript claims that SURFEX is better than ISBA. It has been shown in NWP context (given reference Hamdi et al., 2014), but the authors target climate time scales. They compare an available CORDEX simulation with ISBA against a new climate simulation with SURFEX which was done using a smaller simulation domain. Different domain sizes limit the comparability crucially. Both simulations were driven by the re-analysis ERA-Interim. Therefore, I would expect limited-area simulations are potentially better with a smaller domain. The presented results are not conclusive.*

The reviewer's assumption is true. The domain sizes should be equal to compare well the sensitivity of the regional climate model to the different land surface model. In

the future we plan a separate study for the comparison with ISBA, but with the same model domain to make it consistent. This is not an objective any longer in the revised manuscript.

*The authors cite many re-initialization vs regional climate simulation experiments. Even with ALADIN such a re-initialization experiment has been published, but for a 3-monthly period only (Beck et al. 2004). I suggest to make the manuscript more interesting and publishable by doing the list of re-initialization experiments more exhaustive by adding (a) full re-initialization (i.e. with SURFEX re initialization) and (b) blending (i.e. re-initialization of the large atmospheric scales doing "climate" for the smaller scales, see Beck et al. 2004). Finally, perhaps too much for one paper, I think the ultimate criterium will be how the set-up performs with GCM forcing.*

Thank you for this valuable suggestion. We have included the full re-initialisation experiment in the revised manuscript. The new objective of the manuscript is as follows (Page 3 Lines 3-6):
"The objective of this study was to evaluate the simulation potential of three regional climate downscaling approaches with different update frequencies of the initial conditions: (1) a continuous simulation of both the atmosphere and the surface; (2) a simulation with daily reinitialisations for both the atmosphere and the surface; and (3) a simulation with daily reinitialisations of the the atmosphere while one single initialisation of the surface."

The paper of Beck et al. (2004) describes different nesting methods and concludes that for dynamical downscaling the direct nesting method is acceptable. We have used this method in the experiment, but have not explicitly mentioned it. We decided not to include a blending experiment in the revised manuscript, as the focus is on the sensitivity to the update frequency of the initial conditions. The set-up with forcing from GCM has been done as well, using ARPEGE CMIP5. We will use these results

for a separate study in the future, as the boundary conditions differ a lot and lead to different conclusions. In the conclusions, we give the recommendation of investigating this in a GCM context.

The new version (Page 12 Lines 28-31) reads:

"In conclusion, this study demonstrated that the approach of a daily reinitialised atmosphere was superior over the continuous approach. The use of a continuous surface next to a daily reinitialised atmosphere even improved the winter temperature and summer precipitation. The latter approach is highly recommended in a setup with GCM forcing, as imperfect initial and lateral boundary conditions are applied."

**3   Manuscript version with highlighted changes is supplemented.**

**Supplement:**

**Reinitialised versus continuous regional climate simulations using ALARO-0 coupled to the land surface model  SURFEXv5**

Julie Berckmans[1,2], Olivier Giot[1,2], Rozemien De Troch[1,3], Rafiq Hamdi[1,3], Reinhart Ceulemans[2], and Piet Termonia[1,3]

[1]Royal Meteorological Institute, Brussels, Belgium
[2]Centre of Excellence PLECO (Plant and Vegetation Ecology), Department of Biology, University of Antwerp, Antwerp, Belgium
[3]Department of Physics and Astronomy, Ghent University, Ghent, Belgium

*Correspondence to:* Julie Berckmans (julie.berckmans@meteo.be)

**Abstract.**  For the simulation of the regional climate with limited area models, the common method for dynamical downscaling is the continuous approach with initial and lateral boundary conditions from the reanalysis or the global climate model. The simulation potential can be improved by applying an alternative approach of reinitialising the atmosphere, combined with either a daily reinitialised or a continuous surface. We evaluated the  dependence of the simulation potential on the running mode of the regional climate model ALARO coupled to the land surface model SURFEX, and driven by the European Centre for Medium-Range Weather Forecasts (ECMWF) Interim Re-Analysis (ERA-Interim) data. Three types of downscaling simulations were carried out for a 10-year period covering 1991 to 2000, over a Western European domain at 20 km horizontal resolution: (1) a continuous simulation of both the atmosphere and the surface; (2) a simulation with daily reinitialisations for both the atmosphere and the surface; and (3) a simulation with daily reinitialisations of the atmosphere while the surface is kept continuous  mode. The results  showed that the daily reinitialisation of the atmosphere improved the simulation of the 2 m temperature for all seasons. It revealed a neutral impact on the  atmosphere, daily precipitation totals during winter, but the results were improved for the summer when the surface was kept continuous. The behaviour of the three model simulations varied among different climatic regimes. Their seasonal cycle  for the 2 m temperature and daily precipitation totals was very similar for a Mediterranean climate, but more variable for temperate and continental climate regimes. Commonly, the summer climate is characterised by strong interactions between the atmosphere and the  surface. The results for summer demonstrated that the use of a  daily reinitialised atmosphere improved the representation of the partitioning of the surface energy fluxes.

Therefore, we recommend to use the alternative approach of the daily reinitialisation of the atmosphere for the simulation of the regional climate.

**1   Introduction**

5   The first long-range simulation of the general circulation of the atmosphere dates back to 1956 (**?**). Today it is still the primary tool for climate projections. However, due to limiting computer resources, the current horizontal resolution of 100-200 km is still coarse. A higher resolution and more spatial details can be obtained by nesting a regional climate model (RCM), over a smaller domain, into a coarse-resolution global climate model (GCM). This is also referred to as dynamical downscaling. The  GCM or global reanalysis provides the large-scale

10    meteorological and surface fields to the RCM as initial and lateral boundary conditions. The global features are  translated into regional and local conditions over the region of interest (**?**). Hence, RCMs allow to run climate simulations over a smaller domain with higher horizontal resolution and with an affordable computing cost.

Since the late 60's, the  numerical weather prediction (NWP) community uses  high-resolution limited area models. The numerical approach was first

15   used for a regional climate simulation by **?**. Their climate simulation used the NWP model in forecasting mode with  short-term reinitialisations of the initial conditions. To be able to run  them without these short-term reinitialisations, the regional climate community applied monthly to multidecadal simulations, with only one single initialisation of the large-scale fields and frequent updates of the lateral boundary conditions (**?**). These so-called long-term  continuous simulations required

20   improvements in the representation of physical processes in  the RCMs. This continuous simulation is still the most common in the RCM community (**?**). Nonetheless, the simulated large-scale  fields deviate from the driving  lateral boundary conditions, by applying the continuous  approach (**?**).

The accuracy of the dynamical downscaling has improved by using short-term reinitialisations  (**????**). All these authors showed the advantage of using short-term reinitialisations by  reducing systematic errors. However, only few

25   authors adopted this method, mainly because of its higher computational costs.

 Most studies (**???**) dealing with the evaluation of reinitialised versus continuous climate simulations, covered only short time periods. The 24-hourly reinitialised simulation of the precipitation, in particular of the precipitation pattern, improved as compared to the continuous simulation (**?**).  This last mentioned analysis covered only a short time period, one month in 2002 during a large flooding event in the Elbe river catchment. Changing the period of reinitialisation,

30   from monthly to 10-daily,  a reduction in systematic errors has been shown for precipitation when using the 10-day reinitialisation (**?**). Even in a 20-year RCM simulation forced by reanalysis data, the sequence of events was better preserved by using  short-term reinitialisations (**?**).

~~for its operational numerical weather forecasts. ALARO-0 has already proven its ability for regional climate modelling with daily reinitialisations (??). The model initially used the Interaction Soil-Biosphere-Atmosphere Interaction (ISBA) land surface scheme (??). The setup of ALARO-0 with ISBA has been validated for continuous climate simulations and is now contributing to the EURO-CORDEX project (??). Meanwhile the more recent land surface scheme of Météo-France SURFace EXternalisiée (SURFEX, ?) has been implemented in the ALARO-0 version. With respect to NWP applications,the introduction of SURFEX within ALARO-0 has shown neutral to positive effects on the 2 m temperature, 2 m relative humidity, 10 m wind speed and on the precipitation scores compared to the previously used ISBA scheme (?). Therefore, the evaluation of SURFEX within ALARO-0 is highly demanding for regional climate simulations.~~

 atmosphere have only limited impact on the simulation potential (?). In contrast to the atmosphere, the surface takes a longer time to reach dynamical equilibrium with the overlaying atmosphere, in  order of a few weeks to several seasons, depending on the depth of  soil layer.

The surface interacts with the climate through the soil moisture and soil temperature, by influencing the surface energy budget (?). The soil moisture controls the partitioning of the incoming energy into a latent and sensible heat flux. The  soil moisture limitation on the evapotranspiration is largest during the summer (?). The availability of soil moisture for evapotranspiration is determined by the 2 m temperature (?). As the surface-atmosphere interactions play a crucial role in the representation of the current and future climate, it is important to validate the model with ground observations. The FLUXNET database provides data on the surface energy fluxes, based on eddy covariance measurements (?).

The objective of this study was to evaluate the  simulation potential of three regional climate downscaling approaches with different update frequencies of the initial conditions: (1) a continuous simulation of both the atmosphere and the surface; (2) a simulation with daily reinitialisations for both the atmosphere and the surface; and (3) a simulation with daily reinitialisations of the atmosphere while the surface is kept continuous. We used the ALARO model to dynamically downscale the European Centre for Medium-Range Weather Forecasts (ECMWF) Interim Re-Analysis (ERA-Interim, ?). Within this study, ALARO was coupled to  land surface ~~is not recommended since this setup limits the equilibrium of the surface physics (soil moisture and temperature) , which is particularly desirable in long-term climate modelling (?). The third objective is to examine the performance of the two downscaling approaches with respect to the land surface feedback, more specifically the soil moisture feedback. It was hypothesized that the differences in temperature and~~

 model of Météo-France SURFace Externalisée (SURFEX, ?). We evaluated the mean 2 m temperature and mean daily total precipitation by comparing with the 0.22° ECA&D E-OBS dataset (?), and the surface energy fluxes by comparing with the FLUXNET database (?). The analysis covered a 10-year period from 1991 to 2000, for a domain encompassing Western Europe.

[revised manuscript text omitted]

15  **?** relaxation zone consisted of eight grid points irrespective of the resolution.

**2.3**

 The zonal and meridional wind components, atmospheric temperature, specific humidity, surface pressure and surface components were provided every 6 hrs as lateral boundary conditions and interpolated hourly. They were introduced

20 as initial conditions accross the domain. A spin-up time was considered for the model to reach equilibrium between the lateral boundary conditions and the internal model physics (**?**). For the sake of a good understanding, the following description makes a distinction between atmospheric spin-up time, typically of a  few days, and surface spin-up time, typically of a few months to one year. The analysis covered a 10-year period from 00UTC on 01 January

25  1991 to 00UTC on  01 January  2001. Although the 10-year length is arbitrary, it is sufficiently long to include some inter-annual variability and to generate a reasonable sample of extreme events. The use of a NWP model in a

30 long-term climate setting for the performance of extreme precipitation events for a 10-year period was recently demonstrated

 (**?**). To evaluate the sensitivity of the model to the update frequency of the initial conditions, three types of downscaling approaches were conducted with ALARO-0 coupled to  SURFEXv5.

The first downscaling approach was done by simulating the model in a continuous mode for both the atmosphere and the surface (hereafter called  CON ("CONtinuous"), Fig. 2). The model was simulated from 00UTC on 01  January 1990, and ran continuously until 00UTC on 01 January 2001. The first year was treated as both atmospheric and surface spin-up time, and was excluded from the analysis. The simulations were interrupted and restarted monthly to allow for SSTs to be updated. Other surface parameters that were updated monthly using the climatological values from ECOCLIMAP were the vegetation fraction, surface roughness length, surface emissivity, surface albedo, sand and clay fractions.

In the second downscaling approach, the model was reinitialised daily for both the atmosphere as the surface (herafter called DRI ("Daily ReInitialisation"), Fig. 2). The model started at 12UTC on 01 January 1991 , and each reinitialisation ran for 60 hrs . The first 36  hours were treated as atmospheric spin-up time, and were excluded from the analysis. By applying this downscaling approach, the regional model stays close to the driving fields (**?**). As the driving fields provided daily reanalysed data, a spin-up for the surface was redundant.

 The third downscaling approach tries to find the best compromise between previous approaches. The atmosphere was reinitalised daily and the surface was simulated continuously with one single initialisation (hereafter called FS ("Free Surface"), Fig. 2). This allowed the model to simulate the atmospheric fields close to the driving fields, together with a surface in equilibrium state. The model was simulated from 12UTC on 01  March 1990 until 31  May 1991, and the atmosphere was reinitialised daily for a simulation time of 60 hrs. The first 36 hrs were treated as atmospheric spin-up time, and were excluded from the analysis. The surface conditions were kept continuous and joined after the atmospheric spin-up time with the surface conditions of the previous daily simulation. In contrast to the atmospheric spin-up time, the surface spin-up lasted from 01 March 1990 until 31 May 1990, and this 3-monthly period was excluded from the analysis. Although CON required one year spin-up time, 3 months were sufficient for the FS deep soil moisture to reach equilibrium state, when starting in March (not shown). The simulations were done in parallel for each year from 1990 to 2000

the results of both downscaling strategies. This comparison allowed to examine the effect of SURFEX on the climate mode, as was previously done for NWP (?). , and the 3 monthly spin-up time was replaced by the analysis of the previous year.

The model output at every 3 hrs was used for the model evaluation. The evaluation of the atmospheric variables atmospheric variables for winter and summer was done for seven subdomains across Europe, to cover the spatial variability of the domain (Fig. ??1). This was in agreement with the subdomains that were used in the EURO-CORDEX community (?) and that were defined earlier in the framework of the PRUDENCE project project "Prediction of Regional scenarios and Uncertainties for Defining European Climate change risks and Effects" (PRUDENCE) (?). The subdomains used in this study are were the British Isles (BI), the Iberian Peninsula (IP), Mid-Europe (ME), France (FR), the Alps (AL), the Mediterranean (MD) and Eastern Europe (EA). For both MD the subdomains IP, ME, and EA, only part of their domains were used, as our domain is not covering the total subdomain, and also the relaxation zone was excluded. the yearly cycle of the atmospheric variables was evaluated. These selected subdomains covered a range of climatic regimes. Additionally, the surface energy fluxes were analysed. As land-surface processes play an important role primarily during summer, the model output was stored at every hour for the summer period of June-July-August (JJA) during the 10-year period. We evaluated the partitioning of the sensible and latent heat fluxes by the daily maximum Bowen Ratio (BR, ?) for the summer periods from 1996 to 2000 for the total study domain, and compared the selected FLUXNET stations with their corresponding model grid points. The corresponding daily maximum BRs were analysed for the 10-year summer period from 1991 to 2000. When the value is lower (higher) than 1, the latent heat flux is higher (lower) than the sensible heat flux. The diurnal cycles of all surface energy fluxes were analysed and validated against observations.

**3  Observational data**

**2.1  E-OBS gridded dataset Observational reference data**

The results of the climate simulations were validated against E-OBS, a daily high-resolution gridded observational dataset (?). The dataset consists of the daily mean temperature, the daily maximum and minimum temperature, and the daily precipitation sum total. The most recent version v12.0 was selected on the 0.22° rotated pole grid, corresponding to a 25 km horizontal resolution in Europe. It covers the period 01 January 1950 to 30 June 2015. With respect to previous versions of E-OBS, some improvements include the new precipitation data series for countries southeast of the Baltic Sea, updated Slovakian series for all variables, updated Croatian series for all variables and a highly extended network for Catalonia, Spain. These improvements also concerned our area of interest and time period of interest. In order to validate the model data, the E-OBS data ALARO-0 data at 20 km horizontal resolution were bilinearly interpolated towards the ALARO-0 20 km E-OBS at 25 km horizontal resolution and replotted to our study domain. A careful interpretation of E-OBS was necessary, as this regridded non-homogeneously distributed network implied applied a smoothing out of extreme precipitation and consequently a large underestimation of the mean precipitation (?).

**2.2**

 For the validation of the surface fluxes distribution in the model, we used measurements from the FLUXNET Level 3 flux tower database (**?**). It provides information on the energy  exchange between the ecosystem and the atmosphere. FLUXNET is a global network, and consists of flux towers using the eddy covariance method to monitor carbon dioxide and water vapor exchange rates, and energy flux densities.

5 No gap-filling has been done and the comparison to the model output was only done at hours when no gaps occurred. A number of stations were already part of a separate flux measurement network (**?**). However, only a few stations provided data for the first operating years

10  covering the period 1996 to 2000. Two FLUXNET stations were selected, that provided data during this period and where the model grid cell represented more than 50% of the corresponding land cover, to show energy fluxes that were representative for the particular land cover. The selected ecosystem towers cover different climatic regimes (Fig. 1): (1) Vielsalm in Belgium, a temperate climate, at an altitude of 491 m with a tower height of 40 m and  covered by deciduous broadleaved forest and evergreen coniferous, and (2) Collelongo in Italy, a Mediterranean

15 climate, at an altitude of 1645 m with a tower height of 32 m and mainly covered by deciduous broadleaved forest.

**3 Validation of the mean model state**

**3.1 Spatial distribution**

**3.1.1 Daily mean 2 m temperature**

The spatial  distributions of the 10-year daily mean temperature bias  (absolute, (model - observed))

20 of CON, DRI and FS simulations  were compared to E-OBS (Fig. 3), for the winter (DJF: December-January-February) and summer (JJA: June-July-August) season. The average biases during winter and summer for CON, DRI and FS for the entire domain as well as for specific subdomains are presented in Table 5. CON simulated a cold bias in general, except for northern Africa, with a pronounced orographic effect, for both winter and summer (Fig. 3c,d).

25  The cold bias over the  entire domain was less pronounced in summer  with a value of -0.6  C compared to the winter bias of C (Table 5). Moreover, the Iberian Peninsula  was well simulated during summer as compared to

E-OBS, resulting in a bias of -0.5 °C. Additionally, the biases of the Mediterranean and Eastern Europe resulted in similar small biases, due to compensating errors as can be seen from (Fig. 3d).

With respect to CON, DRI demonstrated a reduction of the cold bias during winter and summer, most prominent at the eastern part of the domain (Fig. 3e,f). This resulted in a smaller bias for Eastern Europe of -0.3 °C and 0.0 °C for DRI relative to CON which had a bias of -1.1 °C and -0.5 °C for winter and summer respectively (Table 5). Other subdomains showing a large improvement of the 2 m temperature simulation by DRI, were Mid-Europe and the Alps with a winter bias of -0.7 °C and -1.4 °C respectively that is about half of the bias of CON, and a summer bias of -0.3 °C and -0.8 °C, even more than half of the bias of CON for these subdomains.

The performance of the FS simulation was different for winter and summer as compared to CON and DRI (Fig. 3g,h). The simulation of the 2 m temperature during winter was best of all three approaches when using FS. Large parts of the domain resulted in biases close to zero, such as the British Isles, France, Mid-Europe and Eastern Europe (Fig. 3g). The bias decreased by ca. 1 °C in FS compared to CON for these subdomains (Table 5). During summer, the sign of the bias reversed from negative to positive, except for some isolated areas (Fig. 3h). The Alps were much better presented by FS, resulting in a zero bias as compared to CON and DRI which showed a bias of -1.8 °C and -0.8 °C (Table 5). For the Iberian Peninsula and the Mediterranean, compensating biases resulted in positive and close to zero summer biases (Fig. 3h). Mid-Europe, France and Eastern Europe were mainly characterised by a positive bias of around 1 °C (Table 5). The summer absolute bias simulated by FS was very similar to CON for the Iberian Peninsula and the Mediterranean, but slightly enhanced for Eastern Europe with ca. 0.6 °C.

In summary, CON underestimated winter and summer 2 m temperature with 1-2 °C on average. With respect to CON, DRI and FS showed a positive effect during winter and summer. In spite of a slight enhancement by FS of the bias during summer for Eastern Europe, the winter bias was improved for most subdomains by using FS. Overall, the use of a daily reinitialised atmosphere improved the representation of the 2 m temperature for both winter and summer compared to a continuous simulation of the atmosphere.

**3.1.2 Daily accumulated precipitation**

The spatial  distributions of the 10-year daily accumulated precipitation bias  (relative, (model-observed)/observed) of CON, DRI and FS were compared to E-OBS, for the winter and the summer seasons (Fig. 4). The mean biases during winter and summer for CON, DRI and FS are presented for the entire domain as well as for the specific subdomains in Table 2. The precipitation pattern of E-OBS during winter  displayed highest values of > 3 mm day$^{-1}$ over Portugal, northwestern Spain, western England, Scotland and Ireland, the Adriatic Coast and the northern flanks of the Alps (Fig. 4a,b). During summer, similar amounts of rainfall  were concentrated over the Alps and the Carpathians, while lowest values of < 1 mm day$^{-1}$ at the Iberian Peninsula, the Mediterranean and northern Africa.

During winter, all simulations demonstrated a similar spatial variability of the wet bias, except for a dry bias in northern Africa (Fig.  4c,e,g). In general, ALARO was forced towards the too wet driving fields of ERA-Interim (?), which can explain part of the overestimated precipitation. More particularly, the overestimation  of winter precipitation was strongest in the Mediterranean and Eastern Europe ~~(46.74 % and 54.07 % ) and during summer for the Iberian Peninsula (49.35 %). In addition, excessive amounts of rainfall appear near the Adriatic coast, southern Italy, and southern France (Fig. ??c,d). Similarly to CRDX, CON overestimates winter precipitation, except for northern Africa (Fig. ??e). The largest wet bias in winter is present in the Mediterranean, while the bias in Eastern Europe is reduced with ≈ 37CRDX (Table 3). The subdomains France and Mid-Europe show 10-20 % less overestimation of the precipitation as compared with CRDX (Table 3).~~ less than 25 % for CON and DRI. This corresponded to a higher precipitation bias of 10-20 % for all specific subdomains and even more than 50% higher for the Mediterranean.

During summer, the  simulations showed different spatial variability (Fig.  4d,f,h). Regarding CON, the sumer precipitation bias was reduced over the continental part as compared to winter and positive and negative biases occurred over the southern part of the domain (Fig. 4d). The Mediterranean expressed a high wet bias of 60.5%,  but the  absolute values  in summer were close to zero, as it is characterised by a climate with dry summers (Fig. 4b).

 The bias pattern over the continental part was very similar for DRI compared to CON during summer, while Southern Europe showed increased wet biases (Fig. 4f). The ~~wet bias during winter is slightly enhanced for all subdomains (Table 3). However, the values are in the same order of magnitude for the continental subdomains. The deterioration is largest for the British Isles, the Iberian Peninsulaand the Mediterranenan. Their wet bias degrades with over 100 %, corresponding to an absolute increase of 1.21 mm day$^{-1}$. These subdomains are exactly under the influence of the Atlantic OceanSea. As winter is mostly characterised by transient stratiform precipitation systems, the problem of the doubled precipitation bias for these~~

subdomains arrives mainly from SSTs that are reinitialised daily instead of simulated continuously. During summer however, the precipitation biases for the British Isles, the Iberian Peninsula, the Alps and the Mediterranean are very similar for FS

35  demonstrated a bias of 30.0%, 18.3% and 84.8% respectively compared to 11.5%, 12.0% and 60.7% with CON (Table  2). The performance of FS was similar to CON for Southern and Eastern Europe (Fig. 4h). This contrasted to the continental part of the domain, where the precipitation signal  reversed and dry biases occurs, though it  was rather small ( -7.0% for France,  -13.4% for Mid-Europe, -8.2% for Eastern Europe respectively).

5  Consequently, the summer precipitation was simulated better by FS than CON and DRI.

In summary, the  model was characterised by a wet bias in winter and summer.  The spatial variability during winter was very similar for all simulations, but during summer the precipitation showed a different behaviour. For the southern part of the domain, DRI

10 established increased precipitation biases, while FS was more different to CON for the continental part, but not so much for the southern part.  The use of  a daily reinitialised atmosphere in DRI and FS had a neutral impact on the winter precipitation  FS improved the summer

15 precipitation bias. Therefore, the combination of the daily reinitialised atmosphere together with a continuous surface is crucial in summer to get the best results.

**3.2 Mean annual cycle**

**3.2.1 Daily mean 2 m temperature**

To validate specific subdomains within the larger domain on a monthly scale, the mean annual cycles of the downscaled

20 simulations  were compared to the observations (Fig. 5). We focused on the following subdomains (Fig. 1): (1) the Iberian Peninsula at the western boundary of the domain with its warm and dry summer climate (2) Mid-Europe with its  temperate climate; and (3) Eastern Europe at the eastern boundary of the domain with its continental climate.

The daily mean 2 m temperature  reached about 23 °C  in the Iberian Peninsula, while it  raised to 20 °C

25 in Mid-Europe and Eastern Europe (Fig. 5a,b,c). For  these selected subdomains,  all downscaled simulations presented very similar autumn (SON: September-October-November) temperatures, but  underestimated them with respect to E-OBS.  Regarding the other seasons, the simulations revealed a different behaviour in the representation of the 2  m temperature with respect to the

30 observations.

For the Iberian Peninsula, the 2 m temperature was generally underestimated for all seasons (Fig. 5a). Except for autumn, FS was closer to the observations as compared to CON and DRI, resulting in a yearly mean temperature of 12.5 °C, which was closer to the observed yearly mean temperature of 13.7 °C as compared to 11.6 °C and 11.9 °C by CON and DRI respectively. The summer 2 m temperature was well simulated by FS for this subdomain. For Mid-Europe, CON and DRI underestimated the 2 m temperature for all seasons, whereas FS was very close to the observations from February to May (Fig. 5b). However, FS overestimated the summer 2 m temperature and CON and DRI underestimated the summer 2 m temperature. Still, the yearly mean value of 9.0 °C by FS was very close to the observational mean of 9.3 °C. In contrast to the Iberian Peninsula and Mid-Europe, DRI and FS demonstrated almost identical behaviour for the simulation of the 2 m temperature for Eastern Europe during winter and spring (MAM: March-April-May) (Fig. 5c). Their simulation was very close to the observations, whereas CON underestimated the 2 m temperature. Similar to Mid-Europe, FS slightly overestimated the summer 2 m temperature with ca. 1 °C and CON underestimated the summer 2 m temperature with ca. 1 °C in Eastern Europe. Yet again, the yearly mean value of 8.5 °C by FS was very similar as compared to the observations with a value of 8.6 °C, while largest differences occurred using CON with a value of 7.5 °C.

In summary, the yearly mean temperature was underestimated by CON for all subdomains. Along the selected subdomains, there were larger differences between the simulations in Mid-Europe and Eastern Europe as compared to the Iberian Peninsula. DRI was able to simulate the 2 m temperature better for Mid-Europe and Eastern Europe as compared to CON for winter, spring, and summer. The yearly mean 2 m temperature was best represented by FS. However, the summer 2 m temperature was overestimated by FS for Mid-Europe and Eastern Europe, but neither CON nor DRI simulated well the summer 2 m temperature with respect to the observations.

**3.2.2 Daily accumulated precipitation**

Similar to temperature, the monthly means of the daily accumulated precipitation, averaged over the 10-year period, are shown in Fig. 5 for the Iberian Peninsula, Mid-Europe and Eastern Europe. When comparing the observations, the yearly cycle was most pronounced at the Iberian Peninsula, with minimum precipitation values of ca. 0.5 mm day$^{-1}$ during summer, and maximum precipitation values of ca. 3 mm day$^{-1}$ during spring, autumn and beginning of the winter (Fig. 5d). The precipitation in Mid-Europe reached highest values of ca. 3 mm day$^{-1}$ during summer (Fig. 5e). The continental climate of Eastern Europe presented average values of 1 mm day$^{-1}$ for winter and spring, while most rainfall occurred in the summer of ca. 2.5 mm day$^{-1}$ (Fig. 5f).

In general, the agreement of the simulations was largest in autumn. For the Iberian Peninsula, the seasonal pattern of the downscaled simulations followed the seasonal pattern of E-OBS (Fig. 5d). The model simulations represented an overestimation of the precipitation for all seasons. This overestimation was stronger in winter and in spring,

and is in agreement with **?**. For these two seasons, E-OBS  showed an undercatch of the precipitation, which might  have amplified the model biases (**?**). CON and DRI were closer to the observations than FS in winter and spring, resulting in  yearly mean values of 1.9, 2.0, and 2.1 mm day$^{-1}$  respectively for CON, DRI and FS, as compared to the observational mean value of 1.7 mm day$^{-1}$ . In Mid-Europe, the model  overestimated the precipitation for most of the year, except for summer (Fig. 5e). During summer, FS  showed a large underestimation, whereas CON and DRI showed a similar pattern of overestimated precipitation. The precipitation in Eastern Europe  was overestimated by the model during most of the year, except for summer. (Fig.  5f, **?**). All simulations demonstrated considerable agreement on the estimation of the summer precipitation. The yearly mean precipitation by CON was lowest with 2.0 mm day$^{-1}$ and highest when using FS with 2.1 mm day$^{-1}$, as compared to 1.6 mm day$^{-1}$ by the observations (Fig. 5f).

In summary, the  three downscaling approaches overestimated the precipitation, except for an underestimation for Mid-Europe and Eastern Europe in particular months. On a yearly basis, the differences between CON, DRI and FS were small, but on a monthly basis, the magnitude of differences  depended strongly on the region of interest. There  were larger differences between the model simulations for Mid-Europe and Eastern Europe compared to the small differences  for the Iberian Peninsula.

**4   Validation of surface fluxes**

The ~~land-atmosphere feedback is strongest during summer, as the land surface is characterised by more local interactions with the overlaying atmosphere (**?**). Therefore, the following analysis focuses on this season only. We evaluated the differences in the 2 m temperature and the daily accumulated precipitation between the two downscaled simulations. We assumed two-way interactions between the precipitation/temperature and the soil moisture. This feedback mainly concerns the soil moisture at root zone, where the soil moisture impacts the climate by affecting the plants' transpiration (**??**).~~

**4.1**

 spatial distributions of the 5-year daily maximum Bowen Ratio (BR) of CON, DRI and FS were compared to FLUXNET observations, for the summer period only (Fig. 6a,b,c). The corresponding spatial distributions of the 10-year daily maximum BR of CON, DRI and FS were evaluated with respect to the results for the 5-year period (Fig. 6d,e,f). The

over France, Mid-Europe and Eastern Europe. The smallest differences of 0.5-1.5 °C are situated at the Iberian Peninsula, the Mediterranean, the British Isles and the Alps. The precipitation differences are mostly negative with values of -20 % to -40 % for Mid-Europe, and mixed positive and negative for the Iberian Peninsula and the Mediterranean mean diurnal cycles of the surface energy fluxes are illustrated over the 5-year summer period 1996-2000 for the FLUXNET stations of Vielsalm and Collelongo and their corresponding model grid points (Fig. ??b). The extreme differences in northern Africa are related to boundary effects, as the relaxation zone was not excluded here, hence they are not physical.

Finally, the soil moisture differences are primarily negative with 20 % drier soils for Mid-Europe, the British Isles and Eastern Europe, while no differences between FS and CON are present at the Iberian Peninsula and Mediterranean 7, Table 3).

The daily maximum BR showed a strong gradient of increasing values towards the south of the domain (Fig. ??6a,b,c). We focused again on the previously selected subdomains with distinct climate regimes, as they represented the largest variation in the soil moisture results. For Mid-Europe and Eastern Europe, the deep soil moisture simulated by FS decreases more sharply during the spring compared to the deep soil moisture simulated However, large differences appeared for the three downscaling approaches, particularly for the continental part of the domain. Relatively low values of 0 to 1 were represented by CON, and reaches lower minima during summer (Fig. ??b,c). Besides, the values simulated by FS are not able to restore to the values simulated by CON towards the end of the year. This leads to drier soils in FS compared to CON for these particular subdomains. The soil moisture deficit simulated by FScould amplify the summer temperature extremes (?). This is in contrast to the Iberian Peninsula, where the difference in deep soil moisture between CON and FS is small during spring and only enhances from the end of summer onwards (Fig. ??a) while DRI showed BR values of 0.5 to 1 and highest values of 2 to 3 were expressed by FS.

**4.1  Soil moisture-temperature/precipitation feedback**

We did not evaluate a one-way effect of the difference (between the downscaled simulations) of one variable on the difference of the other variable, i. e. a cause and effect relationship. Instead, we assumed a two-way interaction between the differences of soil moisture and the differences of temperature/precipitation. Therefore, a correlation analysis was applied, which looked as follows. First, the mean difference was calculated between FS and CON for each summer. Then, the correlation in time per grid point was determined without any spatial correlation.

The temperature differences are mostly negatively correlated to the soil moisture differences When the value is lower (higher) than 1, the latent heat flux is higher (lower) than the sensible heat flux. The FLUXNET observations for Vielsalm and Collelongo were displayed, and indicated best agreement with DRI (Fig. 6b), expressed by values of 1.12 and 1.32 respectively (Table 3). Though this validation was based on 5 summer periods only from 1996 to 2000, it was still robust as indicated by the corresponding plot for the 10-year summer period from 1991 to 2000 (Fig. ??a),while the precipitation differences were mostly positively correlated to the soil moisture differences (Fig. ??b). A different feedback can be observed for France,Mid-Europe, the Alps and Eastern Europe versus the British Isles, the Iberian Peninsula and the Mediterranean (Table ??). With respect to the correlation of soil moisture difference with 2 m temperature (precipitation) difference, the former subdomains show values of 0.32-0.55 (0.24-0.40), considerably larger than the values of 0.05-0.25 (0.15-0.17)for the latter subdomains (Table

[revised manuscript text omitted]

For Collelongo, H was underestimated by the model during daytime and overestimated during nighttime, except for DRI which demonstrated a good agreement with the observations. Yet again, the model overestimated LE during daytime, except for DRI. The daily maximum H for DRI of 247 $W^{-2}$ was close to the observed value of 253 $W^{-2}$, whereas CON and FS

5  simulated much lower values of 159 $W^{-2}$ and 197 $W^{-2}$ respectively (Table 3). The simulated LE showed the largest difference with the observed one using CON. Regarding BR, the simulation by DRI with a value of 1.35 was in very good agreement with the observations.  The DRI simulation resulted in the correct partitioning of the surface energy fluxes at Collelongo. CON was not performing

10 well in simulating the correct partitioning~~of the fluxes, with respect to the observations. In contrast to BE-Vie, the simulated sensible heat flux at IT-Col by both CON and FS is lower than the latent heat flux, resulting a BR of 0.81 and 0.82 at 13UTC respectively, compared to a BR of 1.23 at 13UTC for the observations (Fig. ??c, d). Consequently, the energy heat fluxesat IT-Col are not influenced by the downscaling setup, which is in agreement with the previous findings~~, while FS had already much improved as compared to CON.

15 In summary, RN was underestimated by the model, whereas H was underestimated and LE was overestimated. However, DRI performed well for H at Vielsalm and for LE at Collelongo. For Colellongo, this resulted in a correct simulation of the partitioning of the surface energy fluxes, translated into an excellent value for BR. Least well simulated were CON and G. The use of a daily reinitialised atmosphere improved the correct partitioning of the surface energy fluxes. FS could not improve the representation of the surface energy fluxes for both stations with respect to DRI. The validation of G was not conclusive, as

20 this parameter needs to be revised with an improved residue layer.

**5   Conclusions**

An assessment of  three downscaling approaches has been performed using the regional climate model ALARO-0 coupled to the land surface model  SURFEXv5, with lateral and initial boundary conditions from ERA-Interim.

25  The simulations were applied for a 10-year period from 1991 to 2000, for a Western European domain. The performance of ALARO-0 with SURFEX has already been validated for NWP applications (?), but not yet for long-term climate simulations.

We compared the common used  approach of a continuous climate simulation with  two alternative

30 aprooaches of frequently reinitialising the RCM simulation towards its driving field, combined with either a daily reinitialised or continuous surface. The use of a daily reinitialised atmosphere outperformed

the continuous approach for winter and summer 2 m temperature, and detoriorated the summer precipitation.  However, the use of a continuous surface next to a daily reinitialised atmosphere improved the summer precipitation with respect to the continuous approach. Furthermore, it improved the winter 2 m temperature, whereas it resulted in a neutral impact on the summer 2 m temperature and the winter precipitation, despite a slight deterioration at the Mediterranean. The SSTs were  reini-

5    tialised daily together with the  atmosphere, as compared to the monthly updated SSTs in the continuous approach.

The seasonal cycle of the 2 m temperature and precipitation was different for three selected subdomains that covered large climate variability. Both the temperature climate of Mid-Europe and the continental climate of Eastern Europe indicated more

10   seasonal variability than the Mediterranean climate of the Iberian Peninsula. The simulation of the 2 m temperature had improved when applying daily reinitialised atmosphere with continuous surface, despite an overestimation of the summer 2 m temperature, . The model disagreed more for precipitation, because of the forcing towards the too wet driving field of ERA-Interim and the low spatial coverage by the observations in some regions. It was clear that the agreement

15   for the precipitation between the model and the observations was highest during summer, while other seasons showed stronger deviations.

 During summer, the interaction between the land surface and the overlaying atmosphere is largest. The 2 m temperature  interacts with the soil moisture  and influences the partitioning of the surface energy fluxes. The

20   daily reinitialisation of the atmosphere improved the representation of a correct partitioning, though the latent heat was highly overestimated for Vielsalm and resulted in a too low value as compared to  the ~~Iberian Peninsula and the Mediterranean. In addition, the opposite partitioning of the latent

25   and sensible heat fluxes of the two downscaling simulations at BE-Vie confirms this land-atmosphere feedback. Moreover, the comparison with the energy fluxes distribution at IT-Col supports our finding that the coupling strength has been impacted by the choice of the downscaling setup.~~ FLUXNET observations. Still, this approach outperformed the use of a continuous simulation. For a more comprehensive analysis, we recommend to include more FLUXNET stations. A more in-depth analysis on the interaction between 2 m temperature, precipitation, and surface energy fluxes can reveal soil-moisture-temperature

30   coupling (**?**), but this lies outside the scope of this study.

In conclusion, this study demonstrated that the approach of a daily reinitialised atmosphere was superior over the continuous approach. The use of a continuous surface next to a daily reinitialised atmosphere even improved the winter temperature and

summer precipitation. The latter approach is highly recommended in a setup with GCM forcing, as imperfect initial and lateral boundary conditions are applied.

*Code availability* The used ALADIN codes, along with all related intellectual property rights, are owned by the Members of

5    the ALADIN consortium. Access to the ALADIN System, or elements thereof, can be granted upon request and for research purposes only. The used SURFEX Codes are freely available, together with the ECOCLIMAP database, at http://www.cnrm-game-meteo.fr/

*Data availability* This study is based on large datasets written in .FA and .lfi format. The relevant output is exported to R datasets. Due to licensing restrictions, this model output is not made publicly available. However, for the purpose of the review,

10   the data can be made available for the editor and reviewer upon request, by contacting Julie Berckmans.

*Author contributions.* J. Berckmans performed the model simulations CON, DRI and FS and analysed the results. J. Berckmans drafted the manuscript. O. Giot and R. De Troch designed R-tools for the analysis. O. Giot designed the experiment CON. R. Hamdi designed the experiment DRI and FS and developed the model code for the implementation of SURFEX within ALARO-0. P. Termonia and R. Ceulemans provided overall guidance during the project. R. Ceulemans and R. Hamdi were the project contractor. All co-authors contributed

15   to the writing and the revising of the manuscript.

*Acknowledgements.* We acknowledge the E-OBS dataset from the EU-FP6 project ENSEMBLES (http://ensembles-eu.metoffice.com) and the data providers in the ECA&D project (http://www.ecad.eu). This work used eddy covariance data acquired and shared by the FLUXNET community

610   The validation data have been collected and prepared by the individual site PIs and their teams. We would like to thank Marc Aubinet (Vielsalm) and Giorgio Matteucci (Collelongo) for contributing data to this study. This research was funded by the Belgian Federal Science Policy Office under the BRAIN.be program

615  as MASC contract #BR/121/A2. The authors also thank Annelies Duerinckx (Royal Meteorological Institute of Belgium, Brussels) for the useful discussions.

The daily mean 2 m temperature bias (°C) and RMSE (in brackets) between the downscaled simulations and E-OBS for the total domain and the subdomains (BI, IP, FR, ME, AL, MD, EA) during DJF and JJA for the 10-year period 1991-2000.

The daily mean 2 m temperature bias (°C) and RMSE (in brackets) between the downscaled simulations and E-OBS for the total domain and the subdomains (BI, IP, FR, ME, AL, MD, EA) during DJF and JJA for the 10-year period 1991-2000.**Table 1.** ~~Overview on the experiments with ALARO coupled to SURFEX carried out in the present study and on the existing experiment with ALARO using ISBA.Acronym Description Land surface model Downscaling setup Reference Historical PeriodCRDX ALARO-0 20 km ISBA Continuous atmosphere - E-OBS 01-01-1991 - 31-12-2000 continuous surface CON ALARO-0 20 km SURFEX Continuous atmosphere - E-OBS 01-01-1991 - 31-12-2000 continuous surface FS ALARO-0 20 km SURFEX Reinitialised atmosphere - E-OBS 01-01-1991 - 31-12-2000 continuous surface CON-FS ALARO-0 20 km SURFEX Both downscaling setups FLUXNET 01-06-2000 - 31-08-2000~~

The daily mean 2 m temperature bias (°C) and RMSE (in brackets) between the downscaled simulations and E-OBS for the total domain and the subdomains (BI, IP, FR, ME, AL, MD, EA) during DJF and JJA for the 10-year period 1991-2000.

| | | TOTAL | BI | IP | FR | ME | AL | |
|---|---|---|---|---|---|---|---|---|
| DJF |  CON |  1.8 (2.5) |  1.1 (2.0) |  2.2 (2.7) |  1.5 (2.2) |  1.3 (2.0) |  3.0 (3.8) |  2.4 |
| |  DRI |  1.2 (2.8) |  1.0 (2.7) |  1.6 (2.7) |  1.2 (2.9) |  0.7 (2.6) |  1.4 (3.4) |  2.1 |
| | FS |  1.0 (2.8) |  0.3 (2.8) |  1.3 (2.5) |  0.7 (2.8) |  0.4 (2.6) |  2.1 (3.8) |  1.2 |
| JJA |  CON |  0.6 (2.0) |  1.7 (2.0) |  0.5 (1.7) |  1.2 (1.9) |  1.3 (1.9) |  1.8 (2.6) |  0.5 |
| |  DRI |  0.1 (2.3) |  0.9 (2.0) |  0.3 (2.2) |  0.7 (2.4) |  0.3 (2.1) |  0.8 (2.3) |  0.6 |
| | FS |  0.9 (2.7) |  0.7 (2.2) |  0.5 (2.4) |  1.0 (3.1) |  1.3 (2.8) |  0.0 (2.5) |  0.7 |

**Table 2.** The daily accumulated precipitation bias (%) and RMSE (in brackets) between the downscaled simulations and E-OBS for the total domain and the subdomains (BI, IP, FR, ME, AL, MD, EA) during DJF and JJA for the 10-year period 1991-2000.

| | | TOTAL | BI | IP | FR | ME | AL | |
|---|---|---|---|---|---|---|---|---|
| DJF |  CON |  16.6 (3.8) |  4.5 (4.5) |  16.1 (4.6) |  29.0 (3.6) |  25.4 (2.7) |  11.2 (4.7) | 46. |
| |  DRI |  20.9 (4.8) |  6.6 (5.2) |  21.2 (5.6) |  26.8 (4.8) |  27.9 (3.8) |  24.1 (6.3) | 46. |
| | FS |  36.3 (5.4) |  16.9 (5.5) |  31.3 (6.2) |  38.2 (5.2) |  35.7 (4.0) |  26.7 (6.7) | 109.5 |
| JJA |  CON |  12.1 (4.2) |  24.7 (4.4) |  11.5 (2.9) |  12.0 (4.4) |  11.9 (5.0) |  32.6 (7.3) | 23. |
| |  DRI |  22.5 (4.7) |  27.0 (4.7) |  30.0 (3.4) |  18.3 (5.1) |  8.8 (5.5) |  48.2 (8.9) | 64. |
| | FS |  3.6 (4.5) |  17.4 (4.6) |  13.0 (3.2) |  -7.0 (4.6) |  -13.4 (5.1) |  23.5 (8.3) | 56. |

**Table 3.** The daily maximum surface energy fluxes ($Wm^{-2}$) averaged over the 5-year JJA period 1996-2000 and the 10-year period 1991-2000 (in brackets).

| |  |  RN |  H |  LE |  G |  BR |
|---|---|---|---|---|---|---|
| Vielsalm |  OBS |  417 | 151 | 134 | 11 | 1.12 |
|  |  CON |  395 (404) |  118 (113) |  250 (261) |  47 (47) |  0.47 (0.43) |
| |  DRI |  388 (398) | 151 (159) | 195 (193) | 57 (58) | 0.78 (0.82) |
|  |  FS |  405 (411) |  139 (152) |  229 (221) |  46 (49) |  0.61 (0.69) |
| Collelongo |  OBS |  538 | 253 | 192 | -1.39 | 1.32 |
| | CON | 480 (481) | 159 (147) | 270 (289) | 111 (108) | 0.59 (0.51) |
| | DRI | 496 (494) | 247 (232) | 183 (194) | 143 (140) | 1.35 (1.19) |
| | FS | 501 (498) | 197 (191) | 236 (247) | 111 (110) | 0.83 (0.77) |

[Figure]

**Figure 1.** The total domain on 20 km horizontal resolution and the subdomains (BI, IP, FR, ME, AL, MD, EA) based on the subdomains selected in the EURO-CORDEX framework. The color represents the orography (m) in the ALARO+SURFEX setup. The two black dots represent the FLUXNET stations BE-Vie (Vielsalm ,(Belgium) and IT-Col (Collelongo ,(Italy).

[Figure]

**Figure 2.** The setup of the three downscaling approaches CON, DRI and FS used in this study. It represents the spin-up time for the different simulations, the analysis period of the total experiment and the update frequency of the lateral and initial boundary conditions.

[Figure]

**Figure 3.** Daily mean 2 m temperature  (°C) for E-OBS DJF (a) and JJA (b), and absolute bias (°C) of the model with E-OBS for  CON DJF (c) and JJA (d), for  DRI DJF (e) and JJA (f) and for FS DJF (g) and JJA (h), all at a 20 km horizontal resolution for  the 10-year period 1991-2000. The dots represent the grid points with a significant difference at 5%, using the Student's t-test with a null hypothesis stating that the means of the model and observations are equal.

[Figure]

**Figure 4.** Daily accumulated precipitation  (mm day$^{-1}$) for E-OBS DJF (a) and JJA (b), and relative bias (%) of the model with E-OBS for  CON DJF (c) and JJA (d), for  DRI DJF (e) and JJA (f) and for FS DJF (g) and JJA (h), all at a 20 km horizontal resolution for a 10-year period 1991-2000. The dots represent the grid points with significant different variations at 5%, using the F-test with a null hypothesis stating that the variances of the model and observations are equal.

[Figure]

 , and daily  accumulated precipitation ($m^3/m^3$ mm day$^{-1}$)  for (a) the Iberian Peninsula, (e) Mid-Europe, and (f) Eastern Europe, averaged over the 10-year period 1991-2000. Both the mean and standard deviation (SD) are displayed as text.

 , and daily  accumulated precipitation ($m^3/m^3$ mm day$^{-1}$)  for (a) the Iberian Peninsula, (e) Mid-Europe, and (e) Eastern Europe, averaged over the 10-year period 1991-2000. Both the mean and standard deviation (SD) are displayed as text.

**Figure 5.** Mean annual cycle of the daily 2 m temperature (°C)  with E-OBS, CON and FS for (a) the Iberian Peninsula, (b) Mid-Europe, and (c) Eastern Europe.

 , and daily  accumulated precipitation ($m^3/m^3$ mm day$^{-1}$)  for (a) the Iberian Peninsula, (e) Mid-Europe, and (e) Eastern Europe, averaged over the 10-year period 1991-2000. Both the mean and standard deviation (SD) are displayed as text.

[Figure]

**Figure 6.**  Daily maximum Bowen ratio averaged over the  5-year JJA period 1996-2000 for (a) CON, (c) DRI and (e) FS and averaged over the  10-year JJA period 1991-2000 for (b) CON, (d) DRI and (f) FS. The dots represent the  values for the FLUXNET stations Vielsalm (Belgium) and Collelongo (Italy).

[Figure]

**Figure 7.** Daily cycle of the energy fluxes (W m$^{-2}$) in JJA 1996-2000 for Vielsalm in the top row and Collelongo in bottom row for (a,c) H, and (b,d) LE, for the FLUXNET observations and their corresponding model grid points by CON, DRI and FS. The error bars represent the standard deviation of the observations.

---

## Author Comment (AC3) · 4 Nov 2016

[1,2]JulieBerckmans [1,2]OlivierGiot [1,3]RozemienDe Troch [1,3]RafiqHamdi [2]Reinhart Ceulemans [1,3]PietTermonia

[1]Royal Meteorological Institute, Brussels, Belgium [2]Centre of Excellence PLECO (Plant and Vegetation Ecology), Department of Biology, University of Antwerp, Antwerp, Belgium [3]Department of Physics and Astronomy, Ghent University, Ghent, Belgium

Julie Berckmans (julie.berckmans@meteo.be)

[Figure]

November 4, 2016

**1  Reply to the Editor**

Dear Editor,

we have prepared a majorily improved version of the manuscript by incorporating all suggestions and critical comments raised by both reviewers. Please find attached our response to the referees' comments on our above mentioned manuscript, titled "Reinitialised versus continuous regional climate simulations using ALARO-0 coupled to the land surface model SURFEX". Below mentioned you will find our detailed responses to all the reviewers' comments and suggestions (put in italics and red). We have also explained where and how they were incorporated in the revised manuscipt.

**2  Reply to Anonymous Referee #1**

We would like to thank the Anonymous Referee #1 for the encouraging and constructive comments, which have improved the manuscript. Below is a list of modifications that we have implemented based on your major and specific comments. The supplemented "manuscript_figures.zip" contains Figures 1 and 2, together with the

marked-up manuscript showing the changes made in the new version.

Major comments:

*(1) I was disappointed on the level of detail describing all the different models referred to in this manuscript: ALARO-0, SURFEX, ISBA, TEB and ALADIN and found that if I wanted a reasonable understanding of what the key parameterizations were or the runtime options used or the hierarchy of these models I would need to consult several other manuscripts. . .an exhausting exercise when there are so many models referred to here. Furthermore, I found it confusing to follow how the model is run. In RCMs like WRF, reanalysis (or GCM) data is used to update the BCs on a 6 hourly interval, which is necessary when running climate simulations (rather than short-term forecasts) to avoid drift. Therefore from the beginning I was confused by what the authors mean by "continuous" and "reinitialized" for an RCM when running a 10 year simulation. What variables are reinitialized, how are they reinitialized (at the boundaries or across the domain) and at what frequency? This information was not clearly articulated, making it difficult for someone to reproduce the experimental design.*

The level of detail for the different models has been increased in the revised manuscript for all models described. The setup of the experiment has been clarified, using a schematic diagram (Fig. 2 in revised manuscript), showing the different downscaling simulations. The meaning of continuous and reinitialised is better defined and supported by the diagram. The wording "initial conditions" and "lateral boundary conditions" has been used in its correct context, to get a proper understanding of the differences between the downscaling simulations. The new version (Page 4 Lines 26-28) reads:
"The zonal and meridional wind components, atmospheric temperature, specific humidity, surface pressure and surface components were provided every 6 hrs as lateral boundary conditions and interpolated hourly. They were introduced as initial

conditions accross the domain."

*(2) Although there are 3 objectives detailed in the introduction, I could see two possible aims of the manuscript: (a) Trying to show that forecast skill is improved when a new land surface scheme is added (CRDX vs. CON) – which is documented in Hamdi et al. 2014 (b) Trying to show that forecast skill is improved when reinitializing daily (FS vs. CON) If the aim is to present the benefits of the daily reinitialization, then perhaps excluding the CRDX results would improve the focus of the manuscript.*

The aim of the manuscript was to present the improved simulation skill of the daily reinitialisation. It is a good suggestion to focus on this objective. We have therefore removed the first part on the comparison with the EURO-CORDEX simulations from the revised manuscript. The latter will be used in a separate study.

*(3) There are different spin-up periods for FS (3 months) and CON (1 year). No explanation is provided for why the set up is different between the experiments. In particular, if the same land surface scheme is used, then best practice would be to have the same spin-up period for the land-surface state variables (soil moisture and soil temperature). If FS and CON have different spin-up lengths then how can the authors be sure that the differences in simulation skill are due to the "continuous/reinitialized" configuration and not the spin-up? This would actually make me advocate that the whole experiment needs to be run again with same configuration (e.g. Bcs, spin-up length, IC) so that differences in skill between "continuous" or "reinitialized" runtime modes can be fairly evaluated. At the moment I don't think this is really possible.*

This is a good point. Based on a general consensus within the climate community, we have used one-year spin-up for the continuous simulation. When making FS simulations, we generally run each year in parallel to reduce computing time. Therefore, we apply the smallest possible spin-up time which still provides an equilibrium state. Past

tests with different initialisation times and periods for a domain covering Belgium and the neighbouring countries showed that starting in March and keeping a spin-up time of three months is reasonable for the model to come into an equilibrium state (see Fig. 1).

Fig. 1 represents the deep soil moisture simulated by the three dynamical downscaling approaches DRI, FS, and CON, for the period of January 1990 to January 1991. The initial conditions for the surface restarted every day within DRI. For CON and FS, the initial conditions for the surface were initialised once at the beginning of each run. For FS the initialisation of the surface in March 1990 resulted in a shorter time period to reach equilibrium than the initialisation in January or February 1990. When starting in March 1990, the FS lines were close in June and thereafter. This means that the variability within FS is smaller than the variability between FS and CON. However, when starting in January, we would have to spin-up the model for 6 months until July to reach equilibrium. A similar test was done for our study domain. Fig. 2 shows the deep soil moisture simulated by the two dynamical downscaling approaches CON and FS, for the period of March 1991 to August 1991. FS SHORTSPINUP shows the deep soil moisture when starting the spin up in March 1991, while FS LONGSPINUP shows the deep soil moisture when starting the spin up the previous year in March 1990. The variability within FS is smaller than the variability between FS and CON after 3 months, in June. Both results show that the different spin-up periods for CON and FS do not impact the simulation skill of the two downscaling simulations, whereas the different configurations do impact the simulation skill.
The new version (Page 5 Lines 21-22):
"Although CON required one year spin-up time, 3 months were sufficient for the FS deep soil moisture to reach equilibrium state, when starting in March (not shown)."

*(4) I wasn't convinced at all by the analysis on the land-atmosphere feedback. Perhaps it would be good to consider the coupling metrics in detailed in Lorenz et al. (2015) that*

[Figure]

*are suitable for the fully coupled simulations. It would be more convincing to calculate the coupling metrics for each experiment independently and then evaluate the difference between FS and CON to examine changes in coupling. However, is this analysis relevant here, given that the differences between CON and FS is the frequency that the lateral BCs are updated, not the land surface state as understood from Page 4/5: "The soil variables evolved freely after the first initialisation and were never corrected or nudged in the course of the simulation."*

We agree with the comment of the reviewer; consequently we removed the correlation analysis from the revised manuscript. In the revised manuscript we now concentrate the analysis on the mean state of the atmosphere and the surface, but we do not investigate the feedback. However, we still evaluate the diurnal cycle of the surface energy fluxes, as they are impacted by a different response of the surface to the continuous vs. reinitialised atmosphere. This has been done in section 4 in the revised manuscript, where we validated the spatial distribution of the Bowen Ratio and the diurnal cycle of the surface energy fluxes. The coupling metrics as in Lorenz et al. (2015) might be interesting for a future separate study.

*(5) I am a bit concerned about the limited number of sites used to evaluate the experiments against FLUXNET. This is likely due to the choice of simulation period (1991-2000) where FLUXNET data coverage is limited. It sounds like the authors were already aware of this limitation too. Perhaps a more recent simulation period would resolve this issue when more FLUXNET sites are available for a more rigorous validation. Alternatively, the authors could consider the LandFLUX (Mueller et al., 2013) or GLEAM (Miralles et al., 2013) datasets to validate the surface fluxes more comprehensively.*

The simulations used for this study, driven by ERA-Interim, were done in parallel to the model simulations driven by the global climate model ARPEGE CMIP5 (1976-2005). The period of 1991-2000 was chosen to have an overlapping period for comparison.

At the onset of the present study we had not decided to investigate the surface fluxes. This decision came later on, to look more into detail at the surface processes to explain the differences between the downscaling approaches. In particular, the FLUXNET database was selected for the model validation as our research group is familiar with this network. Besides, one of the co-authors is PI of a FLUXNET station (not used in this study).

*(6) The surface fluxes in the FS and CON configurations are also only evaluated in a second set of shorter simulations (3 months) rather than the original 10 year simulations. It would make better sense to evaluate the fluxes and land-atmosphere feedback in the 10 year simulations given that the purpose of the manuscript is to evaluate the simulation skill of long simulations. Unfortunately this provides the reader with the impression that the experimental design was either poorly designed or that a random bunch of simulations with different set-ups were cobbled together to evaluate the different runtime modes.*

We agree with the reviewer's comment. In the revised manuscript the fluxes were calculated for 10 years of summer, to make this consistent with the overall setup of the experiment. As the observations from FLUXNET only provide data from 1996 onwards, the overlapping period with the model is 5 years. Therefore, the Bowen Ratio was presented in the revised manuscript for 5 years, as well as the results for the stations. Additionally, the Bowen ratio was presented for 10 years, to indicate that the 5-year period is still robust for the 10-year period, and can be used for the model validation of the fluxes.

*(7) Due to the writing style, I found the paper hard to read in many places. The structure also requires refinement, as there are many instances where information is provided in the wrong section that would be more useful in another.*

The structure of the revised text has been improved. The revised text now fits better within each of the sections.

Specific comments:

Abstract

*Here I got the impression that the manuscript was about evaluating the updated land surface scheme rather than the different running modes. Please revise the abstract to appropriately reflect the aim and scope of the manuscript and the key results.*

The Abstract has been revised as suggested.
The new version (Page 1 Lines 4-9) reads:
"We evaluated the dependence of the simulation potential on the running mode of the ALARO model coupled to the land surface model SURFEX, driven by the European Centre for Medium-Range Weather Forecasts (ECMWF) Interim Re-Analysis (ERA-Interim) data. Three types of downscaling simulations were carried out for a 10-year period covering 1991 to 2000, over a Western European domain at 20 km horizontal resolution: ..."

*Please define all acronyms.*

This has been done in the revised manuscript. All acronyms have been properly and clearly defined.

*It is perhaps not necessary to mention the ALADIN modeling system here to avoid overwhelming the reader with acronyms.*

We appreciate the reviewer's suggestion. The acronym has not been mentioned

anymore in the revised Abstract.

*Sentence starting "This contribution . . ." perhaps better to say "We evaluate the dependence of simulation skill on the running mode (continuous or reinitialized) of the ALARO-0 model."*

We thank the reviewer for the suggestion; the sentence has been reworded in the revised manuscript.
The new versions (Page 1 Lines 4-5):
"We evaluated the dependence of the simulation potential on the running mode of the ALARO model coupled to the land surface model SURFEX, and driven by the European Centre for Medium-Range Weather Forecasts (ECMWF) Interim Re-Analysis (ERA-Interim) data."

*Sentence starting with: "The results show that the introduction of SURFEX..." Could be revised to something like: "The results show that the SURFEX land surface scheme improves the simulation of 2 m temperature but has a negligible impact of the simulation skill of daily precipitation totals."*

We thank the reviewer for the suggestion; the sentence has been reworded in the revised manuscript.
The new versions (Page 1 Lines 10-12):
" The results showed that the daily reinitialisation of the atmosphere improved the simulation of the 2 m temperature for all seasons. It revealed a neutral impact on the daily precipitation totals during winter, but the results were improved for the summer when the surface was kept continuous."

Introduction

*The narrative introduces the reader to global climate modeling, numerical weather pre-diction, regional climate modeling, downscaling, limited area modeling. However there is insufficient detail on their differences particularly on the frequency that Bcs are up-dated, what is meant by 'continuous' or how the 'reinitialization' is done (at the bound-aries or across the domain). This needs simplifying, and can perhaps be resolved by limiting to just a few terms that are explicitly relevant to the study.*

We added more detail to the revised manuscript as suggested. The explanation on the initial and the lateral boundary conditions has been significantly improved. We added a figure/schematic diagram (Figure 2 of the revised manuscript) explaining the different downscaling approaches and their simulation periods, spin-up times, frequency of initialisation and update of lateral boundary conditions etc.

*It is never defined explicitly here or in the methods what "frequent reinitializations" and "continuous simulations" means. I got the impression that climate simulations were run using an RCM where the BCs were updated once a month (in CON) or daily (in FS) when most state of the art RCMs would be updating the BCs on a more frequent basis.*

In the revised manuscript this has been explained better now, in combination with a new figure (Figure 2 in the revised manuscript). The LBCs where updated every 6 hours, but the update frequency of the initial conditions was different. For CON there was only one single initialisation, but with monthly updates of the SSTs; for DRI and FS there was a daily reinitialisation. This has been rephrased and clarified in the revised manuscript.
The new version (Page 4 Lines 26-28) reads:
""The zonal and meridional wind components, atmospheric temperature, specific humidity, surface pressure and surface components were provided every 6 hrs as lateral boundary conditions and interpolated hourly. They were introduced as initial conditions accross the domain."
And Page 4-5 Lines 34-2:

" To evaluate the sensitivity of the model to the update frequency of the initial conditions, three types of downscaling approaches were conducted with ALARO version 0 coupled to SURFEX version 5."
And further on in the description of the downscaling approaches.

*Page 2 Line 6: Please check for spelling errors!*

All spelling errors have been corrected in the revised manuscript as requested.

*Page 2 Line 14: "The model used in this study is the ALARO-0 model configuration of the ALADIN system." This won't mean much to those who have never used this model configuration. Perhaps this information is best in the methods where you describe the model and can then elaborate on the specific details of the ALARO-0 model configuration.*

We thank the reviewer for this suggestion. This is moved to the Model section 2.1 and the model configuration has been described more in-depth to get a good idea on the specific details of the model.
The new version (Page 3 Lines 17-18) reads: "The regional climate model used in this study is the ALARO model version 0, a configuration of the Aire Limitée Adaptation Dynamique Développement International (ALADIN) model with improved physical parameterisations (Gerard et al., 2009)."

*Page 2 Line 17: Interaction appears to be used twice here. Please correct.*

Corrected in the revised manuscript as suggested.

*Page 2 Line 19: Please provide the reference evaluating ALARO-0 with ISBA for continuous simulations.*

[Figure]

The reference to Giot et al. (2016) had already been given in the original manuscript, but in the revised manuscript we moved the location of the reference so that it is more clear.
The new version (Page 4 Lines 3-4) reads:
"In addition, this setup has been validated for continuous climate simulations and is now contributing to the EURO-CORDEX project (Giot et al., 2016; Jacob et al., 2014)."

*Page 2 Line 21: "has been implemented in the ALARO-0 version." Seems like the version number is missing at the end of the sentence.*

The version number is zero; this has been corrected and explicitly stated in the revised manuscript in Section 2. In the Introduction, ALARO is mentioned without version number.
The new version (Page 3 Lines 17-18) reads:
"The regional climate model used in this study is the ALARO model version 0, a configuration of the Aire Limitée Adaptation Dynamique Développement International (ALADIN) model with improved physical parameterisations (Gerard et al., 2009)."

*Page 2 Line 22: "the introduction of SURFEX with ALARO-0 has shown neutral to positive. . ." This phrasing is used a couple of times, perhaps its best to be more specific; which variables show no sensitivity, which ones are sensitive and what is the sign and magnitude?*

In the revised manuscript we have included a more detailed description of the variables that show sensitivity as well as the sign, without mentioning the magnitude.
The new version (Page 4 Lines 6-9) reads:
"With respect to NWP applications, the introduction of SURFEXv5 within ALARO-0 has shown neutral effects on the winter 2 m temperature and on the vertical profile of the wind speed. However, it has shown positive effects on the summer 2 m temperature,

2 m relative humidity, and resulted in improved precipitation scores compared to the previously used ISBA model (Hamdi et al., 2014)."

*Page 2 Line 24: "Therefore the evaluation of SURFEX within ALARO-0 is highly demanding for regional climate simulations" Hamdi et al. 2014 already evaluates SUR-FEX within ALARO-0 so perhaps the authors need to be specific here by saying that Hamdi et al. evaluate SURFEX within ALARO-0 for NWP but this manuscript will evaluate the same model environment for longer simulations.*

In the revised manuscript we added a sentence as suggested.
The new version (Page 4 Lines 9-10):
"Next to the validation of this setup for NWP, the implementation of SURFEXv5 within ALARO-0 is highly demanding for long-term climate simulations."

*Page 2 Line 27: "The second objective is to evaluate the continuous setup with an upper air daily reinitialized setup, where the surface is simulated continuously." I think this is where a lot of the confusion on terminology and model runtime configuration stems from. It would help if you can articulate what variables are continuous for each experiment, what variables are reinitialized and the frequency to which this is done. Could also add that information to Table 1.*

The requested information has been added in Figure 2 of the revised manuscript. The initial and lateral boundary conditions for the atmosphere and surface (as stated earlier) are either initialised once at the beginning or reinitialised daily or reinitialised daily for the atmosphere and only initialised once at the beginning for the surface. For the continuous simulation, the SSTs are still reinitialised monthly.

*Page 2 Line 29: "Therefore one expects to see improvements for the second method over continental parts of the domain, but not so much in coastal areas." Why not in the*

*coastal areas?*

When applying a continuous simulation of the regional climate, we made a similar simulation as is commonly done by the climate community, using monthly SST updates. However, when we applied the daily reinitialisation of the atmosphere, the SSTs were updated daily as well. We expect a different behaviour of the model with different SSTs update frequency. As we did not test the sensitivity of the model to daily or monthly updates of SSTs, we are not able to make any statements on the reasons for different behaviour of the model over the coastal areas. This is outside the scope of our study. Therefore, we removed this statement.

*Page 3 Line 2: Update to "Therefore the diurnal cycle of soil moisture was analyzed at particular locations in this study."*

This analysis has been removed from the revised manuscript.

*Page 3 Line 3: This paragraph doesn't quite fit in with the previous narrative; please provide an explanation on why the focus is on the summer season (i.e. when the soil moisture limitation on evapotranspiration is greatest) for those less familiar with the land-atmosphere coupling literature.*

As requested we have provided more background information in the revised manuscript on why we have done this only for summer.
The new version (Page 2 Lines 31-33) reads:
"The surface interacts with the climate through the soil moisture and soil temperature, by influencing the surface energy budget (Giorgi and Mearns, 1999). The soil moisture controls the partitioning of the incoming energy into a latent and sensible heat flux. The soil moisture limitation on the evapotranspiration is largest during the summer (Seneviratne et al., 2010)."

Methods:

*Are all simulations run with the same ERA-Interim BCs? It would be useful to add this information to Table 1 and manuscript text.*

Yes, this information has been added to Figure 2 of the revised manuscript and in the revised text of the manuscript.
The new versions (Page 4 Lines 24-25) reads:
"The regional climate model was driven by initial and lateral boundary conditions provided by the ERA-Interim reanalysis, available at a horizontal resolution of ca. 79 km."

*What data is used for the 'reinitializations' in FS?*

The ERA-Interim reanalysis data have been used to provide the reinitial conditions for the zonal and meridional wind components, atmospheric temperature, specific humidity, surface pressure and surface components. This information has been added to the revised text (see earlier).

*I found most of the narrative of Section 2.1 to go between describing the model/s and information that would be better placed in Section 2.2 Experimental Design – needs revising*

This is a valid comment. The structure of Section 2 has now been revised and improved. The revised Section 2.1 is purely on the model descriptions, and revised Section 2.2 is on the experimental setup of the study.

*Page 3 Line 14: These sentences would be more suitable in Section 2.2 Experimental Design*

Thank you for this suggestion. The sentences have been moved to the revised Section on the Experimental Design.

*Page 3 Line 17: Another model is introduced but not used in this study. Please remove to simplify the narrative.*

We agree that this introduction does not make any sense. Accordingly we have removed this reference from the revised manuscript.

*Page 3 Lines 14-22: There is no detail here on the microphysics, cumulus convection scheme or planetary boundary layer scheme. The reader is not provided with sufficient information on what the ALARO-0 configuration of the ALADIN modeling system actually means. This information is necessary in my opinion, particularly for someone not familiar with the model but interested in what was tested.*

More information on the technical details of the model has been added to the revised manuscript as requested.
The new version (Page 3 Lines 24-27) reads:
"The new physical parameterisation within the ALARO-0 model was specifically designed to be run at convection-permitting scales, with a particular focus on an improved convection and cloud scheme, developed by Gerard and Geleyn (2005) and further improved by Gerard (2007) and Gerard et al. (2009). "

*Page 3 Line 20: Please change "following-terrain" to "terrain-following"*

This has been changed on Page 3 Line 21.

*Page 3 Lines 23-29: Based on what is written here I got the impression that SUR-FEX was basically ISBA with tiling and coupled to TEB. If the intention is to conduct a*

*comparison between SURFEX and ISBA then the model descriptions need to be much
more explicit on what the differences are between these models.*

The model descriptions have been made more explicit in the revised manuscript. We
have clarified what the differences are about between SURFEX and ISBA. SURFEX
is providing tiles for new surface types, uses ECOCLIMAP as input for the land cover
and is externalised.
The new version (Page 4 Lines 16-21) reads:
"The initial parameterisation ISBA for the nature tile was conserved, and parameter-
isations for the other surface tiles were added, such as the Town Energy Balance
scheme (TEB, Masson, 2000) for the town tile. TEB uses a canopy approach with
three urban energy budgets for the layers roof, wall and road. The ISBA and TEB
scheme were combined, together with parameterisation schemes for inland water and
oceans, and externalised, based on the algorithm of Best et al. (2004). Each tile is
divided in different patches, according to the tile type. These patches correspond to
the plant functional types described in ECOCLIMAP (Masson et al., 2003)."

*Page 3 Line 30: 'ECOCLIMAP' Please define all acronyms the first time they are intro-
duced*

ECOCLIMAP is not an acronym, but the name of a database.

*Page 4 Lines 5-10: I gather that this is an explanation of the what variables are ex-
changed between the land surface model and the atmospheric model. The wording
could be revised to make this easier to understand.*

We have removed this from the revised manuscript, as it does not provide added value
to the content of the manuscript. These details can be found in related literature.

*Page 4 Lines 11-19: If the model is not run on the EURO-CORDEX domain then why provide detail on it? Also: "The present study was done in the framework of another project" please tell the reader what this project is.*

This statement has been deleted from the revised version as we did not have any comparison with EURO-CORDEX output anymore.

*In Table 1 a CRDX experiment is listed but not defined in Section 2.2; which one is it?*

This experiment is no longer part of the manuscript anymore and has been removed.

*Page 4 Line 22: "It started at 00UTC on 01 January 1990, and ran continuously until 00UTC on 1 January 2000. The first year was treated as a spin-up year, and the analysis period covers the 10-year period 01 January 1991 to 31 December 2000." Please revise the inconsistency here as two different end dates are mentioned.*

Done, the references to dates and times have been made consistent in the revised manuscript.
The new version (Page 4 Lines 31-32) reads:
"The analysis covered a 10-year period from 00UTC on 01 January 1991 to 00UTC on 01 January 2001."
And (Page 5 Lines 4-5) reads:
"The model was simulated from 00UTC on 01 January 1990, and ran continuously until 00UTC on 01 January 2001."

*Why are simulations run for 1991-2000 when a more recent period would enable comparison to more FLUXNET sites?*

This time period was selected because a similar experiment was done with forcing from the global climate model ARPEGE CMIP5 version. This forcing covers the period

of 1976-2005 and we selected an overlapping period between the two simulations.

*Why does CON have a spin-up of 1 year starting in January 1990 at 00UTC and why does FS have a 3 month spin-up with simulations starting in March each year at 12UTC? Usually the spin-up and start date should be the same unless the focus of the study is on how long a spin-up is necessary to maximize simulation skill which is not the aim of this paper!*

The 1-year spin-up time is commonly done in climate simulations in the continuous mode. Considering the FS simulations, we showed that the ideal setup of the initial-isation period is three months starting in March. This relatively short spin-up period is enough to reach a equilibrium state (shown earlier in this Reply at point (3)). If we had selected a start in January, the model would have needed a longer spin-up time and the computing time of the simulations would significantly increase. This has been rephrased and clarified in the revised manuscript.
The new version (Page 5 Lines 21-22):
"Although CON required one year spin-up time, 3 months were sufficient for the FS deep soil moisture to reach equilibrium state, when starting in March (not shown)."

*Why are atmospheric variables saved at a 3 hourly interval and land surface variables at daily? It would make sense to save them all at the same interval.*

ALARO and SURFEX are two different models, with different output files. The surface output files are much larger than the atmospheric output files. Therefore, we had to limit the surface output files because of storage limitations at our institute. This has been rephrased in the revised manuscript.
The new version (Page 5 Line 25) reads:
" The model output at every 3 hrs was used for the model evaluation. "
And (Page 5 Lines 32-34) reads:

"As land-surface processes play an important role primarily during summer, the model output was stored at every hour for the summer period of June-July-August (JJA) during the 10-year period."

*Page 4 Line 33: "Each daily simulation extended up to 60 hours of which the first 36 hours were treated as spin-up" This seems to be contradicting the previous explanation where there is a 3 month spin-up for 1 year simulations! Please revise the description of how each experiment was run as it is not clear at the moment and limits the reader's ability to reproduce the experiment.*

The original wording was indeed confusing. Therefore, we used the wording "atmospheric spin-up" in the revised manuscript, which is typically in the order of 24 hours, and the "surface spin-up", which is typically in the order of a few months to one year.
The new version (Page 4 Lines 29-31) reads:
"For the sake of a good understanding, the following description makes a distinction between atmospheric spin-up time, typically of a few days, and surface spin-up time, typically of a few months to one year."

*Page 5 Line 3: "A third simulation was applied for both CON and FS" Technically third and fourth simulations?*

This has been removed, as these are technically not different simulations. For the three downscaling approaches, the variables were stored hourly in addition for the summer.
And (Page 5 Lines 32-34) reads:
"As land-surface processes play an important role primarily during summer, the model output was stored at every hour for the summer period of June-July-August (JJA) during the 10-year period."

*Page 5 the extra CON and FS simulations saving hourly output, why wasn't this done*

*from the start rather than doing additional shorter simulations*

Initially we only intended to investigate atmospheric parameters. We noticed the large changes in summer for 2 m temperature between the different simulations and decided to further investigate surface fluxes. Therefore, we needed hourly output to look at the diurnal cycle. The explanation of the output has been reworded accordingly in the revised text (see earlier).

*Page 5 Line 6: "The first simulation. . ." Is this referring to the CRDX experiment?*

This sentence has been removed from the revised manuscript.

*Page 5 Line 13: This is the first time PRUDENCE is mentioned – define acronym*

Ok, we have defined the project acronym in the revised manuscript as requested.
The new version (Page 5 Lines 28-29) reads:
"project "Prediction of Regional scenarios and Uncertainties for Defining European Climate change risks and Effects" (PRUDENCE) (Christensen et al., 2007)"

*Page 5 Line 16: "relaxation zone was excluded" It looks like Figures 3, 5 and 7 need yo be cropped to exclude the relaxation zone.*

The relaxation zone has been excluded from the plots in the revised manuscript. See Figure 3, Figure 4 and Figure 6 in the revised manuscript.

Observational data:

*Perhaps better to combine the model description, experimental design and observational data under one Methods, Models and Datasets section*

Ok, we have followed the reviewer's suggestion and we have combined Model

description, Experimental design and observational data in the revised manuscript.

*Page 5 Line 21: replace 'sum' with 'total'*

Done; has been replaced in the revised manuscript.
The new version (Page 6 Line 8) reads:
"daily precipitation total."

*Page 5 Line 26: Usually one interpolates to the coarser resolution. i.e. interpolate the model data to the E-OBS resolution.*

In line with the reviewer's suggestion we regridded the model to the E-OBS resolution and we recalculated the differences at the coarser resolution. We have clarified this in the revised manuscript.
The new version (Page 6 Lines 12-14) reads:
"In order to validate the model data, the ALARO-0 data at 20 km horizontal resolution were bilinearly interpolated towards E-OBS at 25 km horizontal resolution and replotted to our study domain."

*Page 5 Line 27: replace 'implied' with 'applied'*

Done, this has been replaced in the revised manuscript.
The new version (Page 6 Lines 15) reads:
"applied"

*Page 5 Line 30: replace 'exchanges' with 'exchange'*

Done; has been replace in the revised manuscript.
The new version (Page 6 Lines 18) reads:
"exchange"
*Page 6 Line 1: remove the sentence starting "The technique. . ."*

Done; the sentence has been removed from the revised manuscript as requested.

*Page 6 Line 4: replace 'with regard to the' with 'against these'*

This sentence is rephrased.
The new version (Page 6 Lines 20-21) reads:
"No gap-filling has been done and the comparison to the model output was only done at hours when no gaps occurred."

*Page 6 Line 5: revise sentence starting "The model resolution is quite low..."*

The sentence has been removed in the revised manuscript, as it is straightforward that model resolution is lower than station resolution.

*In Section 5.3 a justification is provided on why the two FLUXNET sites were selected. It would be more appropriate to put that in Section 3.2.*

This has been clarified in Section 2.3 of the revised manuscript.
The new version (Page 6 Lines 23-25) reads:
"Two FLUXNET stations were selected, that provided data already during this period and where the model grid cell represented more than 50% of the corresponding land cover, to show energy fluxes that were representative for the particular land cover."

Results – more general rather than line by line:

*The values in Table 2 and 3 are referred to more often than Figures 2 and 3. There*

*are also several instances where it is not clear which results are being referenced. In particular, reporting the percentage change between experiments was quite confusing given that these values are not presented in either the Tables or Figures. This meant that I had to spend a lot of time checking where the values were coming from, or calculating the percentage changes myself. This could be resolved by adding detail in the manuscript text on the values shown in the figures. If the percentage change between experiments is quoted then please include these values in the Table or replace with something along the lines of: "CON has a larger bias of X relative to FS which has a bias of Y". This will make it easier for a reader to match the narrative to what is presented in Figures and Tables.*

We thank the reviewer for these suggestions. The text has been rephrased and improved. In the revised manuscript we refer both to the Figures and Tables. The percentage change between simulations (%) were indeed very confusing; consequently we have removed them from the revised manuscript. Only the exact numbers are used in the revised manuscript, as they are represented in the Tables.

*There is a tendency to use words such as 'large', 'slight', 'excessive' or 'improved' in the narrative. Please be specific and insert detail. For example, "the bias improves" could be replaced with "the temperature bias decreases by X in experiment Y". There are instances where this is done, but it has not been applied consistently. However doing so will make it easier for the reader to understand the results that are presented.*

Thank you for this suggestion. In the revised manuscript we have avoided the use of these words, and we predominantly use the exact numbers.
One example (Page 7 Lines 14-15) reads:
"This resulted in a smaller bias for Eastern Europe of -0.3 C and 0.0 C for DRI relative to CON which had a bias of -1.1 C and -0.5 C for winter and summer respectively."

*Why are the temperature biases presented in degC but the precipitation biases presented as the percentage change? It would be preferable to use one approach consistently throughout the manuscript. In particular for the precipitation results, the % bias is often large when the observed precipitation is small – this is perhaps an instance where using the bias (MODEL minus OBS) would be more useful where small precipitation values inflate the value of the % bias.*

This approach is generally used in climate model validations, because this domain covers different climatic regimes. By doing so, we can better compare regions. This has not explicitly been clarified in the manuscript.

*Page 7 last sentence: I don't agree with this, why would SSTs only be influential in winter?*

We have removed statements like this, as discussed earlier. As we did not test the sensitivity of the model simulations to the different SSTs update frequency, we are not able to relate directly the differences in winter precipitation to differences in SSTs update frequency. This is outside the scope of this study, and thus has not been explicitly clarified in the revised manuscript.

*There is a tendency to start sentences with "Similarly to Experiment X, Experiment Y . . ." please use either: "Similar to Experiment X, Experiment Y. . ." or just start with "Experiment Y . . ."*

Done; this wording has been improved in the revised manuscript.
One example (Page 9 Line 20) reads:
" Similar to Mid-Europe, ..."

*Sentences such as: "FS (CON) overestimated (underestimated) the summer 2 m tem-*

[Figure]

*perature" are really hard to read and understand. It is actually easier to read: "FS overestimated and CON underestimated the summer 2 m temperature". Please revise all instances of this.*

Done; this has been revised at all instances in the revised manuscript.
The new version (Page 9 Lines 15-16) reads:
"However, FS overestimated the summer 2 m temperature and CON and DRI underestimated the summer 2 m temperature."

Section 5.1 and 5.2:

*Replace: "We assume two-way interactions" with "There are two-way interactions"*

This sentence and the related analysis have been removed from the revised manuscript.

*At no point do we know the depth of the soil layers in the land surface scheme. This would be useful to know.*

Yes indeed, this will be used for a separate, future study where we will analyse the correlations in more detail. We have removed this statement from the revised manuscript.

*It is not clear to me why the land-atmosphere feedback is evaluated by calculating the soil moisture temperature (ST-T) correlation using (FS minus CON / CON). It would be better to calculate the SM-T correlation for FS, the SM-T correlation for CON and then the difference between these two estimates. I think this analysis needs to be redone. Lorenz et al. (2015) provides a good description of different coupling metrics that could be applied to the data.*

The soil-moisture-temperature correlation will be content for a separate, future study,

as the current study primarily focuses on the simulation potential of different downscaling experiments instead of looking at correlations between variables. This has been clarified in the revised manuscript.

The new version (Page 12 Lines 25-27) reads:

"A more in-depth analysis on the interaction between 2 m temperature, precipitation, and surface energy fluxes can reveal soil-moisture-temperature coupling (Jaeger et al., 2009), but this lies outside the scope of this study."

*Here I also think that the spin-up length will have some influence on evaluating the land-atmosphere feedback. If CON has a longer spin-up than FS of the soil moisture and soil temperature then the results are surely already biased as the land surface state fields will be more resolved in the simulations with the longer spin-up.*

We have shown that three months of spin-up is reasonable for a good equilibrium. Accordingly we concluded that the different spin-up periods did not influence the land-atmosphere feedback. However, the land-atmosphere feedback has no longer been evaluated in the revised manuscript.

Conclusions:

*Page 11 Line 31: This definition of 'continuous' and 'reinitialized' simulation should be defined much earlier in the manuscript!*

Done. The definition has been provided much earlier in the manuscript.

*Page 11 Line 7: "The differences in 2 m temperature and precipitation between the downscaling setups during summer are demonstrated by an interaction with the soil moisture." I don't agree with this because it's not actually calculated for the 10 year simulations where the temperature and precipitation differences are evaluated.*

We agree with the reviewer's comment. As the surface fluxes have been calculated for 10 years in the revised manuscript, we can make more reliable statements on this. This has not explicitly been clarified in the Conclusions section.

Tables and Figures

*Table 1 – perhaps remove the CON-FS line or separate into two.*

Table 1 has been removed from the revised manuscript.

*Table 3 – it would be easier if these values were presented as mm day-1 rather than the relative bias because some values are very high but might only be so because these are regions where the precipitation is very low: e.g. MD regions DJF FS experiment.*

As a matter of fact the difference in MD for DJF is high. We have decided to maintain this relative bias in the revised manuscript as explained earlier. See Figure 3 in the revised manuscript.

*Figure 2 – it would be good to include statistical significance as done in Figure 7; perhaps update the labels in the top left hand corner of panels c to h with CRDX –E-OBS*

Done; the statistical significance has been included in the revised manuscript. See Figure 3 and 4 in the revised manuscript.

*Figure 3 – please put all panels in the same units it makes it easier to compare panels c to h with a and b. Is there missing data over Africa where there is a weird white triangle shape at the bottom of all panels? It is obvious in this figure that there are boundary affects for the domain and that the figures have not been cropped to exclude the relaxation zone.*

Different units are used for the absolute precipitation presented by E-OBS and the precipitatiob bias of the model with E-OBS. No precipitation has been measured over this triangle in Africa. This also appears in the EURO-CORDEX output. The relaxation zone has now been excluded from the revised manuscript as rightly requested by the reviewer. See Figure 3, 4 and 6 in the revised manuscript.

*Figure 4 – if the authors choose to keep the CRDX simulations then they should be included here and other figures. It would also be handy if dashed lines were added to each panel to delineate the seasonal breaks referred to in the manuscript text.*

Figure 4 has been removed from the revised manuscript.

*Figure 5 – obvious boundary affect in all panels. Why is the absolute difference used for temperature but the relative difference used for precipitation and soil moisture? It would be easier if they all presented in the same way.*

Figure 5 has been removed from the revised manuscript.

*Figure 7 – This needs to be redone to show SM-T and SM-P correlations for each experiment separately and then their difference.*

The suggested analysis will be done in a separate, future study.

*Figure 8 – It would be easier if FLUXNET, CON and FS were all on the same panel to directly compare differences. QS looks very different between the observations and CON and FS, are the authors certain that they are comparing like for like? QS looks like a flat line in panel c – check if there is a plotting error but this may just be because QS is very small relative to the axis scale or that it is not measured. . .*

This is a good suggestion. In the revised manuscript separate panels are shown for

the different variables. See Figure 7 in the revised manuscript.

**3   Manuscript version with highlighted changes is supplemented.**

---

## Author Response (AR2)

**Author's Response to the Reviewer 1 Comments on "Reinitialised versus continuous regional climate simulations using ALARO-0 coupled to the land surface model SURFEXv5"**

Julie Berckmans[1,2], Olivier Giot[1,2], Rozemien De Troch[1,3], Rafiq Hamdi[1,3], Reinhart Ceulemans[2], and Piet Termonia[1,3]

[1]Royal Meteorological Institute, Brussels, Belgium
[2]Centre of Excellence PLECO (Plant and Vegetation Ecology), Department of Biology, University of Antwerp, Antwerp, Belgium
[3]Department of Physics and Astronomy, Ghent University, Ghent, Belgium

*Correspondence to:* Julie Berckmans (julie.berckmans@meteo.be)

**1    Reply to the Editor**

Dear Editor,

we have prepared a majorily improved version of the manuscript by incorporating all suggestions and critical comments raised by Reviewer 1. Please find attached our response to the referees' comments on our above mentioned manuscript, titled "Reinitialised versus continuous regional climate simulations using ALARO-0 coupled to the land surface model SURFEX". Below mentioned you will find our detailed responses to all the reviewers' comments and suggestions (put in italics and red). We have also explained where and how they were incorporated in the revised manuscipt.

**2    Reply to Anonymous Referee #1**

We would like to thank the Anonymous Referee #1 for the encouraging and constructive comments, which have improved the manuscript. Below is a list of modifications that we have implemented based on your comments.

Comments:

*(1) While the reporting of the results has vastly improved there is still very limited discussion on why certain results are obtained with respect to particular simulation configurations. For example why does FS simulate better temperatures during summer? Is this because this simulation benefits from soil moisture memory by allowing the land surface to be fully interactive? The reduced performance in all configurations for precipitation is explained by the wet bias of the forcing data but perhaps more needs to be said on why FS is particularly worse than DRI and CON in winter.*

The discussion has been extended in the revised manuscript. The discussion points were either based on own findings, or supported by other literature.

The new version (Page 7 Lines 4-7) reads:
"This is due to compensating effects, as the bias represents an average over the subdomain and might be the result of large negative and large positive biases over different parts of the particular subdomain compensating each other. However, the area-averaged bias gives a good impression of the ranking of the experiments (Kotlarski et al., 2014)."

and (Page 7 Lines 13-14):
"The frequent reinitialisations keep the large scales closer to the ERA-Interim forcing, whereas ALARO and ARPEGE are bound to a cold bias (Voldoire et al., 2013; Giot et al., 2016)."

and (Page 7 Lines 19-20):
"The Alps were characterised by a zero bias on the northern flank and mixed cold and warm bias on the southern flank compensating each other (Table 2)"

and (Page 7 Lines 22-24):
"These positive biases for FS might be related to rapidly decreasing soil moisture values in spring and summer (not shown). The temperature-soil moisture relation is strongest for FS, as this simulation benefits from soil moisture memory by allowing the land surface to be fully interactive with the atmosphere (Koster and Suarez, 2001)."

and (Page 8 Lines 7-10 ):
"More specifically, the overestimation of winter precipitation was strongest in the Mediterranean and Eastern Europe with values from 35.3% to 108.5% for all simulation modes (Table 3). The large values in the Mediterranean agreed with the large underestimation of 2 m temperature, as this region is characterised by a strong dependence of temperature and precipitation (Faggian, 2015)."

and (Page 8 Lines 11-12):
"The too wet driving field of ERA-Interim was superimposed on the smaller cold bias of FS, suggesting a higher precipitation bias than CON and DRI."

and (Page 8 Lines 22-24):
"Consequently, the summer precipitation was simulated better by FS than CON and DRI. During summer, the influence of the soil moisture memory on the atmosphere is more important, resulting in an improved representation of the precipitation with

FS."

and (Page 8 Lines 29-30):

"Frequent reinitialisations did not allow the land surface to build up a soil moisture memory, resulting in less skill for the representation of the precipitation."

and (Page 9 Lines 26-28):

"The dry climate of the Iberian Peninsula is less dominated by land surface-atmosphere interactions, as soil moisture does not impact the evapotranspiration availability (Seneviratne et al.,2010)."

*(2) I was really concerned about the massive bias in the ground heat flux. I think the authors were too quick to dismiss this and should say more about why this bias persists particularly because it ranges from 40 to >100 W m-2. In particular, is the ground heat flux calculated or updated to include the residual energy imbalance (e.g. G = RN – H – LE) in order to maintain energy balance… because it certainly looks like it is. Perhaps something needs to be said on what steps will be taken next to improve the surface turbulent energy fluxes and their partitioning because the biases for the surface energy balance are quite large. It looks like including (or improving) a soil resistance for soil evaporation and a stomatal resistance for transpiration may improve the excessive evaporation in the model.*

We agree that the biases are too large and need to be improved in order to represent well the surface energy budget. A detailed study by Napoly et al. (2016) confirmed the large bias for the ground heat flux in ISBA. The authors presented new parameterisations for a multi-source model, which resulted in a reduced ground heat flux and more energy available for the turbulent fluxes. This suggestion is included in our study.

The new version (Page 11 Lines 15-26 ) reads:

"The ground heat flux (G) showed improbably high values compared to the observed ones (Table 4). G is dependent on the soil temperature, which was largely overestimated by the land surface model (not shown). The standard version of ISBA, the nature tile of SURFEX, aggregates soil and vegetation properties for each grid cell (Noilhan and Planton, 1989). The net radiation is directly transferred to the ground, causing an inaccurate partitioning of the incoming energy into turbulent and ground heat fluxes (Napoly et al., 2016). An additional parameterisation for the leaf litter on the surface soil impacts this distribution (Wilson et al., 2012). We suggest to include an explicit formulation of the canopy layer (Napoly et al.,2016) and potentially a parameterisation for the forest litter layer (Napoly et al., 2016; Wilson et al., 2012). The implementation of these explicit formulations in ISBA outperformed the representation of the soil temperature of the original ISBA model (Napoly et al., 2016). They showed that the original ISBA model overestimated the G flux amplitude with several 10's of W m-2 during both daytime and nighttime. However, using the distinct surface energy budgets resolved part of the overestimated G by intercepting most of the downward solar radiation, leaving more energy available for turbulent fluxes. Consequently, less net radiation reaches the forest surface, reducing the energy available for the soil conductance Napoly et al. (2016)."

*(3) In the first review I suggested to augment the validation of the surface energy balance with gridded observational products such as GLEAM or LandFlux. I'm not convinced by the reason given as to why this request was dismissed. It is difficult to evaluate the skill of the land surface fluxes at the grid cell values from a model that represents here a 20 km2 grid cell average to point observations, particularly if the vegetation 'mix' contrasts to that of the point observations. The biases are quite large and it would be good to check if this is associated with comparing grid cell averages to a point observation and whether the model can at least capture the spatial variability of the domain. Comparing against two points is not very rigorous and given how large the biases are, avoiding comparison to products such as the MERRA reanalysis, GLEAM or LandFlux really undermines the credibility of this model validation study. Could you look at how well the model captures the anomalies to assess more critically the ability of the model to simulate the temporal variability?*

We have to admit to dismiss this suggestion to easily last time. The biases of the two stations are indeed too large to build conclusions on this. We have added 5 FLUXNET sites to the validation of the spatial distribution of Bowen Ratio. This gives a better representation of the spatial variability by the model. Even though these 5 stations represent the correct land cover by less than 50%, they are helpful in providing contrasts. The comparison of model grid cell averages to FLUXNET observations was also done by Blyth et al. (2010), Stöckli et al. (2008).

The new version (Page 6 Lines 21-25) reads:

"The model validation was done using grid cell averages compared to point observations, suggesting large differences in the land cover representation. In total, a subset of 7 stations that cover different biome types (Table 1), was selected to demonstrate the spatial variability of the domain by the model. However, the main focus was on the Vielsalm and Collelongo sites (Fig. 1), as their model grid cells represent more than 50% of the corresponding land cover, and cover different climate regimes."

and (Page 12 Lines 8-9):

"In summary, the model presented a good spatial variability of BR, and the agreement with observations was highest for FS. "

In addition, we compared the evapotranspiration (ET) of the model with the LandFLUX product (Fig. **??**). We calculated ET in the model based on the latent heat flux (LE). Next, we aggregated the hourly values to daily values, in order to be comparable with the satellite product that provides daily values based on monthly values. The model data at 20x20 km were upscaled to 1x1 degree, and regridded to the projection of LandFLUX. The results confirm our findings. CON overestimates ET/LE, and FS underestimates ET/LE compared to LandFLUX. DRI provides the best agreement with the observations. Besides, the spatial variability of the model is comparable with the observed one. We will not add this figure to the paper, as it requires more in-depth analysis and a proper description on the features of the LandFLUX dataset. This would serve as content for a follow-up manuscript. We would like to keep the focus of the manuscript to the objective on the potential of the simualation modes. However, we appreciate the author's suggestion on validating with a gridded observational product, as this certainly adds value to point observations.

[Figure]

(a) LandFLUX

(b) CON

(c) DRI

(d) FS

Figure 1: Fig 1. The daily evapotranspiration (in mm) during JJA of [a] LandFLUX, [b] CON, [c] DRI, [d] FS, over Europe with a horizontal resolution of 1 x 1 degree, over a 10-year period.

**v**

*(4) There are still sections of text where the language / sentence composition is awkward. Rather than note them all here I attach these comments in an annotated PDF of the revised manuscript. These are my suggestions for further refinement of the manuscript text and include some minor requests for further clarification.*

We appreciate the reviewer's suggestions for the further refinement of the manuscript. They have been included. The revised manuscript with marked up changes has been added as attachment.

**3 Manuscript version with highlighted changes is supplemented.**

[revised manuscript text omitted]
 water and energy exchanges within the residue layer , their surface model overestimated LE, G and soil temperature and underestimated H. As the net radiationand ground heat flux were simulated very similarly for all simulations, they were not shown in Fig. 7. the canopy layer (Napoly et al., 2016) and potentially a parameterisation for the forest litter layer (Napoly et al., 2016; Wilson et al., 2012). The implementation of these explicit formulations in ISBA outperformed the representation of the soil temperature of the original ISBA model (Napoly et al., 2016). They showed that

10   the original ISBA model overestimated the G flux amplitude with several 10's of W m$^{-2}$ during both daytime and nighttime. However, using the distinct surface energy budgets resolved part of the overestimated G by intercepting most of the downward solar radiation, leaving more energy available for turbulent fluxes. Consequently, less net radiation reaches the forest surface, reducing the energy available for the soil conductance Napoly et al. (2016).

For Vielsalm, H was simulated well by DRI and FS during nighttime and daytime, whereas CON underestimated H during

15   daytime (Fig. 7a). The daily maximum H by CON was only 118 W m$^{-2}$, as compared to 151 and 139 W m$^{-2}$ for DRI and FS respectively (Table 4). Yet again, this This validation was only done for 5 summer periods from 1996 to 2000, but the corresponding daily maximum values for the 10-year summer period 1991-2000 indicated indicate that the 5-year period was representative for the validation of the fluxes (Table 4). The LE was overestimated by all simulations, but the difference with the observations was smallest for DRI, while it was highest due to the frequent land surface reinitialisations, compared to

20   highest values for CON. The daily maximum BR was lower than 1 for all downscaling approaches (Table 4). This means that they all simulated a higher latent than sensible heat flux. Still, DRI and FS showed higher values for BR than CON. Therefore, the partitioning of the surface energy fluxes was better represented by DRI and FS for the station of Vielsalm.

For Collelongo, H was underestimated by the model the model underestimated H during daytime and overestimated H during nighttime, except for DRI which demonstrated a good agreement with the observations. Yet againConsequently, the

25   model overestimated LE during daytime, except for DRI. The daily maximum H for DRI of 247 W m$^{-2}$ was close to the observed value of 253 W m$^{-2}$, whereas CON and FS simulated much lower values of 159 W m$^{-2}$ and 197 W m$^{-2}$ respectively (Table 4). The simulated LE CON showed the largest difference with the observed one using CONLE bias. Regarding BR, the simulation by DRI with a value of 1.35 was in very good agreement with the observations. The DRI simulation resulted in the correct least biased partitioning of the surface energy fluxes at Collelongo. CON was not performing well in simulating the

30   correct partitioning, while FS had already much improved as compared to CONHowever, 
[revised manuscript text omitted]